# $\lambda$: Effective decision-aware reinforcement learning with latent models

## ABSTRACT

The idea of decision-aware model learning, that models should be accurate where it matters for decision-making, has gained prominence in model-based reinforcement learning. While promising theoretical results have been established, the empirical performance of algorithms leveraging a decision-aware loss has been lacking, especially in continuous control problems. In this paper, we present a study on the necessary components for decision-aware reinforcement learning models and we showcase design choices that enable well-performing algorithms. To this end, we provide a theoretical and empirical investigation into algorithmic ideas in the field. We highlight that empirical design decisions established in the MuZero line of works, most importantly the use of a latent model, are vital to achieving good performance for related algorithms. Furthermore, we show that the MuZero loss function is biased in stochastic environments and establish that this bias has practical consequences. Building on these findings, we present an overview of which decision-aware loss functions are best used in what empirical scenarios, providing actionable insights to practitioners in the field.

## 1 INTRODUCTION

In model-based reinforcement learning, an agent collects information in an environment and uses it to learn a model of the world to accelerate and improve value estimation or the agent's policy (Sutton, 1990; Deisenroth & Rasmussen, 2011; Hafner et al., 2020; Schrittwieser et al., 2020). However, as environment complexity increases, learning a model becomes more and more challenging. These model errors can impact the learned policy (Schneider, 1997; Kearns & Singh, 2002; Talvitie, 2017; Lambert et al., 2020) leading to worse performance. In many realistic scenarios, agents are faced with a "big world" where learning to accurately simulate the environment is infeasible. When faced with complex environments, deciding what aspects of the environment to model is crucial. Otherwise, capacity would be spent on irrelevant aspects, e.g., modelling clouds in a vehicle's video feed.

Recently, the paradigm of *decision-aware model learning* (DAML) (Farahmand et al., 2017) or *value equivalence* (Grimm et al., 2020; 2021) has been proposed. The core idea is that a model should make predictions that result in correct value estimation. The two most prominent approaches in the space of decision-aware model learning are the IterVAML (Farahmand, 2018) and MuZero (Schrittwieser et al., 2020) algorithms. While IterVAML is a theoretically grounded algorithm, difficulties in adapting the loss for empirical implementations have been highlighted in the literature (Lovatto et al., 2020; Voelcker et al., 2022). MuZero on the other hand has been shown to perform well in discrete control tasks (Schrittwieser et al., 2020; Ye et al., 2021) but has received little theoretical investigation. Thus, understanding the role of different value-aware losses and determining the factors for achieving strong performance is an open research problem.

**Research question:** In this work, we ask three question: (a) What design decisions explain the performance difference between IterVAML and Muzero? (b) What are theoretical or practical differences between the two? (c) In what scenarios do decision-aware losses help in model-based reinforcement learning? Experimental evaluations focus on the state-based DMC environments (Tunyasuvunakool et al., 2020), as the focus of our paper is not to scale up to image-based observations.

**Contributions:** The main differences between previous works trying to build on the IterVAML and MuZero algorithms are neural network architecture design and the value function learning scheme.

MuZero is built on value prediction networks (Oh et al., 2017), which explicitly incorporate a latent world model, while IterVAML is designed for arbitrary model choices. We show that the use of latent models can explain many previously reported performance differences between MuZero and IterVAML (Voelcker et al., 2022; Lovatto et al., 2020).

As a theoretical contribution, we first show that IterVAML and MuZero-based models can achieve low error in stochastic environments, even when using deterministic world models, a common empirical design decision (Oh et al., 2017; Schrittwieser et al., 2020; Hansen et al., 2022). To the best of our knowledge, we are the first to prove this conjecture and highlight that it is a unique feature of the IterVAML and MuZero losses. The MuZero value function learning scheme however results in a bias in stochastic environments. We show that this bias leads to a quantifiable difference in performance. Finally, we show that the model learning component of MuZero is similar to VAML, which was previously also noted by Grimm et al. (2021).

In summary, our contributions are a) showing that IterVAML is a stable loss when a latent model is used, b) proving and verifying that the MuZero value loss is biased in stochastic environments, and c) showing how these algorithms provide benefit over decision-agnostic baseline losses.

## 2 BACKGROUND

**Reinforcement Learning:**   We consider a standard Markov decision process (MDP) (Puterman, 1994; Sutton & Barto, 2018) $(\mathcal{X}, \mathcal{A}, p, r, \gamma)$, with state space $\mathcal{X}$, action space $\mathcal{A}$, transition kernel $p(x'|x, a)$, reward function $r : \mathcal{X} \times \mathcal{A} \to \mathbb{R}$, and discount factor $\gamma \in [0, 1)$. The goal of an agent is to optimize the obtained average discounted infinite horizon reward under its policy: $\max_\pi \mathbb{E}_{\pi, p} \left[ \sum_{t=0}^\infty \gamma^t r(x_t, a_t) \right]$.

Given a policy $\pi(a|x)$, the value function is defined as the expected return conditioned on a starting state $x$: $V^\pi(x) = \mathbb{E}_\pi \left[ \sum_{t \geq 0} \gamma^t r_t | x_0 = x \right]$, where $r_t = r(x_t, a_t)$ is the reward at time $t$. The value function is the unique stationary point of the *Bellman operator* $\mathcal{T}_p(V)(x) = \mathbb{E}_{\pi(a|s)} \left[ r(x, a) + \gamma \mathbb{E}_{p(x'|x, a)} \left[ V(x') \right] \right]$. The action-value function is defined similarly as $Q^\pi(x, a) = \mathbb{E}_\pi \left[ \sum_{t \geq 0} \gamma^t r_t | x_0 = x, a_0 = a \right]$. In this work, we do not assume access to the reward functions and denote learned approximation as $\hat{r}(x, a)$.

**Model-based RL:**   An *environment model* is a function that approximates the behavior of the transition kernel of the ground-truth MDP.[1] Learned models are used to augment the reinforcement learning process in several ways (Sutton, 1990; Janner et al., 2019; Hafner et al., 2020), we focus on the MVE (Buckman et al., 2018) and SVG (Amos et al., 2021) algorithms presented in details in Subsection 2.2. These were chosen as they fit the decision-aware model learning framework and have been used successfully in strong algorithms such as Dreamer (Hafner et al., 2021).

We will use $\hat{p}$ to refer to a learned probabilistic models, and use $\hat{f}$ for deterministic models. When a model is used to predict the next ground-truth observation $x'$ from $x, a$ (such as the model used in MBPO (Janner et al., 2019)) we call it an *observation-space models*. An alternative are *latent models* of the form $\hat{p}(z'|z, a)$, where $z \in \mathcal{Z}$ is a representation of a state $x \in \mathcal{X}$ given by $\phi : \mathcal{X} \to \mathcal{Z}$. In summary, observation-space models seek to directly predict next states in the representation given by the environment, while latent space models can be used to reconstruct learned features of the next state. We present further details on latent model learning in Subsection 3.1.

The notation $\hat{x}_i^{(n)}$ refers to an $n$-step rollout obtained from a model by starting in state $x_i$ and sampling the model for $n$-steps. We will use $\hat{p}^n \left( x^{(n)} | x \right)$ and $\hat{f}^n(x)$ to refer to the $n$-th step model prediction.

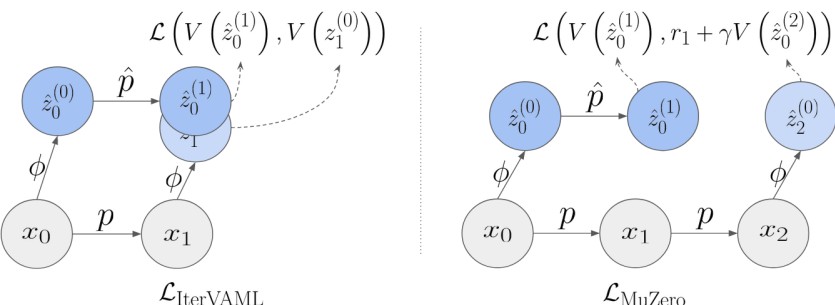

Figure 1: Sketch of the different value-aware losses with latent models. IterVAML computes the value function difference between the latent prediction and the next state encoding, while MuZero computes a single-step bootstrap estimate.

## 2.1 DECISION-AWARE MODEL LOSSES

The losses of the decision-aware learning framework share the goal of finding models that provide good value function estimates. Instead of simply learning a model using maximum likelihood estimation, the

losses are based on differences in value prediction. We present the loss functions below to highlight the close relationship as well as crucial differences between the two. We assume a [2] standard dataset of state, action, and reward sequences $\mathcal{D} = \{x_{i_1}, a_{i_1}, r_{i_1}, \ldots, x_{i_n}, a_{i_n}, r_{i_n}\}_{i=1}^N$ with $x_{i_1}$ sampled from $\mu$ and subsequent states sampled according to the transition function. $N$ denotes the number of samples, and $n$ the sequence length. As all considered algorithms are off-policy, the actions can come from different past policies. Note that the index 1 here denotes the start of a (sub-)sequence, not the start of an episode.

**IterVAML (Farahmand, 2018):** To compute the IterVAML loss, value function estimates are computed over the model prediction and a real sample drawn from the training distribution. The loss minimizes the squared difference between them. While the original paper only considers a single-step variant, we present a multi-step version of the loss (for a discussion on the differences, see Subsection A.5). The resulting IterVAML loss is

$$\hat{\mathcal{L}}_{\text{IterVAML}}^n(\hat{p}; \hat{V}, \mathcal{D}) = \frac{1}{N \cdot n} \sum_{i=1}^N \sum_{j=1}^n \left| \underbrace{\mathbb{E}_{\hat{x}^{(j)} \sim \hat{p}^j(\cdot | x_{i_1}, a_{i_1})} \left[ \hat{V}\left(\hat{x}^{(j)}\right) \right]}_{\text{Model value estimate}} - \underbrace{\hat{V}\left(x_{i_j}\right)}_{\text{Ground truth value estimate}} \right|^2.$$

(1)

**MuZero (Schrittwieser et al., 2020):** MuZero proposes a similar loss. However, the next state value function over the real sample is estimated via an approximate application of the Bellman operator. This core difference between both losses is illustrated in Figure 1. Similar to Mnih et al. (2013), MuZero uses a target network for the bootstrap estimate, which we denote as $V_{\text{target}}$. The MuZero loss is introduced with using deterministic models, which is why we present it here as such. The extension to stochastic models is non-trivial, as we show in Subsection 4.2 and discuss further in Subsection A.2. The loss is written as

$$\hat{\mathcal{L}}_{\text{MuZero}}^n\left(\hat{f}, \hat{V}; \mathcal{D}, V_{\text{target}}\right) = \frac{1}{N \cdot n} \sum_{i=1}^N \sum_{j=i}^n \left| \underbrace{\hat{V}\left(\hat{f}^j(x_{i_1}, a_{i_1})\right)}_{\text{Model value estimate}} - \underbrace{\left[ r_{i_j} + \gamma V_{\text{target}}\left(x_{i_{j+1}}\right) \right]}_{\text{Ground truth bootstrap value estim}} \right|$$

(2)

---

[1] In this paper, we will generally use the term *model* to refer to an environment model, not to a neural network, to keep consistent with the reinforcement learning nomenclature.

[2]

Note that the real environment reward and next states are used in the $j$-th step. The MuZero loss is also used for updating both the value function and the model, while IterVAML is only designed to update the model and uses a model-based bootstrap target to update the value function.

**Stabilizing loss:** Decision-aware losses can be insufficient for stable learning (Ye et al., 2021), especially in continuous state-action spaces (Hansen et al., 2022). Their performance is improved greatly by using a *stabilizing or auxiliary loss*. In practice, latent self-prediction, based on the *Bootstrap your own latent* (BYOL) loss (Grill et al., 2020), has shown to be a strong and simple option for this task. Several different versions of this loss have been proposed for RL (Gelada et al., 2019; Schwarzer et al., 2021; Tang et al., 2022), we consider a simple $n$-step variant:

$$\hat{\mathcal{L}}_{\text{latent}}^n\left(\hat{f}, \phi; \mathcal{D}\right) = \frac{1}{N \cdot n} \sum_{i=1}^N \sum_{j=1}^n \left[\hat{f}^{(j)}(\phi(x_{i_1}), a_{i_1}) - \text{stop-grad}\left[\phi(x_{i_j})\right]\right]^2. \tag{3}$$

Theoretical analysis of this loss by Gelada et al. (2019) and Tang et al. (2022) shows that the learned representations provide a good basis for value function learning.

## 2.2 ACTOR-CRITIC LEARNING

**Value leaning:** Both IterVAML and MuZero can use the model to approximate the value function target. In its original formulation, MuZero is used with a Monte Carlo Tree Search procedure to expand the model which is not directly applicable in continuous state spaces. A simple alternative for obtaining a value function estimate from the model in continuous state spaces is known as model value expansion (MVE) (Feinberg et al., 2018). In MVE, short horizon roll-outs of model predictions $[\hat{x}^{(0)}, \ldots, \hat{x}^{(n)}]$ are computed using the current policy estimate, with a real sample from the environment as a starting point $x^{(0)}$. Using these, $n$-step value function targets are computed $Q_{\text{target}}^j\left(\hat{x}^{(j)}, \pi\left(\hat{x}^{(j)}\right)\right) = \sum_{i=j}^{n-1} \gamma^{i-j} \hat{r}\left(\hat{x}^{(i)}, \pi\left(\hat{x}^{(i)}\right)\right) + \gamma^{n-j} \bar{Q}\left(\hat{x}^{(n)}, \pi\left(\hat{x}^{(n)}\right)\right)$. These targets represent a bootstrap estimate of the value function starting from the $j$-th step of the model rollout. For our IterVAML experiments, we used these targets together with TD3 (Fujimoto et al., 2018) to learn the value function, for MuZero, we used these targets for $V_{\text{target}}$ in Equation 2.

**Policy learning:** In continuous control tasks, estimating a policy is a crucial step for effective learning algorithms. To improve policy gradient estimation with a model, stochastic value gradients (SVG) (Heess et al., 2015; Amos et al., 2021) can be used. These are obtained by differentiating the full $n$-step MVE target with regard to the policy. As most common architectures are fully differentiable, this gradient can be computed using automatic differentiation, which greatly reduces the variance of the policy update compared to traditional policy gradients. We evaluate the usefulness of model-based policy gradients in Section 5, with deterministic policy gradients (Silver et al., 2014; Lillicrap et al., 2016) as a model-free baseline.

3

# 3 LATENT DECISION AWARE MODELS

Previous work in decision-aware MBRL (Farahmand et al., 2017; Farahmand, 2018; Schrittwieser et al., 2020; Grimm et al., 2021) uses differing model architectures, such as an ensemble of stochastic next state prediction networks in the case of Voelcker et al. (2022) or a deterministic latent model such as in Schrittwieser et al. (2020); Hansen et al. (2022). In this section, we highlight the importance of using latent networks for decision-aware losses. We call these variants of the investigate algorithms *LAtent Model-Based Decision-Aware (LAMBDA)* algorithms, or $\lambda$-IterVAML and $\lambda^2$

-MuZero for short, to stress the importance of combining latent models with decision-aware algorithms. We focus on IterVAML and MuZero as the most prominent decision-aware algorithms.

---

### 3.1 THE IMPORTANCE OF LATENT SPACES FOR APPLIED DECISION-AWARE MODEL LEARNING

One of the major differences between MuZero and IterVAML is that the former uses a latent model design, while the IterVAML algorithm is presented without a specific architecture. When used with a state-space prediction model, IterVAML has been shown to diverge, leading to several works claiming that it is an impractical algorithm (Lovatto et al., 2020; Voelcker et al., 2022). However, there is no a priori reason why a latent model should not be used with IterVAML, or why MuZero has to be used with one. In prior work, sharp value function gradients have been hypothesized to be a major concern for IterVAML (Voelcker et al., 2022). The introduction of a learned representation space in the model through a latent embedding can help to alleviate this issue.

The full loss for each experiment is $\mathcal{L}^n_{\text{MuZero/IterVAML}} + \mathcal{L}^n_{\text{latent}} + \mathcal{L}^n_{\text{reward}}$, where $\mathcal{L}_{\text{reward}}$ is an MSE term between the predicted and ground truth rewards. As a baseline, we drop the decision-aware loss and simply consider BYOL alone. For all of our experiments we use eight random seeds and the shaded areas mark three $\sigma$ of the standard error of the mean over random seeds, which represents a 99.7% certainty interval under a Gaussian assumption. Full pseudocode is found in Subsection C.2.

As is evident from **??** the latent space is highly beneficial to achieve good performance. In higher dimensional experiments and without stabilizing losses, we find that IterVAML diverges completely (see Appendix D) This highlights that negative results on the efficacy of IterVAML are likely due to suboptimal choices in the model implementation, not due to limitations of the algorithm.

## 4 ANALYSIS OF DECISION-AWARE LOSSES IN STOCHASTIC ENVIRONMENTS

After accounting for model implementation, both MuZero and IterVAML behave similarly in experiments. However, it is an open question if this holds in general. Almost all commonly used benchmark environments in RL are deterministic, yet stochastic environments are an important class of problems both for theoretical research and practical applications. Therefore, we first present a theoretical investigation into the behavior of both the MuZero and IterVAML losses in stochastic environments, and then evaluate our findings empirically. Proofs are found in Subsection A.1.

In Proposition 1 we establish that $\lambda$-IterVAML leads to an unbiased solution in the infinite sample limit, even when restricting the model class to deterministic functions under measure-theoretic conditions on the function class. This is a unique advantage of value-aware models. Algorithms such as Dreamer (Hafner et al., 2020) or MBPO (Janner et al., 2019) require stochastic models, because they model the complete next-state distribution, instead of the expectation over the next state.

We then show in Proposition 2 that MuZero's joint model- and value function learning algorithm leads to a biased solution, even when choosing a probabilistic model class that contains the ground-truth environment. This highlights that while the losses are similar in deterministic environments, the same does not hold in the stochastic case. Finally, we verify the theoretical results empirically by presenting stochastic extensions of environments from the DMC suite (Tunyasuvunakool et al., 2020).

### 4.1 RESTRICTING ITERVAML TO DETERMINISTIC MODELS

In most cases, deterministic function approximations cannot capture the transition distribution on stochastic environments. However, it is an open question whether a deterministic model is sufficient for learning a *value-equivalent* model, as conjectured by (Oh et al., 2017). We answer this now in the affirmative. Showing the existence of such a model relies on the continuity of the transition kernel and involved functions $\phi$ and $V$.[2]

**Proposition 1** *Let $\mathcal{X}$ be a compact, connected, metrizable space. Let $p$ be a continuous kernel from $\mathcal{X}$ to probability measures over $\mathcal{X}$. Let $\mathcal{Z}$ be a metrizable space. Consider a bijective latent mapping $\phi : \mathcal{X} \to \mathcal{Z}$ and any $V : \mathcal{Z} \to \mathbb{R}$. Assume that they are both continuous. Denote $V_{\mathcal{X}} = V \circ \phi$.*

*Then there exists a measurable function $f^* : \mathcal{Z} \to \mathcal{Z}$ such that we have $V(f^*(\phi(x))) = \mathbb{E}_p[V_{\mathcal{X}}(x')|x]$ for all $x \in \mathcal{X}$.*

---

[2]To reduce notational complexity, we ignore the action dependence in all following propositions; all results hold without loss of generality for the action-conditioned case as well.

*Furthermore, the same $f^*$ is a minimizer of the expected IterVAML loss:*

$$f^* \in \arg\min_{\hat{f}} \mathbb{E}\left[\hat{\mathcal{L}}_{IterVAML}(\hat{f}; V_\mathcal{X})\right].$$

We can conclude that given a sufficiently flexible function class $\mathcal{F}$, IterVAML can recover an optimal deterministic model for value function prediction. Note that our conditions solely ensure the *existence* of a measurable function; the learning problem might still be very challenging. Nonetheless, even without the assumption that the perfect model $f$ is learnable, the IterVAML loss finds the function that is closest to it in the mean squared error sense, as shown by Farahmand (2018).

## 4.2 SAMPLE BIAS OF MUZERO' VALUE FUNCTION LEARNING ALGORITHM

MuZero is a sound algorithm for deterministic environments, however, the same is not true for stochastic ones. While changed model architecture for MuZero have been proposed for stochastic cases (Antonoglou et al., 2022), the value function learning component contains its own bias. To highlight that the problem is neither due to the $n$-step formulation nor due to suboptimal architecture or value function class, we show that the value function estimate does not converge to the model Bellman target using the MuZero loss. Intuitively, the problem results from the fact that two samples drawn from the model and from the environment do not coincide, even when the true model and the learned model are equal (see **??**). This bias is similar to the double sampling issue in Bellman residual minimization, but is distinct as the introduction of the stop-gradient in MuZero's loss function does not address the problem.

**Proposition 2** *Assume a non-deterministic MDP with a fixed, but arbitrary policy $\pi$, and let $p$ be the transition kernel. Let $\mathcal{V}$ be an open set of functions, and assume that it is Bellman complete: $\forall V \in \mathcal{V} : \mathcal{T}V \in \mathcal{V}$.*

*Then for any $V' \in \mathcal{V}$ that is not a constant function, $\mathcal{T}V' \notin \arg\min_{\hat{V} \in \mathcal{V}} \mathbb{E}_\mathcal{D}\left[\hat{\mathcal{L}}^1_{MuZero}(p, \hat{V}; \mathcal{D}, V')\right].$*

The bias indicates that MuZero will not recover the correct value function in environments with stochastic transitions, even when the correct model is used and the function class is Bellman complete. On the other hand, model-based MVE such as used in $\lambda$-IterVAML can recover the model's value function in stochastic environments.

The bias is dependent on the variance of the value function with regard to the transition distributions. This means that in some stochastic environments the MuZero loss might still perform well. But as the variance of the value function increases, the bias to impact the solution.

If the MuZero loss is solely used for model learning and the value function is learned fully model-free or model-based, the IterVAML and MuZero algorithms show strong similarities (compare Subsection A.2). The main difference is that MuZero uses a bootstrap estimate for the value function, while IterVAML uses the value function estimate directly. However, when jointly training value function and model in a stochastic environment, neither the model nor the value function converge to the correct solution, due to the tied updates.

## 4.3 EMPIRICAL VALIDATION OF THE PERFORMANCE IN STOCHASTIC ENVIRONMENTS

To showcase that the bias of MuZero's value learning strategy is not merely a mathematical curiosity, we tested the two losses in the challenging humanoid-run tasks from the DMC benchmarking suite (Tunyasuvunakool et al., 2020) with different noise distributions applied to the action (details on the environment can be found in Appendix C).

As shown in Figure 2, we do see a clear difference between the performance of $\lambda$-IterVAML and $\lambda$-MuZero when increasing the noise level. At small levels of noise, all algorithms retain their performance. This is expected, since small noise on the actions can increase the robustness of a learned policy or improve exploration (Hollenstein et al., 2022). At large levels of noise, however, the $\lambda$-MuZero algorithm drops in performance to the value-agnostic baseline. We provide additional experiments with more stochastic environments in Appendix D which further substantiate our claims.

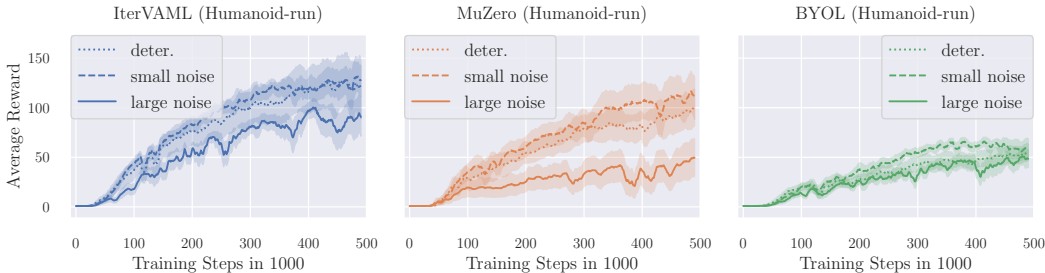

Figure 2: Comparison of the impact of noise across all algorithms. While both IterVAML and MuZero are impacted by the level of noise, we observe a larger drop in MuZero, which does not perform above the BYOL baseline at the highest noise level.

It is important to note that humanoid-run is a challenging environment and strong noise corruption increases difficulty, therefore $\lambda$-IterVAML will not retain the full performance in the stochastic version. Furthermore, as highlighted before, the stabilizing latent prediction loss is necessary for $\lambda$-IterVAML and introduces some level of bias (for details see Subsection A.4). However, as all algorithms are impacted by the stabilization term, it is still noticeable that $\lambda$-MuZero's performance drops more sharply. This raises an important direction for future work, establishing a mechanism for stabilizing value-aware losses that does not introduce bias in stochastic environments.

## 5 EVALUATING MODEL CAPACITY AND ENVIRONMENT CHOICE

After investigating the impact of action noise on different loss function, we now present empirical experiments to further investigate how different implementations of $\lambda$ algorithms behave in empirical settings. Theoretical results (Farahmand et al., 2017; Farahmand, 2018) show that value-aware losses perform best when learning a correct model is impossible due to access to finite samples or model capacity. Even though the expressivity of neural networks has increased in recent years, we argue that such scenarios are still highly relevant in practice. Establishing the necessary size of a model a priori is often impossible, since RL is deployed in scenarios where the complexity of a solution is unclear. Increasing model capacity often comes at greatly increased cost (Kaplan et al., 2020), which makes more efficient alternatives desirable. Therefore, we empirically verify under what conditions decision-aware losses show performance improvements over the simple BYOL loss.

For these experiments, we use environments from the popular DMC suite (Tunyasuvunakool et al., 2020), Walker-run, Quadruped-run, and Humanoid-run. These environments were picked from three different levels of difficulty (Hansen et al., 2022) and represent different challenge levels in the benchmark, with Humanoid-run being a serious challenge for most established algorithms.

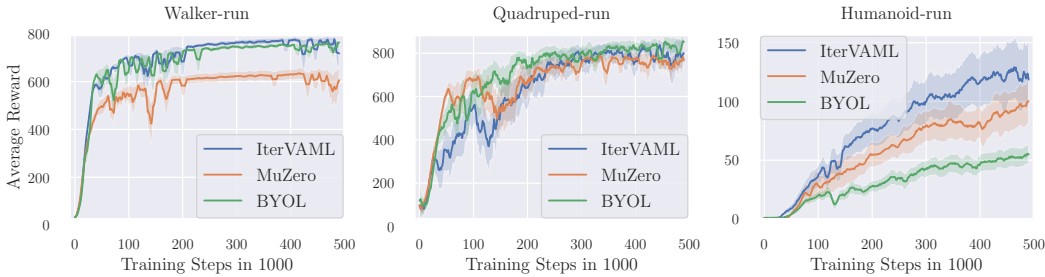

Figure 3: Performance comparison overall test environments. We see that IterVAML performs slightly above MuZero in several environments, but decisive gains from the value-aware losses ($\lambda$-IterVAML, $\lambda$-MuZero) over BYOL can only be observed in the challenging Humanoid-run environment.

The algorithmic framework and all model design choices are summarized in Appendix C. The model implementation follows Hansen et al. (2022). Reported rewards are collected during training, not in an evaluation step, the same protocol used by Hansen et al. (2022).

**When do decision-aware losses show performance improvements over the simple BYOL loss?**
The results in Figure 3 show that both MuZero and IterVAML provide a benefit over the BYOL loss in the most challenging environment, Humanoid-run. This is expected, as modelling the full state space of Humanoid-run is difficult with the model architectures we used. For smaller networks, we see a stable performance benefit in **??** from the MuZero loss. $\lambda$-IterVAML also outperforms the BYOL baseline, but fails to achieve stable returns.

This highlights that in common benchmark environments and with established architectures, value-aware losses are useful in challenging, high-dimensional environments. However, it is important to note that we do not find that the IterVAML or MuZero losses are harmful in deterministic environments, meaning a value-aware loss is always preferable.

The performance improvement of MuZero over IterVAML with very small models is likely due to the effect of using real rewards for the value function target estimation. In cases where the reward prediction has errors, this can lead to better performance over purely model-based value functions such as those used by IterVAML.

**Can decision-aware models be used for both value function learning and policy improvement?**
In all previous experiments, the models were used for both value function target estimation and for policy learning with a model-based gradient. To investigate the performance gain from using the model for policy gradient estimation, we present an ablation on all environments by substituting the model gradient with a simple deep deterministic policy gradient.

As seen in Figure 4, all losses and environments benefit from better policy estimation using the model. Therefore it is advisable to use a $\lambda$ model both for gradient estimation and value function improvement. Compared to MuZero, IterVAML loses more performance without policy gradient computation in the hardest evaluation task. It has been noted that model-based value function estimates might be more useful for policy gradient estimation than for value function learning (Amos et al., 2021; Ghugare et al., 2023). In addition, the grounding of the MuZero loss in real rewards likely leads to better value function prediction in the Humanoid-run environment. Therefore a better update can be obtained with MuZero when the model is not used for policy gradient estimation, since learning is driven by the value function update.

## 6 RELATED WORK

**VAML and MuZero:** Farahmand (2018) established IterVAML based on earlier work (Farahmand et al., 2017). Several extensions to this formulation have been proposed, such as a VAML-

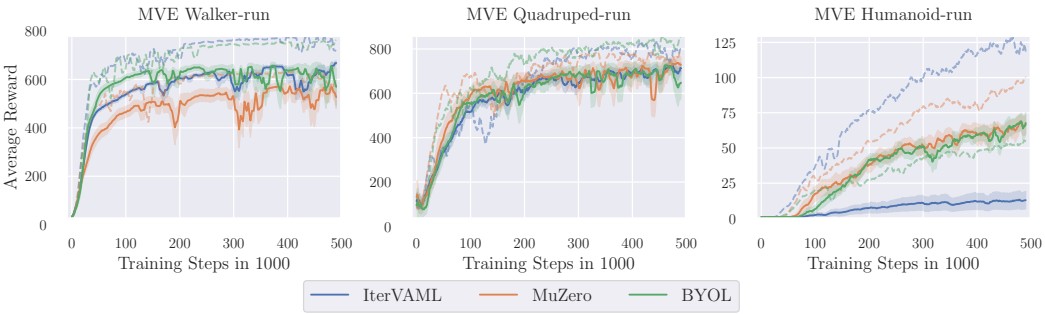

Figure 4: Comparison of the impact removing model policy gradients in all environments. We see decreases in performance across all environments and loss functions, with a drastic decrease in performance for IterVAML in humanoid. Dashed lines show the mean results of the model-based gradient for comparison.

regularized MSE loss (Voelcker et al., 2022) and a policy-gradient aware loss (Abachi et al., 2020). Combining IterVAML with latent spaces was first developed by Abachi et al. (2022), but no experimental results were provided. MuZero (Schrittwieser et al., 2020; Ye et al., 2021) is build based on earlier works which introduce the ideas of learning a latent model jointly with the value function (Silver et al., 2017; Oh et al., 2017). However, none of these works highlight the importance of considering the bias in stochastic environments that result from such a formulation. Antonoglou et al. (2022) propose an extension to MuZero in stochastic environments, but focus on the planning procedure, not the biased value function loss. Hansen et al. (2022) adapted the MuZero loss to continuous control environments but decisiondid not extend their formulation to stochastic variants. Grimm et al. (2020) and Grimm et al. (2021) consider how the set of value equivalent models relates to value functions. They are the first to highlight the close connection between the notions of value-awareness and MuZero.

**Other decision-aware algorithms:** Several other works propose decision-aware variants that do not minimize a value function difference. D'Oro et al. (2020) weigh the samples used for model learning by their impact on the policy gradient. Nikishin et al. (2021) uses implicit differentiation to obtain a loss for the model function with regard to the policy performance measure. To achieve the same goal, Eysenbach et al. (2022) and Ghugare et al. (2023) choose a variational formulation. Modhe et al. (2021) proposes to compute the advantage function resulting from different models instead of using the value function. Ayoub et al. (2020) presents an algorithm based on selecting models based on their ability to predict value function estimates and provide regret bounds with this algorithm.

**Learning with suboptimal models:** Several works have focused on the broader goal of using models with errors without addressing the loss functions of the model. Among these, several focus on correcting models using information obtained during exploration (Joseph et al., 2013; Talvitie, 2017; Modi et al., 2020; Rakhsha et al., 2022), or limiting interaction with wrong models (Buckman et al., 2018; Janner et al., 2019; Abbas et al., 2020). Several of these techniques can be applied to improve the value function of a $\lambda$ world model further. Finally, we do not focus on exploration, but Guo et al. (2022) show that a similar loss to ours can be exploited for targeted exploration.

## 7 CONCLUSIONS

In this paper, we investigated model-based reinforcement learning with decision-aware models with three main question focused on (a) implementation of the model, (b) theoretical and practical differences between major approaches in the field, and (c) scenarios in which decision-aware losses help in model-based reinforcement learning.

Empirically, we show that the design decisions established for MuZero are a strong foundation for decision-aware losses. Previous performance differences (Lovatto et al., 2020; Voelcker et al., 2022) can be overcome with latent model architectures. We furthermore establish a formal limitation on the performance of MuZero in stochastic environments, and verify this empirically. Finally, we conduct a series of experiments to establish which algorithmic choices lead to good performance empirically.

Our results highlight the importance of decision-aware model learning in continuous control and allow us to make algorithmic recommendations. When the necessary capacity for an environment model cannot be established, using a decision-aware loss will improve the robustness of the learning algorithm with regard to the model capacity. In deterministic environments with deterministic models, MuZero's value learning approach can be a good choice, as the use of real rewards seemingly provides a grounded learning signal for the value function. In stochastic environments, a model-based bootstrap is more effective, as the model-based loss does not suffer from MuZero's bias.

Overall, we find that decision aware learning is an important addition to the RL toolbox in complex environments where other modelling approaches fail. However, previously established algorithms contain previously unknown flaws or improper design decisions that have made their adoption difficult, which we overcome. In future work, evaluating other design decisions, such as probabilistic models (Ghugare et al., 2023) and alternative exploration strategies (Hansen et al., 2022; Guo et al., 2022) can provide important insights.

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

# A    PROOFS AND MATHEMATICAL CLARIFICATIONS

We provide the proofs for Section 4 in this section.

The first proposition relies on the existence of a deterministic mapping, which we prove here as a lemma. The second proposition requires a statement over the minimizers of an equation with a variance-like term, we prove a general result as a lemma.

The proof of the first lemma relies heavily on several propositions from Bertsekas & Shreve (1978), which are restated in Subsection A.3 for reader's convenience. Other topological statements are standard and can be find in textbooks such as Munkres (2018).

**Lemma 1 (Deterministic Representation Lemma)** *Let $\mathcal{X}$ be a compact, connected, metrizable space. Let $p$ be a continuous kernel from $\mathcal{X}$ to probability measures over $\mathcal{X}$. Let $\mathcal{Z}$ be a metrizable space. Consider a bijective latent mapping $\phi : \mathcal{X} \to \mathcal{Z}$ and any $V : \mathcal{Z} \to \mathbb{R}$. Assume that they are both continuous. Denote $V_{\mathcal{X}} = V \circ \phi$.*

*Then there exists a measurable function $f^* : \mathcal{Z} \to \mathcal{Z}$ such that we have $V(f^*(\phi(x))) = \mathbb{E}_p[V_{\mathcal{X}}(x')|x]$ for all $x \in \mathcal{X}$.*

**Proof:**    Since $\phi$ is a bijective continuous function over a compact space and maps to a Hausdorff space ($\mathcal{Z}$ is metrizable, which implies Hausdorff), it is a homeomorphism. The image of $\mathcal{X}$ under $\phi$, $\mathcal{Z}_{\mathcal{X}}$ is then connected and compact. Since $\mathcal{X}$ is metrizable and compact and $\phi$ is a homeomorphism, $\mathcal{Z}_{\mathcal{X}}$ is metrizable and compact. Let $\theta_{V,\mathcal{X}}(x) = \mathbb{E}_{x' \sim p(\cdot|x)}[V(x')]$. Then, $\theta_{V,\mathcal{X}}$ is continuous (Proposition 3). Define $\theta_{V,\mathcal{X}} = \theta_{V,\mathcal{Z}} \circ \phi$. Since $\phi$ is a homeomorphism, $\phi^{-1}$ is continuous. The function $\theta_{V,\mathcal{Z}}$ can be represented as a composition of continuous functions $\theta_{V,\mathcal{Z}} = \theta_{V,\mathcal{X}} \circ \phi^{-1}$ and is therefore continuous.

As $\mathcal{Z}_{\mathcal{X}}$ is compact, the continuous function $V$ takes a maximum and minimum over the set $\mathcal{Z}_{\mathcal{X}}$. This follows from the compactness of $\mathcal{Z}_{\mathcal{X}}$ and the extreme value theorem. Furthermore $V_{\min} \leq \theta_{V,\mathcal{Z}}(z) \leq V_{\max}$ for every $z \in \mathcal{Z}_{\mathcal{X}}$. By the intermediate value theorem over compact, connected spaces, and the continuity of $V$, for every value $V_{\min} \leq v \leq V_{\max}$, there exists a $z \in \mathcal{Z}_{\mathcal{X}}$ so that $V(z) = v$.

Let $h : \mathcal{Z}_{\mathcal{X}} \times \mathcal{Z}_{\mathcal{X}} \to \mathbb{R}$ be the function $h(z, z') = |\theta_{V,\mathcal{Z}}(z) - V(z')|^2$. As $h$ is a composition of continuous functions, it is itself continuous. Let $h^*(z) = \min_{z' \in \mathcal{Z}_{\mathcal{X}}} h(z, z')$. For any $z \in \mathcal{Z}_{\mathcal{X}}$, by the intermediate value argument, there exist $z'$ such that $V(z') = v$. Therefore $h^*(z)$ can be minimized perfectly for all $z \in \mathcal{Z}_{\mathcal{X}}$.

Since $\mathcal{Z}_{\mathcal{X}}$ is compact, $h$ is defined over a compact subset of $\mathcal{Z}$. By Proposition 4, there exists a measurable function $f^*(z)$ so that $\min_{z'} h(z, z') = h(z, f^*(z)) = 0$. Therefore, the function $f^*$ has the property that $V(f^*(z)) = \mathbb{E}_p[V(z')|z]$, as this minimizes the function $h$.

Now consider any $x \in \mathcal{X}$ and its corresponding $z = \phi(x)$. As $h(z, f^*(z)) = |\theta_{V,\mathcal{Z}}(z) - V(f^*(z))|^2 = 0$ for any $z \in \mathcal{Z}_{\mathcal{X}}$, $V(f^*(\phi(x))) = \theta_{v,\mathcal{Z}}(z) = \mathbb{E}_p[V_{\mathcal{X}}(x')|x]$ as desired.

$\square$

The following lemma shows that a function that minimizes a quadratic and a variance term cannot be the minimum function of the quadratic. This is used to show that the minimum of the MuZero value function learning term is not the same as applying the model-based Bellman operator.

**Lemma 2** *Let $g : \mathcal{X} \to \mathbb{R}$ be a function that is not constant almost everywhere and let $\mu$ be a non-degenerate probability distribution over $\mathcal{X}$. Let $\mathcal{F}$ be an open function space with $g \in \mathcal{F}$. Let $\mathcal{L}(f) = \mathbb{E}_{x \sim \mu}\left[(f(x) - g(x))^2\right] + \mathbb{E}_{x \sim \mu}[f(x)g(x)] - \mathbb{E}_{x \sim \mu}[f(x)]\mathbb{E}_\mu[g(x)]$. Then $g \notin \arg\min_{f \in \mathcal{F}} \mathcal{L}(f)$.*

**Proof:**    The proof follows by showing that there is a descent direction from $g$ that improves upon $\mathcal{L}$. For this, we construct the auxiliary function $\hat{g}(x) = g(x) - \epsilon g(x)$. Substituting $\hat{g}$ into $\mathcal{L}$ yields

$$\epsilon^2 \mathbb{E}_\mu[g(x)^2] + \mathbb{E}_\mu[(g(x) - \epsilon g(x))g(x)] - \mathbb{E}_\mu[(g(x) - \epsilon g(x))]\mathbb{E}_\mu[g(x)]$$
$$= \epsilon^2 \mathbb{E}_\mu[g(x)^2] + (1 - \epsilon)\mathbb{E}_\mu[g(x)^2] - (1 - \epsilon)\mathbb{E}_\mu[g(x)]^2.$$

Taking the derivative of this function wrt to $\epsilon$ yields

$$\frac{\mathrm{d}}{\mathrm{d}\epsilon}\epsilon^2\mathbb{E}_\mu\left[g(x)^2\right] + (1-\epsilon)\mathbb{E}_\mu\left[g(x)^2\right] - (1-\epsilon)\mathbb{E}_\mu\left[g(x)\right]^2$$
$$= 2\epsilon\,\mathbb{E}_\mu\left[g(x)^2\right] - \mathbb{E}_\mu\left[g(x)^2\right] + \mathbb{E}_\mu\left[g(x)\right]^2.$$

Setting $\epsilon$ to 0, obtain

$$\mathbb{E}_\mu\left[g(x)\right]^2 - \mathbb{E}_\mu\left[g(x)^2\right] = \mathrm{Var}_\mu\left[g(x)\right]$$

By the Cauchy-Schwartz inequality, the variance is only 0 for a $g(x)$ constant almost everywhere. However, this violates the assumption. Therefore for any $\epsilon > 0$ $\mathcal{L}(\hat{g}) \leq \mathcal{L}(g)$, due to the descent direction given by $-g(x)$. As we assume that the function class is open, there also exists an $\epsilon > 0$ for which $g(x) - \epsilon g(x) \in \mathcal{F}$.

$\square$

## A.1 MAIN PROPOSITIONS

**Proposition 1** *Let $M$ be an MDP with a compact, connected state space $\mathcal{X} \subseteq \mathcal{Y}$, where $\mathcal{Y}$ is a metrizable space. Let the transition kernel $p$ be continuous. Let $\mathcal{Z}$ be a metrizable space. Consider a bijective latent mapping $\phi : \mathcal{Y} \to \mathcal{Z}$ and any value function approximation $V : \mathcal{Z} \to \mathbb{R}$. Assume that they are both continuous. Denote $V_\mathcal{X} = V \circ \phi$.*

*Then there exists a measurable function $f^* : \mathcal{Z} \to \mathcal{Z}$ such that we have $V(f^*(\phi(x))) = \mathbb{E}_p\left[V_\mathcal{X}(x')|x\right]$ for all $x \in \mathcal{X}$.*

*Furthermore, the same $f^*$ is a minimizer of the population IterVAML loss:*

$$f^* \in \arg\min_{\hat{f}}\mathbb{E}\left[\hat{\mathcal{L}}_{IterVAML}(\hat{f}; V_\mathcal{X})\right].$$

**Proof:** The existence of $f^*$ follows under the stated assumptions (compact, connected and metrizable state space, metrizable latent space, continuity of all involved functions) from Lemma 1.

The rest of the proof follows standard approaches in textbooks such as Györfi et al. (2002). First, expand the equation to obtain:

$$\mathbb{E}\left[\hat{\mathcal{L}}^n_{\text{IterVAML}}(f; V_\mathcal{X}, \mathcal{D})\right] = \mathbb{E}\left[\frac{1}{N}\sum_{i=1}^N\left[V\left(f\left(\phi(x_i)\right)\right) - V_\mathcal{X}(x'_i)\right]^2\right]$$
$$= \mathbb{E}\left[\left[V\left(f\left(\phi(x)\right)\right) - \mathbb{E}\left[V_\mathcal{X}(x')|x\right] + \mathbb{E}\left[V_\mathcal{X}(x')|x\right] - V_\mathcal{X}(x')\right]^2\right].$$

After expanding the square, we obtain three terms:

$$\mathbb{E}\left[\left|V\left(f\left(\phi(x)\right)\right) - \mathbb{E}\left[V_\mathcal{X}(x')|x\right]\right|^2\right]$$
$$+2\mathbb{E}\left[\left[V\left(f\left(\phi(x)\right)\right) - \mathbb{E}\left[V_\mathcal{X}(x')|x\right]\right]\left[\mathbb{E}\left[V_\mathcal{X}(x')|x\right] - V_\mathcal{X}(x')\right]\right]$$
$$+\mathbb{E}\left[\left|\mathbb{E}\left[V_\mathcal{X}(x')|x\right] - V_\mathcal{X}(x')\right|^2\right]$$

Apply the tower property to the inner term to obtain:

$$2\mathbb{E}\left[\left[V\left(f\left(\phi(x)\right)\right) - \mathbb{E}\left[V_\mathcal{X}(x')|x\right]\right]\left[\mathbb{E}\left[V_\mathcal{X}(x')|x\right] - V_\mathcal{X}(x')\right]\right]$$
$$=2\mathbb{E}\left[\left[V\left(f\left(\phi(x)\right)\right) - \mathbb{E}\left[V_\mathcal{X}(x')|x\right]\right]\underbrace{\mathbb{E}\left[\mathbb{E}\left[V_\mathcal{X}(x')|x\right] - V_\mathcal{X}(x')|x'\right]}_{=0}\right] = 0.$$

Since the statement we are proving only applies to the minimum of the IterVAML loss, we will work with the $\arg\min$ of the loss function above. The resulting equation contains a term dependent on $f$ and one independent of $f$:

$$\arg\min_f \ \mathbb{E}\left[|V\left(f\left(\phi(x)\right)\right) - \mathbb{E}\left[V_{\mathcal{X}}(x')|x]\right|^2\right] + \mathbb{E}\left[|\mathbb{E}\left[V_{\mathcal{X}}(x')|x] - V_{\mathcal{X}}(x')\right|^2\right]$$

$$= \ \arg\min_f \ \mathbb{E}\left[|V\left(f\left(\phi(x)\right)\right) - \mathbb{E}\left[V_{\mathcal{X}}(x')|x]\right|^2\right].$$

Finally, it is easy to notice that $V\left(f^*\left(\phi(x)\right)\right) = \mathbb{E}\left[V_{\mathcal{X}}(x')|x]$ by the definition of $f^*$. Therefore $f^*$ minimizes the final loss term and, due to that, the IterVAML loss.

$\square$

**Proposition 2** *Assume a non-deterministic MDP with a fixed, but arbitrary policy $\pi$, and let $p$ be the transition kernel. Let $\mathcal{V}$ be an open set of functions, and assume that it is Bellman complete: $\forall V \in \mathcal{V} : \mathcal{T}V \in \mathcal{V}$.*

*Then for any $V' \in \mathcal{V}$ that is not a constant function, $\mathcal{T}V' \notin \arg\min_{\hat{V} \in \mathcal{V}} \mathbb{E}_{\mathcal{D}}\left[\hat{\mathcal{L}}^1_{MuZero}(p, \hat{V}; \mathcal{D}, V')\right].$*

**Notation:** For clarity of presentation denote samples from the real environment as $x^{(n)}$ for the $n$-th sample after a starting point $x^{(0)}$. This means that $x^{(n+1)}$ is drawn from $p\left(\cdot|x^{(n)}\right)$. Similarly, $\hat{x}^{(n)}$ is the $n$-th sample drawn from the model, with $\hat{x}^{(0)} = x^{(0)}$. All expectations are taken over $x_i^{(0)} \sim \mu$ where $\mu$ is the data distribution, $\hat{x}_i^{(1)} \sim \hat{p}\left(\cdot\big|x_i^{(0)}\right)$, $x_i^{(1)} \sim p\left(\cdot\big|x_i^{(0)}\right)$, and $x_i^{(2)} \sim p\left(\cdot\big|x_i^{(1)}\right)$. We use the tower property several times, all expectations are conditioned on $x_i^{(0)}{}_i$.

**Proof:** By assumption, let $\hat{p}$ in the MuZero loss be the true transition kernel $p$. Expand the MuZero loss by $\left[r\left(x_i^{(1)}\right) + \gamma V'\left(x_i^{(2)}\right)\right]$ and take its expectation:

$$\mathbb{E}\left[\hat{\mathcal{L}}^1_{\text{MuZero}}(\hat{p}, \hat{V}; \mathcal{D}, V')\right]$$

$$= \mathbb{E}\left[\frac{1}{N}\sum_{i=1}^N \left[\hat{V}\left(\hat{x}_i^{(1)}\right) - \left[r\left(x_i^{(1)}\right) + \gamma V'\left(x_i^{(2)}\right)\right]\right]^2\right]$$

$$= \mathbb{E}\left[\left[\hat{V}\left(\hat{x}_i^{(1)}\right) - (\mathcal{T}V')\left(\hat{x}_i^{(1)}\right) + (\mathcal{T}V')\left(\hat{x}_i^{(1)}\right) - \left[r\left(x_i^{(1)}\right) + \gamma V'\left(x_i^{(2)}\right)\right]\right]^2\right]$$

$$= \mathbb{E}\left[\left(\hat{V}\left(\hat{x}_i^{(1)}\right) - (\mathcal{T}V')\left(\hat{x}_i^{(1)}\right)\right)^2\right] + \tag{4}$$

$$2\,\mathbb{E}\left[\left(\hat{V}\left(\hat{x}_i^{(1)}\right) - (\mathcal{T}V')\left(\hat{x}_i^{(1)}\right)\right)\left((\mathcal{T}V')\left(\hat{x}_i^{(1)}\right) - \left[r\left(x_i^{(1)}\right) + \gamma V'\left(x_i^{(2)}\right)\right]\right)\right] + \tag{5}$$

$$\mathbb{E}\left[\left((\mathcal{T}V')\left(\hat{x}_i^{(1)}\right) - \left[r\left(x_i^{(1)}\right) + \gamma V'\left(x_i^{(2)}\right)\right]\right)^2\right] \tag{6}$$

We aim to study the minimizer of this term. The first term (Equation 4) is the regular bootstrapped Bellman residual with a target $V'$. The third term (Equation 6) is independent of $\hat{V}$, so we can drop it when analyzing the minimization problem.

The second term (Equation 5) simplifies to

$$\mathbb{E}\left[\hat{V}\left(\hat{x}_i^{(1)}\right)\left((\mathcal{T}V')\left(\hat{x}_i^{(1)}\right) - \left[r\left(x_i^{(1)}\right) + \gamma V'\left(x_i^{(2)}\right)\right]\right)\right]$$

as the remainder is independent of $\hat{V}$ again.

This remaining term however is not independent of $\hat{V}$ and not equal to $0$ either. Instead, it decomposes into a variance-like term, using the conditional independence of $\hat{x}_i^{(1)}$ and $x_i^{(1)}$ given $x_i^{(0)}$:

$$
\mathbb{E}\left[\hat{V}\left(\hat{x}_i^{(1)}\right)\left((\mathcal{T}V')\left(\hat{x}_i^{(1)}\right) - \left[r\left(x_i^{(1)}\right) + \gamma V'\left(x_i^{(2)}\right)\right]\right)\right]
$$
$$
= \mathbb{E}\left[\hat{V}\left(\hat{x}_i^{(1)}\right)(\mathcal{T}V')\left(\hat{x}_i^{(1)}\right)\right] - \mathbb{E}\left[\hat{V}\left(\hat{x}_i^{(1)}\right)\left[r\left(x_i^{(1)}\right) + \gamma V'\left(x_i^{(2)}\right)\right]\right]
$$
$$
= \mathbb{E}\left[\hat{V}\left(\hat{x}_i^{(1)}\right)(\mathcal{T}V')\left(\hat{x}_i^{(1)}\right)\right] - \mathbb{E}\left[\hat{V}\left(\hat{x}_i^{(1)}\right)\right]\mathbb{E}\left[\left[r\left(x_i^{(1)}\right) + \gamma V'\left(x_i^{(2)}\right)\right]\right].
$$

Combining this with Equation 4, we obtain

$$
\mathbb{E}\left[\hat{\mathcal{L}}_{\mathrm{MuZero}}^1(p, \hat{V}; \mathcal{D}, V')\right]
$$
$$
= \mathbb{E}\left[\left(\hat{V}\left(\hat{x}_i^{(1)}\right) - (\mathcal{T}V')\left(\hat{x}_i^{(1)}\right)\right)^2\right] +
$$
$$
\mathbb{E}\left[\hat{V}\left(\hat{x}_i^{(1)}\right)(\mathcal{T}V')\left(\hat{x}_i^{(1)}\right)\right] - \mathbb{E}\left[\hat{V}\left(\hat{x}_i^{(1)}\right)\right]\mathbb{E}\left[\left[r\left(x_i^{(1)}\right) + \gamma V'\left(x_i^{(2)}\right)\right]\right].
$$

The first summand is the Bellman residual, which is minimized by $\mathcal{T}V'$, which is in the function class by assumption. However, by Lemma 2, $\mathcal{T}V'$ does not minimize the whole loss term under the conditions (open function class, non-constant value functions, and non-degenerate transition kernel).

$\square$

**Discussion:** The proof uses Bellman completeness, which is generally a strong assumption. However, this is only used to simplify showing the contradiction at the end, removing it does not remove the problems with the loss. The proof of Lemma 2 can be adapted to the case where $f(x)$ minimizes the difference to $g(x)$, instead of using $g(x)$ as the global minimum, but some further technical assumptions about the existence of minimizers and boundary conditions are needed. The purpose here is to show that even with very favorable assumptions such as Bellman completeness, the MuZero value function learning algorithm will not converge to an expected solution.

Similarly, the condition of openness of the function class simply ensures that there exists a function "nearby" that minimizes the loss better. This is mostly to remove edge cases, such as the case where the function class exactly contains the correct solution. Such cases, while mathematically valid, are uninteresting from the perspective of learning functions with flexible function approximations.

We only show the proof for the single step version and remove action dependence to remove notational clutter, the action-dependent and multi-step versions follow naturally.

## A.2 COMPARISON OF ITERVAML AND MUZERO FOR MODEL LEARNING

If MuZero is instead used to only update the model $\hat{p}$, we obtain a similarity between MuZero and IterVAML. This result is similar to the one presented in Grimm et al. (2021), so we only present it for completeness sake, and not claim it as a fully novel contribution of our paper. While Grimm et al. (2021) show that the whole MuZero loss is an upper bound on an IterVAML-like term, we highlight the exact term the model learning component minimizes. However, we think it is still a useful derivation as it highlights some of the intuitive similarities and differences between IterVAML and MuZero and shows that they exist as algorithms on a spectrum spanned by different estimates of the target value function.

We will choose a slightly different expansion than before, using $\mathbb{E}_{x_i^{(1)}, x_i^{(2)} \sim p}\left[\left[r\left(x_i^{(1)}\right) + \gamma V'\left(x_i^{(2)}\right)\right]\right] = \mathbb{E}\left[\mathcal{T}V'\left(x_i^{(1)}\right)\right]$

$$\mathbb{E}\left[\hat{\mathcal{L}}^1_{\text{MuZero}}(\hat{p}, V; \mathcal{D}, V')\right]$$

$$= \mathbb{E}\left[\left[V\left(\hat{x}_i^{(1)}\right) - \mathbb{E}\left[\mathcal{T}V'\left(x_i^{(1)}\right)\right] + \mathbb{E}\left[\mathcal{T}V'\left(x_i^{(1)}\right)\right] - \left[r\left(x_i^{(1)}\right) + \gamma V'\left(x_i^{(2)}\right)\right]\right]^2\right]$$

$$= \mathbb{E}\left[\left(V\left(\hat{x}_i^{(1)}\right) - \mathbb{E}\left[\mathcal{T}V'\left(x_i^{(1)}\right)\right]\right)^2\right] + \tag{7}$$

$$2\mathbb{E}\left[\left(V\left(\hat{x}_i^{(1)}\right) - \mathbb{E}\left[\mathcal{T}V'\left(x_i^{(1)}\right)\right]\right)\left(\mathbb{E}\left[\mathcal{T}V'\left(x_i^{(1)}\right)\right] - \left[r\left(x_i^{(1)}\right) + \gamma V'\left(x_i^{(2)}\right)\right]\right)\right] + \tag{8}$$

$$\mathbb{E}\left[\left(\mathbb{E}\left[\mathcal{T}V'\left(x_i^{(1)}\right)\right] - \left[r\left(x_i^{(1)}\right) + \gamma V'\left(x_i^{(2)}\right)\right]\right)^2\right]. \tag{9}$$

The first summand (Equation 8) is similar to the IterVAML loss, instead of using the next state's value function, the one-step bootstrap estimate of the Bellman operator is used.

The third term (Equation 8) is independent of $\hat{p}$ and can therefore be dropped. The second term decomposes into two terms again,

$$\mathbb{E}\left[\left(V\left(\hat{x}_i^{(1)}\right) - \mathbb{E}\left[\mathcal{T}V'\left(x_i^{(1)}\right)\right]\right)\left(\mathbb{E}\left[\mathcal{T}V'\left(x_i^{(1)}\right)\right] - \left[r\left(x_i^{(1)}\right) + \gamma V'\left(x_i^{(2)}\right)\right]\right)\right]$$

$$= \mathbb{E}\left[\left(V\left(\hat{x}_i^{(1)}\right)\right)\left(\mathbb{E}\left[\mathcal{T}V'\left(x_i^{(1)}\right)\right] - \left[r\left(x_i^{(1)}\right) + \gamma V'\left(x_i^{(2)}\right)\right]\right)\right] - $$

$$\mathbb{E}\left[\mathbb{E}\left[\mathcal{T}V'\left(x_i^{(1)}\right)\right]\left(\mathbb{E}\left[\mathcal{T}V'\left(x_i^{(1)}\right)\right] - \left[r\left(x_i^{(1)}\right) + \gamma V'\left(x_i^{(2)}\right)\right]\right)\right].$$

The first summand is equal to 0, due to the conditional independence of $\hat{x}_i^{(1)}$ and $x_i^{(1)}$,

$$\mathbb{E}\left[\left(V\left(\hat{x}_i^{(1)}\right)\right)\left(\mathbb{E}\left[\mathcal{T}V'\left(x_i^{(1)}\right)\right] - \left[r\left(x_i^{(1)}\right) + \gamma V'\left(x_i^{(2)}\right)\right]\right)\right]$$

$$= \mathbb{E}\left[V\left(\hat{x}_i^{(1)}\right)\right]\underbrace{\left(\mathbb{E}\left[\mathbb{E}\left[\mathcal{T}V'\left(x_i^{(1)}\right)\right]\right] - \mathbb{E}\left[\left[r\left(x_i^{(1)}\right) + \gamma V'\left(x_i^{(2)}\right)\right]\right]\right)}_{=0} = 0.$$

The second remaining summand is independent of $\hat{p}$ and therefore irrelevant to the minimization problem.

Therefore, the MuZero model learning loss minimizes

$$\mathbb{E}\left[\hat{\mathcal{L}}^1_{\text{MuZero}}(\hat{p}, V; \mathcal{D}, V')\right] = \mathbb{E}\left[\left(V\left(\hat{x}_i^{(1)}\right) - \mathbb{E}\left[\mathcal{T}V'\left(x_i^{(1)}\right)\right]\right)^2\right].$$

In conclusion, the MuZero loss optimizes a closely related function to the IterVAML loss when used solely to update the model. There are three differences: First, the bootstrap value function estimator is used instead of the value function as the target value. Second, the current value function estimate is used for the model sample and the target network (if used) is applied for the bootstrap estimate. If the target network is equal to the value function estimate, this difference disappears. Finally, the loss does not contain the inner expectation around the model value function. This can easily be added to the loss and its omission in MuZero is unsurprising, as the loss was designed for deterministic environments and models.

The similarity between the losses suggests a potential family of decision-aware algorithms with different bias-variance characteristics, of which MuZero and IterVAML can be seen as two instances. It is also interesting to note that even without updating the value function, the MuZero loss performs an implicit minimization of the difference between the current value estimate and the Bellman operator via the model prediction. This is an avenue for further research, as it might explain some of the empirical success of the method disentangled from the value function update.

### A.3 Propositions from Bertsekas & Shreve (1978)

For convenience, we quote some results from Bertsekas & Shreve (1978). These are used in the proof of Lemma 1.

**Proposition 3 (Proposition 7.30 of Bertsekas & Shreve 1978)** *Let $\mathcal{X}$ and $\mathcal{Y}$ be separable metrizable spaces and let $q(\mathrm{d}y|x)$ be a continuous stochastic kernel on $\mathcal{Y}$ given $\mathcal{X}$. If $f \in \mathcal{C}(\mathcal{X} \times \mathcal{Y})$, the function $\lambda : \mathcal{X} \to \mathbb{R}$ defined by*

$$\lambda(x) = \int f(x, y) q(\mathrm{d}y|x)$$

*is continuous.*

**Proposition 4 (Proposition 7.33 of Bertsekas & Shreve 1978)** *Let $\mathcal{X}$ be a metrizable space, $\mathcal{Y}$ a compact metrizable space, $\mathcal{D}$ a closed subset of $\mathcal{X} \times \mathcal{Y}$, $\mathcal{D}_x = \{y|(x, y) \in \mathcal{D}\}$, and let $f : \mathcal{D} \to \mathbb{R}^*$ be lower semicontinuous. Let $f^* : proj_{\mathcal{X}}(\mathcal{D}) \to \mathbb{R}^*$ be given by*

$$f^*(x) = \min_{y \in \mathcal{D}_x} f(x, y).$$

*Then $proj_{\mathcal{X}}(\mathcal{D})$ is closed in $\mathcal{X}$, $f^*$ is lower semicontinuous, and there exists a Borel-measurable function $\phi : proj_{\mathcal{X}}(\mathcal{D}) \to \mathcal{Y}$ such that $range(\phi) \subset \mathcal{D}$ and*

$$f[x, \phi(x)] = f^*(x), \quad \forall x \in proj_{\mathcal{X}}(\mathcal{D}).$$

In our proof, we construct $f^*$ as the minimum of an IterVAML style loss and equate $\phi$ with the function we call $f$ in our proof. The change in notation is chosen to reflect the modern notation in MBRL – in the textbook, older notation is used.

### A.4 Bias due to the stabilizing loss

As highlighted in Subsection 3.1, the addition of a stabilizing loss is necessary to achieve good performance with any of the loss functions. With deterministic models, the combination $\hat{\mathcal{L}}_{\text{IterVAML}} + \hat{\mathcal{L}}^n_{\text{latent}}$ is stable, but the conditions for recovering the optimal model are not met anymore. This is due to the fact that $\arg\min_{\hat{f}} \mathbb{E}\left[\mathcal{L}\hat{}_{\text{latent}}(\hat{f}, \phi; \mathcal{D})\right] = \mathbb{E}[\phi(x')]$, but in general $\mathbb{E}[V(\phi(x'))] \neq V(\mathbb{E}[\phi(x')])$. While another stabilization technique could be found that does not have this problem, we leave this for future work.

### A.5 Multi-step IterVAML

In Subsection 2.1 the multi-step extension of IterVAML is introduced. As MVE and SVG require multi-step rollouts to be effective, it became apparent that simply forcing the one-step prediction of the value function to be correct is insufficient to obtain good performance. We therefore extended the loss into a multi-step variant.

Using linear algebra notation for simplicity, the single-step IterVAML loss enforces

$$\min \left| \langle V, P(x, a) - \hat{P}(x, a) \rangle \right|^2$$

The $n$-step variant then seeks to enforce a minimum between $n$ applications of the respective transition operators

$$\min \left| \langle V, P^n(x, a) - \hat{P}^n(x, a) \rangle \right|^2$$

The sample-based variant is a proper regression target, as $V(x^{(j)}$ is an unbiased sample from $P^n(x, a)$. It is easy to show following the same techniques as used in the proofs of propositions 1 and 2 that the sample-based version indeed minimizes the IterVAML loss in expectation.

Finally, we simply sum over intermediate $n$-step versions which results in the network being forced to learn a compromise between single-step and multi-step fidelity to the value function.

|  | Model loss | Value est. | Policy est. | Actor policy | DAML | Latent |
|---|---|---|---|---|---|---|
| $\lambda$-IterVAML | BYOL | MVE | SVG | direct | ✓ | ✓ |
| $\lambda$-MuZero | BYOL | MuZero | SVG | direct | ✓ | ✓ |
| MuZero (Schrittwieser et al., 2020) | - | MuZero | - | MCTS | ✓ | ✓ |
| Eff.-MuZero (Ye et al., 2021) | - | MuZero | - | MCTS | ✓ | ✓ |
| ALM (Ghugare et al., 2023) | ALM-ELBO | m.-free | ALM-SVG | direct | ✓ | ✓ |
| TD-MPC (Hansen et al., 2022) | BYOL | m.-free | DDPG | MPC | ✓ | ✓ |
| MBPO (Janner et al., 2019) | MLE | SAC | SAC | direct | - | - |
| Dreamer (Hafner et al., 2020) | ELBO | MVE | SVG | direct | - | ✓ |
| IterVAML (Farahmand, 2018) | - | Dyna | - | direct | ✓ | - |
| VaGraM (Voelcker et al., 2022) | weigthed MLE | SAC | SAC | direct | ✓(?) | - |

Table 1: An overview of different model-based RL algorithms and how they fit into the $\lambda$ family. The first two are the empirical algorithms tested in this work. The next section contains work that falls well into the $\lambda$ family as described in this paper. The similarities between these highlight that further algorithms are easily constructed, i.e. ALM (Ghugare et al., 2023) combined with MPC. The final section contains both popular algorithms and closely related work which inform the classification, but are not part of it since they are either not latent or not decision-aware.

## B    $\lambda$-REINFORCEMENT LEARNING ALGORITHMS

We introduce the idea of the $\lambda$ designation in Section 3. The characterization of the family is fairly broad, and it contains more algorithms than those directly discussed in this paper. An overview of related algorithms is presented in Table 1. However, we found it useful to establish that many recently proposed algorithms are closely related and contain components that can be combined freely with one another. While the community treats, i.e. MuZero, as a fixed, well-defined algorithm, it might be more useful to treat it as a certain set of implementation and design decisions under an umbrella of related algorithms. Given the findings of this paper for example, it is feasible to evaluate a MuZero alternative which simply replaces the value function learning algorithm with MVE, which should be more robust in stochastic environments.

A full benchmarking and comparison effort is out of scope for this work. However, we believe that a more integrative and holistic view over the many related algorithms in this family is useful for the community, which is why we present it here.

## C    IMPLEMENTATION DETAILS

### C.1    ENVIRONMENTS

Since all used environments are deterministic, we designed a stochastic extension to investigate the behavior of the algorithms. These are constructed by adding noise to the actions before they are passed to the simulator, but after they are recorded in the replay buffer. We used uniform and mixture of Gaussian noise of different magnitudes to mimic different levels of stochasticity in the environments. Uniform noise is sampled in the interval $[-0.2, 2]$ (uniform-small) and $[-0.4, .4]$ (uniform-large) and added to the action $a$. The Gaussian mixture distribution is constructed by first sampling a mean at $\mu = a - x$ or $a + x$ for noise levels $x$. Then the perturbation it sampled from Gaussians $\mathcal{N}(\mu, \sigma = x)$. For small and large noise we picked $x = 0.1$ and $x = 0.4$ respectively

The experiments in the main body are conducted with the uniform noise formulation, the evaluation over the Gaussian mixture noise is presented in Appendix D.

### C.2    ALGORITHM AND PSEUDOCODE

The full code is provided in the supplementary material and will be publicly released on Github after the review period is over. A pseudo-code for the IterVAML implementation of $\lambda$ is provided for reference (algorithm 1). Instead of taking the simple $n$-step rollout presented in the algorithm, we use an average over all bootstrap targets up to $n$ for the critic estimate, but used a single $n$-step bootstrap target for the actor update. We found that this slightly increases the stability of the TD update. A TD-$\lambda$ procedure as used by Hafner et al. (2020) did not lead to significant performance changes.

---

**Algorithm 1:** $\lambda$-Actor Critic (IterVAML)

---

Initialize , latent encoder $\phi_\theta$, model $\hat{f}_\theta$, policy $\pi_\omega$, value function $Q_\psi$, dataset $\mathcal{D}$;
**for** $i$ *environment steps* **do**
    Take action in env according to $\pi_\phi$; add to $\mathcal{D}$;
    Sample batch $(x_0, a_0, r_0, \ldots, x_n, r_n, a_n)$ from $\mathcal{D}_{\text{env}}$;
    # Model update
    $(z_0, \ldots, z_n) \leftarrow \phi(x_0, \ldots, x_n)$;
    $\hat{z}_0 \leftarrow z_0$;
    $\mathcal{L}_{\text{IterVAML}} \leftarrow 0$;
    $\mathcal{L}_{\text{Reward}} \leftarrow 0$;
    $\mathcal{L}_{\text{Latent}} \leftarrow 0$;
    **for** $i \leftarrow 1$; $i \leq n$; $i \leftarrow i+1$ **do**
        $\hat{z}_i, \hat{r}_{i-1} \leftarrow \hat{f}_\theta(\hat{z}_{i-1}, a_{i-1})$;
        $\mathcal{L}_{\text{IterVAML}} \leftarrow \mathcal{L}_{\text{IterVAML}} + \rho^i \left[ Q(\hat{z}_i, \pi(\hat{z}_i)) - [Q(z_i, \pi(z_i))]_{\text{sg}} \right]^2$;
        $\mathcal{L}_{\text{Reward}} \leftarrow \mathcal{L}_{\text{Reward}} + \rho^i \left[ \hat{r}_{i-1} - r_{i-1} \right]^2$;
        $\mathcal{L}_{\text{Latent}} \leftarrow \mathcal{L}_{\text{Latent}} + \rho^i \left[ \hat{z}_i - [z_i]_{\text{sg}} \right]^2$
    **end**
    $\theta \leftarrow \theta + \alpha_\theta \nabla_\theta \left( \mathcal{L}_{\text{IterVAML}} + \mathcal{L}_{\text{Reward}} + \mathcal{L}_{\text{Latent}} \right)$;
    # RL update
    **for** $i \leftarrow 1$; $i \leq n$; $i \leftarrow i+1$ **do**
        $\hat{z}_i^k \leftarrow \hat{z}_i$;
        $\hat{r} \leftarrow 0$;
        **for** $j \leftarrow 0$; $j < k$; $j \leftarrow j+1$ **do**
            $\hat{z}^k, r_j \leftarrow \hat{f}\left(\hat{z}^k, \pi\left(\hat{z}^k\right)\right)$;
            $\hat{r} \leftarrow \hat{r} + \gamma^j \hat{r}_j$
        **end**
        $J \leftarrow \hat{r} + \gamma^k Q\left(\hat{z}^k, \pi\left(\hat{z}^k\right)\right)$;
        $\mathcal{L}_q = \left[ Q\left(\hat{z}_i, \pi(\hat{z}_i) - [J]_{\text{sg}} \right]^2$;
        $\psi \leftarrow \psi + \alpha_\psi \nabla \mathcal{L}_Q$;
        $\omega \leftarrow \omega + \alpha_\omega \nabla_\omega J$
    **end**
**end**

---

Table 2: Hyperparameters. $[i, j]$ *in* $k$ refers to a linear schedule from $i$ to $j$ over $k$ env steps. The feature dimension was increased for Humanoid.

| RL HP | Value |
|---|---|
| Discount factor $\gamma$ | 0.99 |
| Polyak average factor $\tau$ | 0.005 |
| Batch size | 1024 |
| Initial steps | 5000 |
| Model rollout depth (k) | [0,4] in 25.000 |
| Actor Learning rate | 0.001 |
| Critic Learning rate | 0.001 |
| Grad clip (RL) threshold | 10 |
| hidden_dim | 512 |
| feature_dim | 50 (100 for hum.) |

| Model HP | Value |
|---|---|
| Reward loss coef. | 1.0 |
| Value loss coef. | 1.0 |
| Model depth discounting ($\rho$) | 0.99 |
| Value learning coef. (MuZero) | 0.1 |
| Model learning horizon (n) | 5 |
| Batch size | 1024 |
| Encoder Learning Rate | 0.001 |
| Model Learning Rate | 0.001 |
| Grad clip (model) threshold | 10 |
| hidden_dim | 512 |
| feature_dim | 50 (100 for hum.) |

Table 3: Model architectures. The encoder is shared between all models. The state prediction and reward prediction use the output of the model core layers.

| Encoder layers | Size |
|---|---|
| Linear | hidden_dim |
| ELU | – |
| Linear | feature_dim |

| Model core layers | Size |
|---|---|
| Linear | hidden_dim |
| ELU | – |
| Linear | hidden_dim |
| ELU | – |
| Linear | hidden_dim |

| State prediction layers | Size |
|---|---|
| Linear | hidden_dim |
| ELU | – |
| Linear | hidden_dim |
| ELU | – |
| Linear | feature_dim |

| Reward prediction layers | Size |
|---|---|
| Linear | hidden_dim |
| ELU | – |
| Linear | hidden_dim |
| ELU | – |
| Linear | 1 |

| Q function layer | Size |
|---|---|
| Linear | hidden_dim |
| LayerNorm, | – |
| Tanh | – |
| Linear | feature_dim |
| ELU | – |
| Linear | 1 |

| Actor layer | Size |
|---|---|
| Linear | hidden_dim |
| LayerNorm, | – |
| Tanh | – |
| Linear | feature_dim |
| ELU | – |
| Linear | action_dim |

## C.3 Model architecture and hyperparameter choices

Similar to previous work in MBRL (Janner et al., 2019) we slowly increased the length of model rollouts for actor and critic training following a linear schedule over the first 25.000 steps. All hyperparameters are detailed in Table 2. Where relevant, variables refer to those used in algorithm 1 for clarity.

All neural networks are implemented as two or three layer MLPs, adapted from Hansen et al. (2022). We found that minor architecture variations, such as adapting the recurrent architecture from Amos et al. (2021) did not have a large impact on the performance of the algorithm in the evaluated environments. Similarly, adding regularization such as Layer Norm or Batch Norm to the model as in Paster et al. (2021) or Ghugare et al. (2023) did not change the outcome of the experiments noticeably. We did keep the LayerNorm in the critic and actor input, as it had some stabilizing effect for the value-aware losses. The full impact of regularization in end-to-end MBRL architectures is an open question that deserves future study.

The full architecture of the model implementation is presented in Table 4 for reference. This architecture was not varied between experiments, except for the feature dimension for Humanoid experiments and the small model experiments where layer count in the model was reduced by 1 and the hidden dimension was reduced to 128.

## D Additional experiments and ablations

### D.1 Runtime estimate

Computationally efficiency was estimated on a dedicated machine with a RTX 2090 and rounded to the closest 5 minute mark. Exact runtimes can vary depending on hardware setup. Overall we find that the $\lambda$ algorithms obtains similar runtime efficiency as related approaches which also differentiate the model for policy gradients (Ghugare et al., 2023) and is slightly slower than just using the model

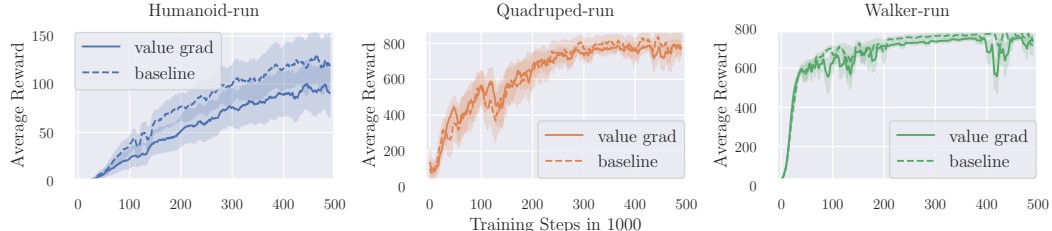

Figure 5: Comparison of using the TD gradient for model learning as well in $\lambda$-IterVAML.

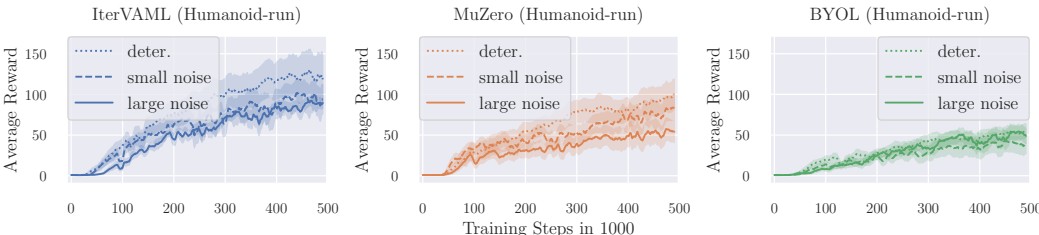

Figure 6: Impact of mixture noise on the learning performance.

for MPC (Hansen et al., 2022). Using the IterVAML loss requires another network pass through the value function compared to MuZero, therefore it is slightly slower in our current setup. However, these differences can most likely be improved with a more efficient implementation with more clever reuse of intermediate computation.

All model-based algorithms have a slowdown of roughly the model depth over SAC ($\times 4 - 6$). This is expected, since the model is rolled out sequentially and the complexity of the computational graph grows linearly with the model rollout depth. Since SAC requires large networks and high UTD to obtain similar performance to the model-based approaches, MBRL is still runtime competitive to model-free RL (compare Ghugare et al. (2023)).

## D.2 GRADIENT PROPAGATION IN ITERVAML

We tested propagating the gradient from the IterVAML value function learning into the model and encoder as well. Results are presented in Figure 5. It is clear that propagating the TD error gradient into the model does not improve the quality of the model. Especially in Humanoid we observe a drop in average performance, although it is not statistically significant under our 99.7 certainty interval.

## D.3 FURTHER EXPERIMENTS IN NOISY ENVIRONMENTS

Here we present the impact of another noise distribution on Humanoid. We tested with the mixture of Gaussian noise as outlined above. The results are presented in Figure 6. It is clear that the same pattern holds independent of the exact form of the noise distribution.

Table 4: Runtime estimates for all used algorithms. Runtime estimates were obtained on a dedicated machine with a RTX 2090 and rounded. Exact runtimes can vary depending on hardware setup.

|  | $\lambda$-IterVAML | $\lambda$-MuZero | BYOL ablation | SAC |
|---|---|---|---|---|
| Runtime (500k) | 7:45h | 5:40h | 5:20h | 1:30h |

### D.3.1 TABULAR BENCHMARK

As discussed, no widely used large-scale benchmark for stochastic MDPs exists. Therefore, we created a small scale benchmark based on the Garnet MDP framework (Archibald et al., 1995) as follows:

We constructed a deterministic Markov Reward Process with a finite state space consisting of states ($|\mathcal{S}| = n$). Its transition matrix is a permutation of the identity matrix. The reward was randomly sampled from a standard normal distribution and an additional reward of 10 was assigned to the first state to further increase the variance of the reward over all states, as this is the critical scenario for MuZero's bias.

To introduce stochasticity, we allowed for $m - 1$ additional successor states for each state $\mathcal{S}_x \subset \mathcal{S}$. We also introduced an additional parameter $\rho$ to interpolate between the deterministic Markov Reward Process and the stochastic one. The first successor state taken from the identity matrix was assigned the transition probability $\rho$ and all additional successor states were assigned probability $(1 - \rho)/(m - 1)$. This allowed us to control the stochasticity of the problem simply by varying $\rho$.

Since finite-state, deterministic models are non differentiable, we instead modeled the impact of the model class bias by constraining the capacity of the model. The model class is constructed from low rank factorized matrices, represented by a random representation matrix $\Phi$ and a learnable weight matrix $\Psi$ of dimensions $(n, k)$, so that $\hat{P} = \sigma(\Phi^T \Psi)$. The probability constraint are ensured via the row-wise softmax operation $\sigma$. By reducing k, the capacity of the model is constrained so that a correct model cannot be represented.

We did not constrain the representation of the value function and simply learned tabular value functions using fitted value iteration with a model-based target estimate and with the MuZero value function learning scheme. We used the original formulation of each loss as accurately as possible, with some minor adaptation for MuZero. This was necessary as the end-to-end gradient through the value function is impossible to compute with discrete samples. Instead, we used the difference between the expected value over the next state and the bootstrapped target value, similarly to the IterVAML formulation presented in **??**.

We conducted our experiment over 16 randomly generated garnets, rewards and representation matrices with $n = 50$ and $m = 10$. The data was sampled from a uniform distribution and next states according to the transition matrix. The reward function was not learned to simplify the setup. We sampled sufficient data points $(100, 000)$ to ensure that an unconstrained MLE model can be learned to high accuracy. The model and value function were learned with full batch gradient descent using Adam. All experiments were simple value estimation for a fixed reward process with constrained model classes; we did not address policy improvement. We varied the constraint k between 10 and 1 to assess the impact of reduced model capacity on the algorithms, with varying $\rho \in [0.5, 0.75, 1.0]$. We also added experiments for $k = 50$ to assess an unconstrained model, although in most cases, performance was already close to optimal with $k = 10$.

The results of these experiments are presented in Table 5.

We found that for all values of stochasticity, the IterVAML model performs best for value function learning. Curiously, in all environments it is matched closely or outperformed slightly by the MLE solution with extremely low values of $k$, which suggests that the model does not have sufficient flexibility to model any value estimate correctly. In these cases, the MLE can provide a more stable learning target, even though none of the evaluated algorithms will be useful under extreme constraints.

Curiously, even in the deterministic case, MuZero does not perform well unless the model capacity is large enough. We investigated this further and found that due to our constraint setup, the model is unable to fully capture deterministic dynamics in the constrained case and will instead remain stochastic. Therefore the bias of the MuZero loss still impacts the system, as the variance over next state values induced by the model is high. We also noticed that the gradient descent based optimization of value function and model in MuZero has a tendency to converge to suboptimal local minima, which might explain some of the extreme outliers.

When replacing the value learning scheme with model-based TD learning, we recover almost identical performance between IterVAML and MuZero, which further highlights the results presented in this paper and by Grimm et al. (2021) that both losses optimize a similar target.

| $\rho = 0.5$ | IterVAML | MuZero | MLE |
|---|---|---|---|
| 1 | 0.9406 +/- 0.0955 | 3.6476 +/- 0.11205 | **0.8417 +/- 0.0532** |
| 2 | 0.9149 +/- 0.02915 | 3.6486 +/- 0.115875 | **0.8671 +/- 0.035975** |
| 3 | 0.8949 +/- 0.07915 | 3.5668 +/- 0.1151 | **0.7687 +/- 0.031925** |
| 4 | **0.5728 +/- 0.0723** | 3.5281 +/- 0.123425 | 0.6941 +/- 0.05015 |
| 5 | **0.29 +/- 0.023825** | 3.3755 +/- 0.136025 | 0.5256 +/- 0.0268 |
| 6 | **0.2325 +/- 0.01965** | 3.1761 +/- 0.116925 | 0.5618 +/- 0.046075 |
| 7 | **0.174 +/- 0.02195** | 3.1889 +/- 0.160375 | 0.4792 +/- 0.047225 |
| 8 | **0.1391 +/- 0.0179** | 2.9965 +/- 0.12555 | 0.3936 +/- 0.038925 |
| 9 | **0.1032 +/- 0.0139** | 2.8649 +/- 0.11805 | 0.3623 +/- 0.029225 |
| 10 | **0.0577 +/- 0.005925** | 2.8381 +/- 0.151475 | 0.273 +/- 0.022625 |
| 50 | **0.0017 +/- 0.000225** | 2.5494 +/- 0.115075 | **0.0017 +/- 0.0002** |

| $\rho = 0.75$ | IterVAML | MuZero | MLE |
|---|---|---|---|
| 1 | 2.8279 +/- 0.283675 | 5.1395 +/- 0.257475 | **2.6313 +/- 0.210275** |
| 2 | **2.7452 +/- 0.1289** | 5.1149 +/- 0.27055 | **2.76 +/- 0.1613** |
| 3 | **1.8714 +/- 0.195725** | 4.8157 +/- 0.27655 | 2.4053 +/- 0.153125 |
| 4 | **0.7264 +/- 0.043275** | 4.5975 +/- 0.2393 | 1.6601 +/- 0.1318 |
| 5 | **0.3794 +/- 0.026425** | 4.3195 +/- 0.2547 | 1.1971 +/- 0.042725 |
| 6 | **0.2066 +/- 0.016675** | 3.4099 +/- 0.24005 | 0.9698 +/- 0.0657 |
| 7 | **0.1213 +/- 0.013075** | 3.1378 +/- 0.231275 | 0.5805 +/- 0.05625 |
| 8 | **0.1086 +/- 0.006675** | 3.4061 +/- 0.167275 | 0.6449 +/- 0.080075 |
| 9 | **0.0799 +/- 0.009** | 3.0232 +/- 0.23425 | 0.4361 +/- 0.042275 |
| 10 | **0.0456 +/- 0.0036** | 3.2613 +/- 0.29505 | 0.2999 +/- 0.028375 |
| 50 | **0.0027 +/- 0.00025** | 2.5132 +/- 0.194225 | **0.0026 +/- 0.000275** |

| $\rho = 1.$ | IterVAML | MuZero | MLE |
|---|---|---|---|
| 1 | 14.0457 +/- 1.50455 | 16.5522 +/- 1.79805 | 14.2642 +/- 1.527775 |
| 2 | 13.0414 +/- 1.100675 | 16.9774 +/- 1.636875 | 13.7411 +/- 1.30145 |
| 3 | **6.1846 +/- 0.688375** | 11.9329 +/- 1.19215 | 9.5593 +/- 1.193025 |
| 4 | **1.1092 +/- 0.219475** | 3.335 +/- 0.7467 | 2.248 +/- 0.355225 |
| 5 | **0.0977 +/- 0.026175** | 0.3616 +/- 0.107525 | 0.2503 +/- 0.08265 |
| 6 | **0.01 +/- 0.0031** | 36.276 +/- 23.983825 | **0.0042 +/- 0.001** |
| 7 | 0.0065 +/- 0.0028 | 0.0134 +/- 0.001125 | 0.0052 +/- 0.001325 |
| 8 | 0.0083 +/- 0.0021 | 0.0128 +/- 0.001275 | 0.0045 +/- 0.001475 |
| 9 | 0.0032 +/- 0.0004 | 0.0117 +/- 0.001075 | 0.0021 +/- 0.000425 |
| 10 | 0.0054 +/- 0.001675 | 0.0092 +/- 0.00105 | 0.0035 +/- 0.000725 |
| 50 | 0.0023 +/- 0.00035 | 0.0048 +/- 0.0013 | 0.0044 +/- 0.000775 |

Table 5: Garnet experiment results. Each cell depicts mean and standard deviation of value estimation over 16 randomly generated garnet MDPs for each algorithm. Minimum value highlighted (when ambiguous due to confidence interval, multiple values are highlighted). For the case of $\rho = 1$, all losses achieve strong performance with $k \geq 7$, therefore no highlighting was necessary.

Overall, these experiments provide further evidence that the MuZero value learning scheme is flawed in stochastic environments, and with stochastic models.

### D.4 REMOVING LATENT FORMULATION AND STABILIZING LOSS

Finally, we present the results of removing both the latent and the auxiliary stabilization in all $\lambda$ algorithms in Figure 7. IterVAML diverges catastrophically, with value function error causing numerical issues, which is why the resulting graph is cut off: no run in 8 seeds was able to progress beyond 100.000 steps. This is similar to results described by Voelcker et al. (2022). MuZero on the other hand profits from not using a stabilizing BYOL loss when operating in observation space. This substantiates claims made by Tang et al. (2022), which conjecture that latent self-prediction losses such as BYOL can capture good representation of MDPs for reinforcement learning but reconstruction accuracy might not be a useful goal.

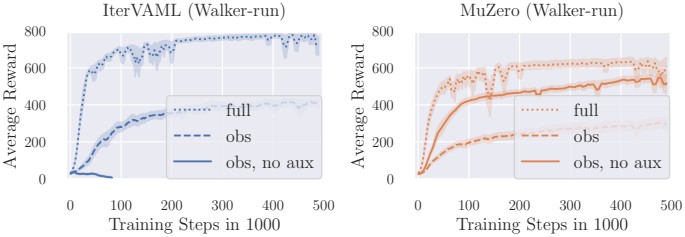

Figure 7: Comparison for removing both the latent formulation and the stabilization loss

# λ:   Effective   decision-aware   reinforcement learning with latent models

**Anonymous authors**

## Abstract

The idea of decision-aware model learning, that models should be accurate where it matters for decision-making, has gained prominence in model-based reinforcement learning. While promising theoretical results have been established, the empirical performance of algorithms leveraging a decision-aware loss has been lacking, especially in continuous control problems. In this paper, we present a study on the necessary components for decision-aware reinforcement learning models and we showcase design choices that enable well-performing algorithms. To this end, we provide a theoretical and empirical investigation into algorithmic ideas in the field. We highlight that empirical design decisions established in the MuZero line of works, most importantly the use of a latent model, are vital to achieving good performance for related algorithms. Furthermore, we show that the MuZero loss function is biased in stochastic environments and establish that this bias has practical consequences. Building on these findings, we present an overview of which decision-aware loss functions are best used in what empirical scenarios, providing actionable insights to practitioners in the field.

## 1   Introduction

In model-based reinforcement learning, an agent collects information in an environment and uses it to learn a model of the world to accelerate and improve value estimation or the agent's policy (Sutton, 1990; Deisenroth & Rasmussen, 2011; Hafner et al., 2020; Schrittwieser et al., 2020). However, as environment complexity increases, learning a model becomes more and more challenging. These model errors can impact the learned policy (Schneider, 1997; Kearns & Singh, 2002; Talvitie, 2017; Lambert et al., 2020) leading to worse performance. In many realistic scenarios, agents are faced with a "big world" where learning to accurately simulate the environment is infeasible. When faced with complex environments, deciding what aspects of the environment to model is crucial. Otherwise, capacity would be spent on irrelevant aspects, e.g., modelling clouds in a vehicle's video feed.

Recently, the paradigm of *decision-aware model learning* (DAML) (Farahmand et al., 2017) or *value equivalence* (Grimm et al., 2020; 2021) has been proposed. The core idea is that a model should make predictions that result in correct value estimation. The two most prominent approaches in the space of decision-aware model learning are the IterVAML (Farahmand, 2018) and MuZero (Schrittwieser et al., 2020) algorithms. While IterVAML is a theoretically grounded algorithm, difficulties in adapting the loss for empirical implementations have been highlighted in the literature (Lovatto et al., 2020; Voelcker et al., 2022). MuZero on the other hand has been shown to perform well in discrete control tasks (Schrittwieser et al., 2020; Ye et al., 2021) but has received little theoretical investigation. Thus, understanding the role of different value-aware losses and determining the factors for achieving strong performance is an open research problem.

**Research question:**   In this work, we ask three question: (a) What design decisions explain the performance difference between IterVAML and Muzero? (b) What are theoretical or practical differences between the two? (c) In what scenarios do decision-aware losses help in model-based reinforcement learning? Experimental evaluations focus on the state-based DMC environments (Tunyasuvunakool et al., 2020), as the focus of our paper is not to scale up to image-based observations.

**Contributions:**   The main differences between previous works trying to build on the IterVAML and MuZero algorithms are neural network architecture design and the value function learning scheme.

MuZero is built on value prediction networks (Oh et al., 2017), which explicitly incorporate a latent world model, while IterVAML is designed for arbitrary model choices. We show that the use of latent models can explain many previously reported performance differences between MuZero and IterVAML (Voelcker et al., 2022; Lovatto et al., 2020).

As a theoretical contribution, we first show that IterVAML and MuZero-based models can achieve low error in stochastic environments, even when using deterministic world models, a common empirical design decision (Oh et al., 2017; Schrittwieser et al., 2020; Hansen et al., 2022). To the best of our knowledge, we are the first to prove this conjecture and highlight that it is a unique feature of the IterVAML and MuZero losses. The MuZero value function learning scheme however results in a bias in stochastic environments. We show that this bias leads to a quantifiable difference in performance. Finally, we show that the model learning component of MuZero is similar to VAML, which was previously also noted by Grimm et al. (2021).

In summary, our contributions are a) showing that IterVAML is a stable loss when a latent model is used, b) proving and verifying that the MuZero value loss is biased in stochastic environments, and c) showing how these algorithms provide benefit over decision-agnostic baseline losses.

## 2 BACKGROUND

**Reinforcement Learning:** We consider a standard Markov decision process (MDP) (Puterman, 1994; Sutton & Barto, 2018) $(\mathcal{X}, \mathcal{A}, p, r, \gamma)$, with state space $\mathcal{X}$, action space $\mathcal{A}$, transition kernel $p(x'|x, a)$, reward function $r : \mathcal{X} \times \mathcal{A} \to \mathbb{R}$, and discount factor $\gamma \in [0, 1)$. The goal of an agent is to optimize the obtained average discounted infinite horizon reward under its policy: $\max_\pi \mathbb{E}_{\pi, p} \left[ \sum_{t=0}^\infty \gamma^t r(x_t, a_t) \right]$.

Given a policy $\pi(a|x)$, the value function is defined as the expected return conditioned on a starting state $x$: $V^\pi(x) = \mathbb{E}_\pi \left[ \sum_{t \geq 0} \gamma^t r_t | x_0 = x \right]$, where $r_t = r(x_t, a_t)$ is the reward at time $t$. The value function is the unique stationary point of the *Bellman operator* $\mathcal{T}_p(V)(x) = \mathbb{E}_{\pi(a|s)} \left[ r(x, a) + \gamma \mathbb{E}_{p(x'|x,a)} \left[ V(x') \right] \right]$. The action-value function is defined similarly as $Q^\pi(x, a) = \mathbb{E}_\pi \left[ \sum_{t \geq 0} \gamma^t r_t | x_0 = x, a_0 = a \right]$. In this work, we do not assume access to the reward functions and denote learned approximation as $\hat{r}(x, a)$.

**Model-based RL:** An *environment model* is a function that approximates the behavior of the transition kernel of the ground-truth MDP.[1] Learned models are used to augment the reinforcement learning process in several ways (Sutton, 1990; Janner et al., 2019; Hafner et al., 2020), we focus on the MVE (Buckman et al., 2018) and SVG (Amos et al., 2021) algorithms presented in details in Subsection 2.2. These were chosen as they fit the decision-aware model learning framework and have been used successfully in strong algorithms such as Dreamer (Hafner et al., 2021).

We will use $\hat{p}$ to refer to a learned probabilistic models, and use $\hat{f}$ for deterministic models. When a model is used to predict the next ground-truth observation $x'$ from $x, a$ (such as the model used in MBPO (Janner et al., 2019)) we call it an *observation-space models*. An alternative are *latent models* of the form $\hat{p}(z'|z, a)$, where $z \in \mathcal{Z}$ is a representation of a state $x \in \mathcal{X}$ given by $\phi : \mathcal{X} \to \mathcal{Z}$. In summary, observation-space models seek to directly predict next states in the representation given by the environment, while latent space models can be used to reconstruct learned features of the next state. We present further details on latent model learning in Subsection 3.1.

The notation $\hat{x}_i^{(n)}$ refers to an $n$-step rollout obtained from a model by starting in state $x_i$ and sampling the model for $n$-steps. We will use $\hat{p}^n \left( x^{(n)}|x \right)$ and $\hat{f}^n(x)$ to refer to the $n$-th step model prediction.

### 2.1 DECISION-AWARE MODEL LOSSES

The losses of the decision-aware learning framework share the goal of finding models that provide good value function estimates. Instead of simply learning a model using maximum likelihood

---

[1] In this paper, we will generally use the term *model* to refer to an environment model, not to a neural network, to keep consistent with the reinforcement learning nomenclature.

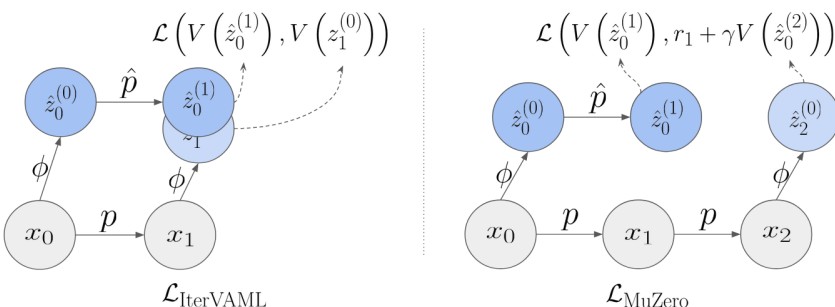

Figure 1: Sketch of the different value-aware losses with latent models. IterVAML computes the value function difference between the latent prediction and the next state encoding, while MuZero computes a single-step bootstrap estimate.

estimation, the losses are based on differences in value prediction. We present the loss functions below to highlight the close relationship as well as crucial differences between the two. We assume a standard dataset of state, action, and reward sequences $\mathcal{D} = \{x_{i_1}, a_{i_1}, r_{i_1}, \ldots, x_{i_n}, a_{i_n}, r_{i_n}\}_{i=1}^N$ with $x_{i_1}$ sampled from $\mu$ and subsequent states sampled according to the transition function. $N$ denotes the number of samples, and $n$ the sequence length. As all considered algorithms are off-policy, the actions can come from different past policies. Note that the index 1 here denotes the start of a (sub-)sequence, not the start of an episode.

**IterVAML (Farahmand, 2018):** To compute the IterVAML loss, value function estimates are computed over the model prediction and a real sample drawn from the training distribution. The loss minimizes the squared difference between them. While the original paper only considers a single-step variant, we present a multi-step version of the loss (for a discussion on the differences, see Subsection A.5). The resulting IterVAML loss is

$$\hat{\mathcal{L}}_{\text{IterVAML}}^n(\hat{p}; \hat{V}, \mathcal{D}) = \frac{1}{N \cdot n} \sum_{i=1}^N \sum_{j=1}^n \left| \underbrace{\mathbb{E}_{\hat{x}^{(j)} \sim \hat{p}^j(\cdot | x_{i_1}, a_{i_1})} \left[ \hat{V} \left( \hat{x}^{(j)} \right) \right]}_{\text{Model value estimate}} - \underbrace{\hat{V}\left( x_{i_j} \right)}_{\text{Ground truth value estimate}} \right|^2. \quad (1)$$

**MuZero (Schrittwieser et al., 2020):** MuZero proposes a similar loss. However, the next state value function over the real sample is estimated via an approximate application of the Bellman operator. This core difference between both losses is illustrated in Figure 1. Similar to Mnih et al. (2013), MuZero uses a target network for the bootstrap estimate, which we denote as $V_{\text{target}}$. The MuZero loss is introduced with using deterministic models, which is why we present it here as such. The extension to stochastic models is non-trivial, as we show in Subsection 4.2 and discuss further in Subsection A.2. The loss is written as

$$\hat{\mathcal{L}}_{\text{MuZero}}^n\left( \hat{f}, \hat{V}; \mathcal{D}, V_{\text{target}} \right) = \frac{1}{N \cdot n} \sum_{i=1}^N \sum_{j=i}^n \left| \underbrace{\hat{V}\left( \hat{f}^j(x_{i_1}, a_{i_1}) \right)}_{\text{Model value estimate}} - \underbrace{\left[ r_{i_j} + \gamma V_{\text{target}}\left( x_{i_{j+1}} \right) \right]}_{\text{Ground truth bootstrap value estimate}} \right|^2. \quad (2)$$

Note that the real environment reward and next states are used in the $j$-th step. The MuZero loss is also used for updating both the value function and the model, while IterVAML is only designed to update the model and uses a model-based bootstrap target to update the value function.

**Stabilizing loss:** Decision-aware losses can be insufficient for stable learning (Ye et al., 2021), especially in continuous state-action spaces (Hansen et al., 2022). Their performance is improved greatly by using a *stabilizing or auxiliary loss*. In practice, latent self-prediction, based on the *Bootstrap your own latent* (BYOL) loss (Grill et al., 2020), has shown to be a strong and simple option for this task. Several different versions of this loss have been proposed for RL (Gelada et al., 2019; Schwarzer et al., 2021; Tang et al., 2022), we consider a simple $n$-step variant:

$$\hat{\mathcal{L}}_{\text{latent}}^n\left( \hat{f}, \phi; \mathcal{D} \right) = \frac{1}{N \cdot n} \sum_{i=1}^N \sum_{j=1}^n \left[ \hat{f}^{(j)}(\phi(x_{i_1}), a_{i_1}) - \text{stop-grad}\left[ \phi(x_{i_j}) \right] \right]^2. \quad (3)$$

Theoretical analysis of this loss by Gelada et al. (2019) and Tang et al. (2022) shows that the learned representations provide a good basis for value function learning.

## 2.2 ACTOR-CRITIC LEARNING

**Value leaning:** Both IterVAML and MuZero can use the model to approximate the value function target. In its original formulation, MuZero is used with a Monte Carlo Tree Search procedure to expand the model which is not directly applicable in continuous state spaces. A simple alternative for obtaining a value function estimate from the model in continuous state spaces is known as model value expansion (MVE) (Feinberg et al., 2018). In MVE, short horizon roll-outs of model predictions $[\hat{x}^{(0)}, \ldots, \hat{x}^{(n)}]$ are computed using the current policy estimate, with a real sample from the environment as a starting point $x^{(0)}$. Using these, $n$-step value function targets are computed $Q_{\text{target}}^j\left(\hat{x}^{(j)}, \pi\left(\hat{x}^{(j)}\right)\right) = \sum_{i=j}^{n-1} \gamma^{i-j} \hat{r}\left(\hat{x}^{(i)}, \pi\left(\hat{x}^{(i)}\right)\right) + \gamma^{n-j} \bar{Q}\left(\hat{x}^{(n)}, \pi\left(\hat{x}^{(n)}\right)\right)$. These targets represent a bootstrap estimate of the value function starting from the $j$-th step of the model rollout. For our IterVAML experiments, we used these targets together with TD3 (Fujimoto et al., 2018) to learn the value function, for MuZero, we used these targets for $V_{\text{target}}$ in Equation 2.

**Policy learning:** In continuous control tasks, estimating a policy is a crucial step for effective learning algorithms. To improve policy gradient estimation with a model, stochastic value gradients (SVG) (Heess et al., 2015; Amos et al., 2021) can be used. These are obtained by differentiating the full $n$-step MVE target with regard to the policy. As most common architectures are fully differentiable, this gradient can be computed using automatic differentiation, which greatly reduces the variance of the policy update compared to traditional policy gradients. We evaluate the usefulness of model-based policy gradients in Section 5, with deterministic policy gradients (Silver et al., 2014; Lillicrap et al., 2016) as a model-free baseline.

## 3 LATENT DECISION AWARE MODELS

Previous work in decision-aware MBRL (Farahmand et al., 2017; Farahmand, 2018; Schrittwieser et al., 2020; Grimm et al., 2021) uses differing model architectures, such as an ensemble of stochastic next state prediction networks in the case of Voelcker et al. (2022) or a deterministic latent model such as in Schrittwieser et al. (2020); Hansen et al. (2022). In this section, we highlight the importance of using latent networks for decision-aware losses. We call these variants of the investigate algorithms *LAtent Model-Based Decision-Aware (LAMBDA)* algorithms, or $\lambda$-IterVAML and $\lambda$-MuZero for short, to stress the importance of combining latent models with decision-aware algorithms. We focus on IterVAML and MuZero as the most prominent decision-aware algorithms.

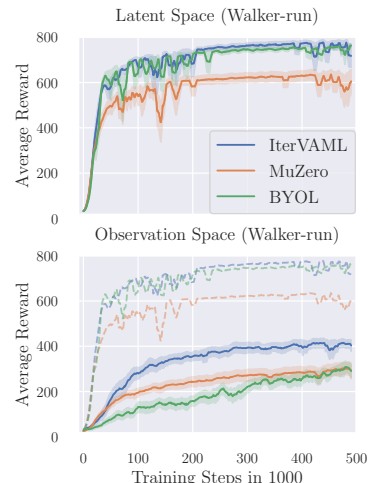

### 3.1 THE IMPORTANCE OF LATENT SPACES
FOR APPLIED DECISION-AWARE MODEL LEARNING

One of the major differences between MuZero and IterVAML is that the former uses a latent model design, while the Iter-VAML algorithm is presented without a specific architecture. When used with a state-space prediction model, IterVAML has been shown to diverge, leading to several works claiming that it is an impractical algorithm (Lovatto et al., 2020; Voelcker et al., 2022). However, there is no a priori reason why a latent model should not be used with IterVAML, or why MuZero has to be used with one. In prior work, sharp

Figure 2: Comparison on the use of an explicit latent space with different loss functions. All losses improve in performance from the addition of an explicit latent space transformation. Dashed lines represent the mean of the latent space result.

value function gradients have been hypothesized to be a major concern for IterVAML (Voelcker et al., 2022). The introduction of a learned representation space in the model through a latent embedding can help to alleviate this issue.

The full loss for each experiment is $\mathcal{L}^n_{\text{MuZero/IterVAML}} + \mathcal{L}^n_{\text{latent}} + \mathcal{L}^n_{\text{reward}}$, where $\mathcal{L}_{\text{reward}}$ is an MSE term between the predicted and ground truth rewards. As a baseline, we drop the decision-aware loss and simply consider BYOL alone. For all of our experiments we use eight random seeds and the shaded areas mark three $\sigma$ of the standard error of the mean over random seeds, which represents a $99.7\%$ certainty interval under a Gaussian assumption. Full pseudocode is found in Subsection C.2.

As is evident from Figure 2 the latent space is highly beneficial to achieve good performance. In higher dimensional experiments and without stabilizing losses, we find that IterVAML diverges completely (see Appendix D) This highlights that negative results on the efficacy of IterVAML are likely due to suboptimal choices in the model implementation, not due to limitations of the algorithm.

## 4 ANALYSIS OF DECISION-AWARE LOSSES IN STOCHASTIC ENVIRONMENTS

After accounting for model implementation, both MuZero and IterVAML behave similarly in experiments. However, it is an open question if this holds in general. Almost all commonly used benchmark environments in RL are deterministic, yet stochastic environments are an important class of problems both for theoretical research and practical applications. Therefore, we first present a theoretical investigation into the behavior of both the MuZero and IterVAML losses in stochastic environments, and then evaluate our findings empirically. Proofs are found in Subsection A.1.

In Proposition 1 we establish that $\lambda$-IterVAML leads to an unbiased solution in the infinite sample limit, even when restricting the model class to deterministic functions under measure-theoretic conditions on the function class. This is a unique advantage of value-aware models. Algorithms such as Dreamer (Hafner et al., 2020) or MBPO (Janner et al., 2019) require stochastic models, because they model the complete next-state distribution, instead of the expectation over the next state.

We then show in Proposition 2 that MuZero's joint model- and value function learning algorithm leads to a biased solution, even when choosing a probabilistic model class that contains the ground-truth environment. This highlights that while the losses are similar in deterministic environments, the same does not hold in the stochastic case. Finally, we verify the theoretical results empirically by presenting stochastic extensions of environments from the DMC suite (Tunyasuvunakool et al., 2020).

### 4.1 RESTRICTING ITERVAML TO DETERMINISTIC MODELS

In most cases, deterministic function approximations cannot capture the transition distribution on stochastic environments. However, it is an open question whether a deterministic model is sufficient for learning a *value-equivalent* model, as conjectured by (Oh et al., 2017). We answer this now in the affirmative. Showing the existence of such a model relies on the continuity of the transition kernel and involved functions $\phi$ and $V$.[2]

**Proposition 1** *Let $\mathcal{X}$ be a compact, connected, metrizable space. Let $p$ be a continuous kernel from $\mathcal{X}$ to probability measures over $\mathcal{X}$. Let $\mathcal{Z}$ be a metrizable space. Consider a bijective latent mapping $\phi : \mathcal{X} \to \mathcal{Z}$ and any $V : \mathcal{Z} \to \mathbb{R}$. Assume that they are both continuous. Denote $V_{\mathcal{X}} = V \circ \phi$.*

*Then there exists a measurable function $f^* : \mathcal{Z} \to \mathcal{Z}$ such that we have $V(f^*(\phi(x))) = \mathbb{E}_p[V_{\mathcal{X}}(x')|x]$ for all $x \in \mathcal{X}$.*

*Furthermore, the same $f^*$ is a minimizer of the expected IterVAML loss:*

$$f^* \in \arg\min_{\hat{f}} \mathbb{E}\left[\hat{\mathcal{L}}_{\text{IterVAML}}(\hat{f}; V_{\mathcal{X}})\right].$$

We can conclude that given a sufficiently flexible function class $\mathcal{F}$, IterVAML can recover an optimal deterministic model for value function prediction. Note that our conditions solely ensure the *existence* of a measurable function; the learning problem might still be very challenging. Nonetheless, even without the assumption that the perfect model $f$ is learnable, the IterVAML loss finds the function that is closest to it in the mean squared error sense, as shown by Farahmand (2018).

---

[2]To reduce notational complexity, we ignore the action dependence in all following propositions; all results hold without loss of generality for the action-conditioned case as well.

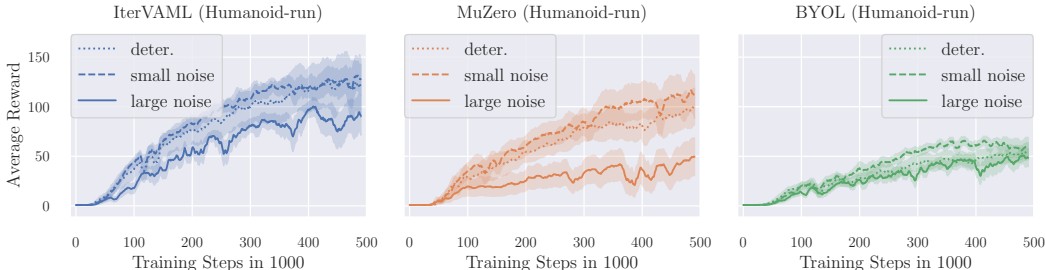

Figure 4: Comparison of the impact of noise across all algorithms. While both IterVAML and MuZero are impacted by the level of noise, we observe a larger drop in MuZero, which does not perform above the BYOL baseline at the highest noise level.

## 4.2 SAMPLE BIAS OF MUZERO' VALUE FUNCTION LEARNING ALGORITHM

MuZero is a sound algorithm for deterministic environments, however, the same is not true for stochastic ones. While changed model architecture for MuZero have been proposed for stochastic cases (Antonoglou et al., 2022), the value function learning component contains its own bias. To highlight that the problem is neither due to the $n$-step formulation nor due to suboptimal architecture or value function class, we show that the value function estimate does not converge to the model Bellman target using the MuZero loss. Intuitively, the problem results from the fact that two samples drawn from the model and from the environment do not coincide, even when the true model and the learned model are equal (see Figure 3). This bias is similar to the double sampling issue in Bellman residual minimization, but is distinct as the introduction of the stop-gradient in MuZero's loss function does not address the problem.

**Proposition 2** *Assume a non-deterministic MDP with a fixed, but arbitrary policy $\pi$, and let $p$ be the transition kernel. Let $\mathcal{V}$ be an open set of functions, and assume that it is Bellman complete: $\forall V \in \mathcal{V} : \mathcal{T}V \in \mathcal{V}$.*

*Then for any $V' \in \mathcal{V}$ that is not a constant function, $\mathcal{T}V' \notin \arg\min_{\hat{V} \in \mathcal{V}} \mathbb{E}_{\mathcal{D}} \left[ \hat{\mathcal{L}}^1_{MuZero}(p, \hat{V}; \mathcal{D}, V') \right]$.*

The bias indicates that MuZero will not recover the correct value function in environments with stochastic transitions, even when the correct model is used and the function class is Bellman complete. On the other hand, model-based MVE such as used in $\lambda$-IterVAML can recover the model's value function in stochastic environments.

The bias is dependent on the variance of the value function with regard to the transition distributions. This means that in some stochastic environments the MuZero loss might still perform well. But as the variance of the value function increases, the bias to impact the solution.

If the MuZero loss is solely used for model learning and the value function is learned fully model-free or model-based, the IterVAML and MuZero algorithms show strong similarities (compare Subsection A.2). The main difference is that MuZero uses a bootstrap estimate for the value function, while IterVAML uses the value function estimate directly. However, when jointly training value function and model in a stochastic environment, neither the model nor the value function converge to the correct solution, due to the tied updates.

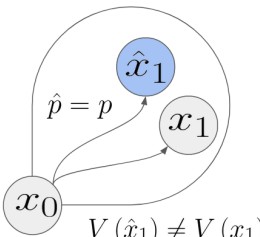

Figure 3: The source of the MuZero bias. Though $x_1$ and $\hat{x}_1$ are drawn from the same distribution, their values do not coincide.

## 4.3 EMPIRICAL VALIDATION OF THE PERFORMANCE IN STOCHASTIC ENVIRONMENTS

To showcase that the bias of MuZero's value learning strategy is not merely a mathematical curiosity, we tested the two losses in the challenging humanoid-run tasks from the DMC benchmarking

suite (Tunyasuvunakool et al., 2020) with different noise distributions applied to the action (details on the environment can be found in Appendix C).

As shown in Figure 4, we do see a clear difference between the performance of $\lambda$-IterVAML and $\lambda$-MuZero when increasing the noise level. At small levels of noise, all algorithms retain their performance. This is expected, since small noise on the actions can increase the robustness of a learned policy or improve exploration (Hollenstein et al., 2022). At large levels of noise, however, the $\lambda$-MuZero algorithm drops in performance to the value-agnostic baseline. We provide additional experiments with more stochastic environments in Appendix D which further substantiate our claims.

It is important to note that humanoid-run is a challenging environment and strong noise corruption increases difficulty, therefore $\lambda$-IterVAML will not retain the full performance in the stochastic version. Furthermore, as highlighted before, the stabilizing latent prediction loss is necessary for $\lambda$-IterVAML and introduces some level of bias (for details see Subsection A.4). However, as all algorithms are impacted by the stabilization term, it is still noticeable that $\lambda$-MuZero's performance drops more sharply. This raises an important direction for future work, establishing a mechanism for stabilizing value-aware losses that does not introduce bias in stochastic environments.

## 5 EVALUATING MODEL CAPACITY AND ENVIRONMENT CHOICE

After investigating the impact of action noise on different loss function, we now present empirical experiments to further investigate how different implementations of $\lambda$ algorithms behave in empirical settings. Theoretical results (Farahmand et al., 2017; Farahmand, 2018) show that value-aware losses perform best when learning a correct model is impossible due to access to finite samples or model capacity. Even though the expressivity of neural networks has increased in recent years, we argue that such scenarios are still highly relevant in practice. Establishing the necessary size of a model a priori is often impossible, since RL is deployed in scenarios where the complexity of a solution is unclear. Increasing model capacity often comes at greatly increased cost (Kaplan et al., 2020), which makes more efficient alternatives desirable. Therefore, we empirically verify under what conditions decision-aware losses show performance improvements over the simple BYOL loss.

For these experiments, we use environments from the popular DMC suite (Tunyasuvunakool et al., 2020), Walker-run, Quadruped-run, and Humanoid-run. These environments were picked from three different levels of difficulty (Hansen et al., 2022) and represent different challenge levels in the benchmark, with Humanoid-run being a serious challenge for most established algorithms.

The algorithmic framework and all model design choices are summarized in Appendix C. The model implementation follows Hansen et al. (2022). Reported rewards are collected during training, not in an evaluation step, the same protocol used by Hansen et al. (2022).

**When do decision-aware losses show performance improvements over the simple BYOL loss?**
The results in Figure 5 show that both MuZero and IterVAML provide a benefit over the BYOL loss in the most challenging environment, Humanoid-run. This is expected, as modelling the full state space of Humanoid-run is difficult with the model architectures we used. For smaller networks, we

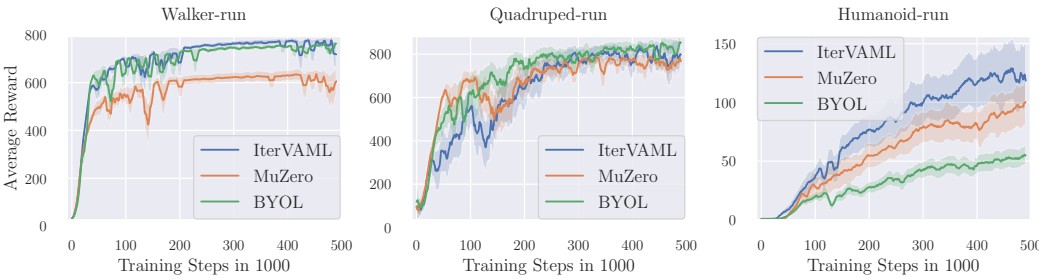

Figure 5: Performance comparison overall test environments. We see that IterVAML performs slightly above MuZero in several environments, but decisive gains from the value-aware losses ($\lambda$-IterVAML, $\lambda$-MuZero) over BYOL can only be observed in the challenging Humanoid-run environment.

see a stable performance benefit in Figure 6 from the MuZero loss. $\lambda$-IterVAML also outperforms the BYOL baseline, but fails to achieve stable returns.

This highlights that in common benchmark environments and with established architectures, value-aware losses are useful in challenging, high-dimensional environments. However, it is important to note that we do not find that the IterVAML or MuZero losses are harmful in deterministic environments, meaning a value-aware loss is always preferable.

The performance improvement of MuZero over IterVAML with very small models is likely due to the effect of using real rewards for the value function target estimation. In cases where the reward prediction has errors, this can lead to better performance over purely model-based value functions such as those used by IterVAML.

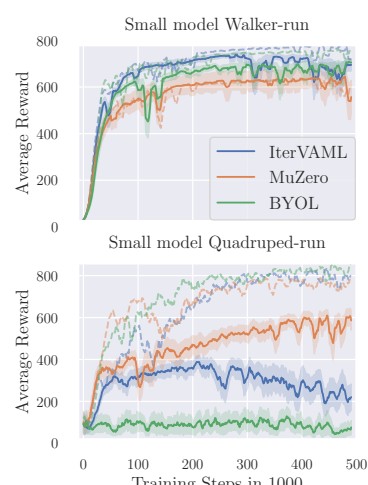

**Can decision-aware models be used for both value function learning and policy improvement?** In all previous experiments, the models were used for both value function target estimation and for policy learning with a model-based gradient. To investigate the performance gain from using the model for policy gradient estimation, we present an ablation on all environments by substituting the model gradient with a simple deep deterministic policy gradient.

As seen in Figure 7, all losses and environments benefit from better policy estimation using the model. Therefore it is advisable to use a $\lambda$ model both for gradient estimation and value function improvement. Compared to MuZero, IterVAML loses more performance without policy gradient computation in the hardest evaluation task. It has been noted that model-based value function estimates might be more useful

Figure 6: Comparison with smaller models. IterVAML is more impacted by decreasing model size on more challenging tasks, due to the lack of real reward signal in the value function loss. Humanoid is omitted as no loss outperforms a random baseline. Dashed lines show the mean results of the bigger models for comparison.

for policy gradient estimation than for value function learning (Amos et al., 2021; Ghugare et al., 2023). In addition, the grounding of the MuZero loss in real rewards likely leads to better value function prediction in the Humanoid-run environment. Therefore a better update can be obtained with MuZero when the model is not used for policy gradient estimation, since learning is driven by the value function update.

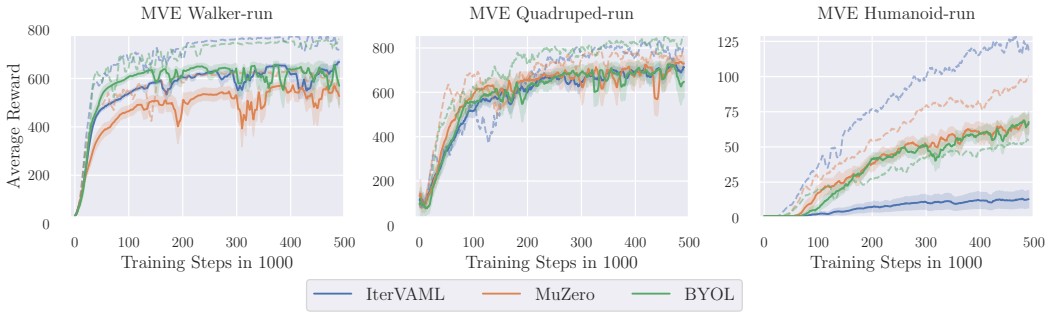

Figure 7: Comparison of the impact removing model policy gradients in all environments. We see decreases in performance across all environments and loss functions, with a drastic decrease in performance for IterVAML in humanoid. Dashed lines show the mean results of the model-based gradient for comparison.

## 6    RELATED WORK

**VAML and MuZero:**  Farahmand (2018) established IterVAML based on earlier work (Farahmand et al., 2017). Several extensions to this formulation have been proposed, such as a VAML-regularized MSE loss (Voelcker et al., 2022) and a policy-gradient aware loss (Abachi et al., 2020). Combining IterVAML with latent spaces was first developed by Abachi et al. (2022), but no experimental results were provided. MuZero (Schrittwieser et al., 2020; Ye et al., 2021) is build based on earlier works which introduce the ideas of learning a latent model jointly with the value function (Silver et al., 2017; Oh et al., 2017). However, none of these works highlight the importance of considering the bias in stochastic environments that result from such a formulation. Antonoglou et al. (2022) propose an extension to MuZero in stochastic environments, but focus on the planning procedure, not the biased value function loss. Hansen et al. (2022) adapted the MuZero loss to continuous control environments but decisiondid not extend their formulation to stochastic variants. Grimm et al. (2020) and Grimm et al. (2021) consider how the set of value equivalent models relates to value functions. They are the first to highlight the close connection between the notions of value-awareness and MuZero.

**Other decision-aware algorithms:**  Several other works propose decision-aware variants that do not minimize a value function difference. D'Oro et al. (2020) weigh the samples used for model learning by their impact on the policy gradient. Nikishin et al. (2021) uses implicit differentiation to obtain a loss for the model function with regard to the policy performance measure. To achieve the same goal, Eysenbach et al. (2022) and Ghugare et al. (2023) choose a variational formulation. Modhe et al. (2021) proposes to compute the advantage function resulting from different models instead of using the value function. Ayoub et al. (2020) presents an algorithm based on selecting models based on their ability to predict value function estimates and provide regret bounds with this algorithm.

**Learning with suboptimal models:**  Several works have focused on the broader goal of using models with errors without addressing the loss functions of the model. Among these, several focus on correcting models using information obtained during exploration (Joseph et al., 2013; Talvitie, 2017; Modi et al., 2020; Rakhsha et al., 2022), or limiting interaction with wrong models (Buckman et al., 2018; Janner et al., 2019; Abbas et al., 2020). Several of these techniques can be applied to improve the value function of a $\lambda$ world model further. Finally, we do not focus on exploration, but Guo et al. (2022) show that a similar loss to ours can be exploited for targeted exploration.

## 7    CONCLUSIONS

In this paper, we investigated model-based reinforcement learning with decision-aware models with three main question focused on (a) implementation of the model, (b) theoretical and practical differences between major approaches in the field, and (c) scenarios in which decision-aware losses help in model-based reinforcement learning.

Empirically, we show that the design decisions established for MuZero are a strong foundation for decision-aware losses. Previous performance differences (Lovatto et al., 2020; Voelcker et al., 2022) can be overcome with latent model architectures. We furthermore establish a formal limitation on the performance of MuZero in stochastic environments, and verify this empirically. Finally, we conduct a series of experiments to establish which algorithmic choices lead to good performance empirically.

Our results highlight the importance of decision-aware model learning in continuous control and allow us to make algorithmic recommendations. When the necessary capacity for an environment model cannot be established, using a decision-aware loss will improve the robustness of the learning algorithm with regard to the model capacity. In deterministic environments with deterministic models, MuZero's value learning approach can be a good choice, as the use of real rewards seemingly provides a grounded learning signal for the value function. In stochastic environments, a model-based bootstrap is more effective, as the model-based loss does not suffer from MuZero's bias.

Overall, we find that decision aware learning is an important addition to the RL toolbox in complex environments where other modelling approaches fail. However, previously established algorithms contain previously unknown flaws or improper design decisions that have made their adoption difficult, which we overcome. In future work, evaluating other design decisions, such as probabilistic models (Ghugare et al., 2023) and alternative exploration strategies (Hansen et al., 2022; Guo et al., 2022) can provide important insights.

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

# A PROOFS AND MATHEMATICAL CLARIFICATIONS

We provide the proofs for Section 4 in this section.

The first proposition relies on the existence of a deterministic mapping, which we prove here as a lemma. The second proposition requires a statement over the minimizers of an equation with a variance-like term, we prove a general result as a lemma.

The proof of the first lemma relies heavily on several propositions from Bertsekas & Shreve (1978), which are restated in Subsection A.3 for reader's convenience. Other topological statements are standard and can be find in textbooks such as Munkres (2018).

**Lemma 1 (Deterministic Representation Lemma)** *Let $\mathcal{X}$ be a compact, connected, metrizable space. Let $p$ be a continuous kernel from $\mathcal{X}$ to probability measures over $\mathcal{X}$. Let $\mathcal{Z}$ be a metrizable space. Consider a bijective latent mapping $\phi : \mathcal{X} \to \mathcal{Z}$ and any $V : \mathcal{Z} \to \mathbb{R}$. Assume that they are both continuous. Denote $V_{\mathcal{X}} = V \circ \phi$.*

*Then there exists a measurable function $f^* : \mathcal{Z} \to \mathcal{Z}$ such that we have $V(f^*(\phi(x))) = \mathbb{E}_p[V_{\mathcal{X}}(x')|x]$ for all $x \in \mathcal{X}$.*

**Proof:** Since $\phi$ is a bijective continuous function over a compact space and maps to a Hausdorff space ($\mathcal{Z}$ is metrizable, which implies Hausdorff), it is a homeomorphism. The image of $\mathcal{X}$ under $\phi$, $\mathcal{Z}_{\mathcal{X}}$ is then connected and compact. Since $\mathcal{X}$ is metrizable and compact and $\phi$ is a homeomorphism, $\mathcal{Z}_{\mathcal{X}}$ is metrizable and compact. Let $\theta_{V,\mathcal{X}}(x) = \mathbb{E}_{x' \sim p(\cdot|x)}[V(x')]$. Then, $\theta_{V,\mathcal{X}}$ is continuous (Proposition 3). Define $\theta_{V,\mathcal{X}} = \theta_{V,\mathcal{Z}} \circ \phi$. Since $\phi$ is a homeomorphism, $\phi^{-1}$ is continuous. The function $\theta_{V,\mathcal{Z}}$ can be represented as a composition of continuous functions $\theta_{V,\mathcal{Z}} = \theta_{V,\mathcal{X}} \circ \phi^{-1}$ and is therefore continuous.

As $\mathcal{Z}_{\mathcal{X}}$ is compact, the continuous function $V$ takes a maximum and minimum over the set $\mathcal{Z}_{\mathcal{X}}$. This follows from the compactness of $\mathcal{Z}_{\mathcal{X}}$ and the extreme value theorem. Furthermore $V_{\min} \le \theta_{V,\mathcal{Z}}(z) \le V_{\max}$ for every $z \in \mathcal{Z}_{\mathcal{X}}$. By the intermediate value theorem over compact, connected spaces, and the continuity of $V$, for every value $V_{\min} \le v \le V_{\max}$, there exists a $z \in \mathcal{Z}_{\mathcal{X}}$ so that $V(z) = v$.

Let $h : \mathcal{Z}_{\mathcal{X}} \times \mathcal{Z}_{\mathcal{X}} \to \mathbb{R}$ be the function $h(z, z') = |\theta_{V,\mathcal{Z}}(z) - V(z')|^2$. As $h$ is a composition of continuous functions, it is itself continuous. Let $h^*(z) = \min_{z' \in \mathcal{Z}_{\mathcal{X}}} h(z, z')$. For any $z \in \mathcal{Z}_{\mathcal{X}}$, by the intermediate value argument, there exist $z'$ such that $V(z') = v$. Therefore $h^*(z)$ can be minimized perfectly for all $z \in \mathcal{Z}_{\mathcal{X}}$.

Since $\mathcal{Z}_{\mathcal{X}}$ is compact, $h$ is defined over a compact subset of $\mathcal{Z}$. By Proposition 4, there exists a measurable function $f^*(z)$ so that $\min_{z'} h(z, z') = h(z, f^*(z)) = 0$. Therefore, the function $f^*$ has the property that $V(f^*(z)) = \mathbb{E}_p[V(z')|z]$, as this minimizes the function $h$.

Now consider any $x \in \mathcal{X}$ and its corresponding $z = \phi(x)$. As $h(z, f^*(z)) = |\theta_{V,\mathcal{Z}}(z) - V(f^*(z))|^2 = 0$ for any $z \in \mathcal{Z}_{\mathcal{X}}$, $V(f^*(\phi(x))) = \theta_{v,\mathcal{Z}}(z) = \mathbb{E}_p[V_{\mathcal{X}}(x')|x]$ as desired.

$\square$

The following lemma shows that a function that minimizes a quadratic and a variance term cannot be the minimum function of the quadratic. This is used to show that the minimum of the MuZero value function learning term is not the same as applying the model-based Bellman operator.

**Lemma 2** *Let $g : \mathcal{X} \to \mathbb{R}$ be a function that is not constant almost everywhere and let $\mu$ be a non-degenerate probability distribution over $\mathcal{X}$. Let $\mathcal{F}$ be an open function space with $g \in \mathcal{F}$. Let $\mathcal{L}(f) = \mathbb{E}_{x \sim \mu}\left[(f(x) - g(x))^2\right] + \mathbb{E}_{x \sim \mu}[f(x)g(x)] - \mathbb{E}_{x \sim \mu}[f(x)]\mathbb{E}_\mu[g(x)]$. Then $g \notin \arg\min_{f \in \mathcal{F}} \mathcal{L}(f)$.*

**Proof:** The proof follows by showing that there is a descent direction from $g$ that improves upon $\mathcal{L}$. For this, we construct the auxiliary function $\hat{g}(x) = g(x) - \epsilon g(x)$. Substituting $\hat{g}$ into $\mathcal{L}$ yields

$$\epsilon^2 \mathbb{E}_\mu[g(x)^2] + \mathbb{E}_\mu[(g(x) - \epsilon g(x))g(x)] - \mathbb{E}_\mu[(g(x) - \epsilon g(x))]\mathbb{E}_\mu[g(x)]$$
$$= \epsilon^2 \mathbb{E}_\mu[g(x)^2] + (1 - \epsilon)\mathbb{E}_\mu[g(x)^2] - (1 - \epsilon)\mathbb{E}_\mu[g(x)]^2.$$

Taking the derivative of this function wrt to $\epsilon$ yields

$$\frac{\mathrm{d}}{\mathrm{d}\epsilon}\epsilon^2 \mathbb{E}_\mu\left[g(x)^2\right] + (1-\epsilon)\mathbb{E}_\mu\left[g(x)^2\right] - (1-\epsilon)\mathbb{E}_\mu\left[g(x)\right]^2$$
$$= 2\epsilon\,\mathbb{E}_\mu\left[g(x)^2\right] - \mathbb{E}_\mu\left[g(x)^2\right] + \mathbb{E}_\mu\left[g(x)\right]^2.$$

Setting $\epsilon$ to 0, obtain

$$\mathbb{E}_\mu\left[g(x)\right]^2 - \mathbb{E}_\mu\left[g(x)^2\right] = \mathrm{Var}_\mu\left[g(x)\right]$$

By the Cauchy-Schwartz inequality, the variance is only 0 for a $g(x)$ constant almost everywhere. However, this violates the assumption. Therefore for any $\epsilon > 0$ $\mathcal{L}(\hat{g}) \leq \mathcal{L}(g)$, due to the descent direction given by $-g(x)$. As we assume that the function class is open, there also exists an $\epsilon > 0$ for which $g(x) - \epsilon g(x) \in \mathcal{F}$.

$\square$

## A.1 MAIN PROPOSITIONS

**Proposition 1** *Let $M$ be an MDP with a compact, connected state space $\mathcal{X} \subseteq \mathcal{Y}$, where $\mathcal{Y}$ is a metrizable space. Let the transition kernel $p$ be continuous. Let $\mathcal{Z}$ be a metrizable space. Consider a bijective latent mapping $\phi : \mathcal{Y} \to \mathcal{Z}$ and any value function approximation $V : \mathcal{Z} \to \mathbb{R}$. Assume that they are both continuous. Denote $V_{\mathcal{X}} = V \circ \phi$.*

*Then there exists a measurable function $f^* : \mathcal{Z} \to \mathcal{Z}$ such that we have $V(f^*(\phi(x))) = \mathbb{E}_p\left[V_{\mathcal{X}}(x')|x\right]$ for all $x \in \mathcal{X}$.*

*Furthermore, the same $f^*$ is a minimizer of the population IterVAML loss:*

$$f^* \in \arg\min_{\hat{f}} \mathbb{E}\left[\hat{\mathcal{L}}_{IterVAML}(\hat{f}; V_{\mathcal{X}})\right].$$

**Proof:** The existence of $f^*$ follows under the stated assumptions (compact, connected and metrizable state space, metrizable latent space, continuity of all involved functions) from Lemma 1.

The rest of the proof follows standard approaches in textbooks such as Györfi et al. (2002). First, expand the equation to obtain:

$$\mathbb{E}\left[\hat{\mathcal{L}}^n_{\text{IterVAML}}(f; V_{\mathcal{X}}, \mathcal{D})\right] = \mathbb{E}\left[\frac{1}{N}\sum_{i=1}^N \left[V\left(f\left(\phi(x_i)\right)\right) - V_{\mathcal{X}}(x_i')\right]^2\right]$$
$$= \mathbb{E}\left[\left[V\left(f\left(\phi(x)\right)\right) - \mathbb{E}\left[V_{\mathcal{X}}(x')|x\right] + \mathbb{E}\left[V_{\mathcal{X}}(x')|x\right] - V_{\mathcal{X}}(x')\right]^2\right].$$

After expanding the square, we obtain three terms:

$$\mathbb{E}\left[\left|V\left(f\left(\phi(x)\right)\right) - \mathbb{E}\left[V_{\mathcal{X}}(x')|x\right]\right|^2\right]$$
$$+2\mathbb{E}\left[\left[V\left(f\left(\phi(x)\right)\right) - \mathbb{E}\left[V_{\mathcal{X}}(x')|x\right]\right]\left[\mathbb{E}\left[V_{\mathcal{X}}(x')|x\right] - V_{\mathcal{X}}(x')\right]\right]$$
$$+\mathbb{E}\left[\left|\mathbb{E}\left[V_{\mathcal{X}}(x')|x\right] - V_{\mathcal{X}}(x')\right|^2\right]$$

Apply the tower property to the inner term to obtain:

$$2\mathbb{E}\left[\left[V\left(f\left(\phi(x)\right)\right) - \mathbb{E}\left[V_{\mathcal{X}}(x')|x\right]\right]\left[\mathbb{E}\left[V_{\mathcal{X}}(x')|x\right] - V_{\mathcal{X}}(x')\right]\right]$$
$$=2\mathbb{E}\left[\left[V\left(f\left(\phi(x)\right)\right) - \mathbb{E}\left[V_{\mathcal{X}}(x')|x\right]\right]\underbrace{\mathbb{E}\left[\mathbb{E}\left[V_{\mathcal{X}}(x')|x\right] - V_{\mathcal{X}}(x')|x'\right]}_{=0}\right] = 0.$$

Since the statement we are proving only applies to the minimum of the IterVAML loss, we will work with the $\arg\min$ of the loss function above. The resulting equation contains a term dependent on $f$ and one independent of $f$:

$$\arg\min_f \; \mathbb{E}\left[\left|V\left(f\left(\phi(x)\right)\right) - \mathbb{E}\left[V_{\mathcal{X}}(x')|x\right]\right|^2\right] + \mathbb{E}\left[\left|\mathbb{E}\left[V_{\mathcal{X}}(x')|x\right] - V_{\mathcal{X}}(x')\right|^2\right]$$

$$= \arg\min_f \; \mathbb{E}\left[\left|V\left(f\left(\phi(x)\right)\right) - \mathbb{E}\left[V_{\mathcal{X}}(x')|x\right]\right|^2\right].$$

Finally, it is easy to notice that $V\left(f^*\left(\phi(x)\right)\right) = \mathbb{E}\left[V_{\mathcal{X}}(x')|x\right]$ by the definition of $f^*$. Therefore $f^*$ minimizes the final loss term and, due to that, the IterVAML loss.

$\square$

**Proposition 2** *Assume a non-deterministic MDP with a fixed, but arbitrary policy $\pi$, and let $p$ be the transition kernel. Let $\mathcal{V}$ be an open set of functions, and assume that it is Bellman complete: $\forall V \in \mathcal{V} : \mathcal{T}V \in \mathcal{V}$.*

*Then for any $V' \in \mathcal{V}$ that is not a constant function, $\mathcal{T}V' \notin \arg\min_{\hat{V} \in \mathcal{V}} \mathbb{E}_{\mathcal{D}}\left[\hat{\mathcal{L}}^1_{MuZero}(p, \hat{V}; \mathcal{D}, V')\right]$.*

**Notation:** For clarity of presentation denote samples from the real environment as $x^{(n)}$ for the $n$-th sample after a starting point $x^{(0)}$. This means that $x^{(n+1)}$ is drawn from $p\left(\cdot|x^{(n)}\right)$. Similarly, $\hat{x}^{(n)}$ is the $n$-th sample drawn from the model, with $\hat{x}^{(0)} = x^{(0)}$. All expectations are taken over $x_i^{(0)} \sim \mu$ where $\mu$ is the data distribution, $\hat{x}_i^{(1)} \sim \hat{p}\left(\cdot\middle|x_i^{(0)}\right)$, $x_i^{(1)} \sim p\left(\cdot\middle|x_i^{(0)}\right)$, and $x_i^{(2)} \sim p\left(\cdot\middle|x_i^{(1)}\right)$. We use the tower property several times, all expectations are conditioned on $x_i^{(0)}{}_i$.

**Proof:** By assumption, let $\hat{p}$ in the MuZero loss be the true transition kernel $p$. Expand the MuZero loss by $\left[r\left(x_i^{(1)}\right) + \gamma V'\left(x_i^{(2)}\right)\right]$ and take its expectation:

$$\mathbb{E}\left[\hat{\mathcal{L}}^1_{\text{MuZero}}(\hat{p}, \hat{V}; \mathcal{D}, V')\right]$$

$$= \mathbb{E}\left[\frac{1}{N}\sum_{i=1}^{N}\left[\hat{V}\left(\hat{x}_i^{(1)}\right) - \left[r\left(x_i^{(1)}\right) + \gamma V'\left(x_i^{(2)}\right)\right]\right]^2\right]$$

$$= \mathbb{E}\left[\left[\hat{V}\left(\hat{x}_i^{(1)}\right) - (\mathcal{T}V')\left(\hat{x}_i^{(1)}\right) + (\mathcal{T}V')\left(\hat{x}_i^{(1)}\right) - \left[r\left(x_i^{(1)}\right) + \gamma V'\left(x_i^{(2)}\right)\right]\right]^2\right]$$

$$= \mathbb{E}\left[\left(\hat{V}\left(\hat{x}_i^{(1)}\right) - (\mathcal{T}V')\left(\hat{x}_i^{(1)}\right)\right)^2\right] + \tag{4}$$

$$2\,\mathbb{E}\left[\left(\hat{V}\left(\hat{x}_i^{(1)}\right) - (\mathcal{T}V')\left(\hat{x}_i^{(1)}\right)\right)\left((\mathcal{T}V')\left(\hat{x}_i^{(1)}\right) - \left[r\left(x_i^{(1)}\right) + \gamma V'\left(x_i^{(2)}\right)\right]\right)\right] + \tag{5}$$

$$\mathbb{E}\left[\left((\mathcal{T}V')\left(\hat{x}_i^{(1)}\right) - \left[r\left(x_i^{(1)}\right) + \gamma V'\left(x_i^{(2)}\right)\right]\right)^2\right] \tag{6}$$

We aim to study the minimizer of this term. The first term (Equation 4) is the regular bootstrapped Bellman residual with a target $V'$. The third term (Equation 6) is independent of $\hat{V}$, so we can drop it when analyzing the minimization problem.

The second term (Equation 5) simplifies to

$$\mathbb{E}\left[\hat{V}\left(\hat{x}_i^{(1)}\right)\left((\mathcal{T}V')\left(\hat{x}_i^{(1)}\right) - \left[r\left(x_i^{(1)}\right) + \gamma V'\left(x_i^{(2)}\right)\right]\right)\right]$$

as the remainder is independent of $\hat{V}$ again.

This remaining term however is not independent of $\hat{V}$ and not equal to $0$ either. Instead, it decomposes into a variance-like term, using the conditional independence of $\hat{x}_i^{(1)}$ and $x_i^{(1)}$ given $x_i^{(0)}$:

$$\mathbb{E}\left[\hat{V}\left(\hat{x}_i^{(1)}\right)\left((\mathcal{T}V')\left(\hat{x}_i^{(1)}\right) - \left[r\left(x_i^{(1)}\right) + \gamma V'\left(x_i^{(2)}\right)\right]\right)\right]$$

$$= \mathbb{E}\left[\hat{V}\left(\hat{x}_i^{(1)}\right)(\mathcal{T}V')\left(\hat{x}_i^{(1)}\right)\right] - \mathbb{E}\left[\hat{V}\left(\hat{x}_i^{(1)}\right)\left[r\left(x_i^{(1)}\right) + \gamma V'\left(x_i^{(2)}\right)\right]\right]$$

$$= \mathbb{E}\left[\hat{V}\left(\hat{x}_i^{(1)}\right)(\mathcal{T}V')\left(\hat{x}_i^{(1)}\right)\right] - \mathbb{E}\left[\hat{V}\left(\hat{x}_i^{(1)}\right)\right]\mathbb{E}\left[\left[r\left(x_i^{(1)}\right) + \gamma V'\left(x_i^{(2)}\right)\right]\right].$$

Combining this with Equation 4, we obtain

$$\mathbb{E}\left[\hat{\mathcal{L}}_{\text{MuZero}}^1(p, \hat{V}; \mathcal{D}, V')\right]$$

$$= \mathbb{E}\left[\left(\hat{V}\left(\hat{x}_i^{(1)}\right) - (\mathcal{T}V')\left(\hat{x}_i^{(1)}\right)\right)^2\right] +$$

$$\mathbb{E}\left[\hat{V}\left(\hat{x}_i^{(1)}\right)(\mathcal{T}V')\left(\hat{x}_i^{(1)}\right)\right] - \mathbb{E}\left[\hat{V}\left(\hat{x}_i^{(1)}\right)\right]\mathbb{E}\left[\left[r\left(x_i^{(1)}\right) + \gamma V'\left(x_i^{(2)}\right)\right]\right].$$

The first summand is the Bellman residual, which is minimized by $\mathcal{T}V'$, which is in the function class by assumption. However, by Lemma 2, $\mathcal{T}V'$ does not minimize the whole loss term under the conditions (open function class, non-constant value functions, and non-degenerate transition kernel).

$\square$

**Discussion:** The proof uses Bellman completeness, which is generally a strong assumption. However, this is only used to simplify showing the contradiction at the end, removing it does not remove the problems with the loss. The proof of Lemma 2 can be adapted to the case where $f(x)$ minimizes the difference to $g(x)$, instead of using $g(x)$ as the global minimum, but some further technical assumptions about the existence of minimizers and boundary conditions are needed. The purpose here is to show that even with very favorable assumptions such as Bellman completeness, the MuZero value function learning algorithm will not converge to an expected solution.

Similarly, the condition of openness of the function class simply ensures that there exists a function "nearby" that minimizes the loss better. This is mostly to remove edge cases, such as the case where the function class exactly contains the correct solution. Such cases, while mathematically valid, are uninteresting from the perspective of learning functions with flexible function approximations.

We only show the proof for the single step version and remove action dependence to remove notational clutter, the action-dependent and multi-step versions follow naturally.

### A.2 COMPARISON OF ITERVAML AND MUZERO FOR MODEL LEARNING

If MuZero is instead used to only update the model $\hat{p}$, we obtain a similarity between MuZero and IterVAML. This result is similar to the one presented in Grimm et al. (2021), so we only present it for completeness sake, and not claim it as a fully novel contribution of our paper. While Grimm et al. (2021) show that the whole MuZero loss is an upper bound on an IterVAML-like term, we highlight the exact term the model learning component minimizes. However, we think it is still a useful derivation as it highlights some of the intuitive similarities and differences between IterVAML and MuZero and shows that they exist as algorithms on a spectrum spanned by different estimates of the target value function.

We will choose a slightly different expansion than before, using $\mathbb{E}_{x_i^{(1)}, x_i^{(2)} \sim p}\left[\left[r\left(x_i^{(1)}\right) + \gamma V'\left(x_i^{(2)}\right)\right]\right] = \mathbb{E}\left[\mathcal{T}V'\left(x_i^{(1)}\right)\right]$

$$\mathbb{E}\left[\hat{\mathcal{L}}^1_{\text{MuZero}}(\hat{p}, V; \mathcal{D}, V')\right]$$

$$= \mathbb{E}\left[\left[V\left(\hat{x}_i^{(1)}\right) - \mathbb{E}\left[\mathcal{T}V'\left(x_i^{(1)}\right)\right] + \mathbb{E}\left[\mathcal{T}V'\left(x_i^{(1)}\right)\right] - \left[r\left(x_i^{(1)}\right) + \gamma V'\left(x_i^{(2)}\right)\right]\right]^2\right]$$

$$= \mathbb{E}\left[\left(V\left(\hat{x}_i^{(1)}\right) - \mathbb{E}\left[\mathcal{T}V'\left(x_i^{(1)}\right)\right]\right)^2\right] + \tag{7}$$

$$2\mathbb{E}\left[\left(V\left(\hat{x}_i^{(1)}\right) - \mathbb{E}\left[\mathcal{T}V'\left(x_i^{(1)}\right)\right]\right)\left(\mathbb{E}\left[\mathcal{T}V'\left(x_i^{(1)}\right)\right] - \left[r\left(x_i^{(1)}\right) + \gamma V'\left(x_i^{(2)}\right)\right]\right)\right] + \tag{8}$$

$$\mathbb{E}\left[\left(\mathbb{E}\left[\mathcal{T}V'\left(x_i^{(1)}\right)\right] - \left[r\left(x_i^{(1)}\right) + \gamma V'\left(x_i^{(2)}\right)\right]\right)^2\right]. \tag{9}$$

The first summand (Equation 8) is similar to the IterVAML loss, instead of using the next state's value function, the one-step bootstrap estimate of the Bellman operator is used.

The third term (Equation 8) is independent of $\hat{p}$ and can therefore be dropped. The second term decomposes into two terms again,

$$\mathbb{E}\left[\left(V\left(\hat{x}_i^{(1)}\right) - \mathbb{E}\left[\mathcal{T}V'\left(x_i^{(1)}\right)\right]\right)\left(\mathbb{E}\left[\mathcal{T}V'\left(x_i^{(1)}\right)\right] - \left[r\left(x_i^{(1)}\right) + \gamma V'\left(x_i^{(2)}\right)\right]\right)\right]$$

$$= \mathbb{E}\left[\left(V\left(\hat{x}_i^{(1)}\right)\right)\left(\mathbb{E}\left[\mathcal{T}V'\left(x_i^{(1)}\right)\right] - \left[r\left(x_i^{(1)}\right) + \gamma V'\left(x_i^{(2)}\right)\right]\right)\right] -$$

$$\mathbb{E}\left[\mathbb{E}\left[\mathcal{T}V'\left(x_i^{(1)}\right)\right]\left(\mathbb{E}\left[\mathcal{T}V'\left(x_i^{(1)}\right)\right] - \left[r\left(x_i^{(1)}\right) + \gamma V'\left(x_i^{(2)}\right)\right]\right)\right].$$

The first summand is equal to 0, due to the conditional independence of $\hat{x}_i^{(1)}$ and $x_i^{(1)}$,

$$\mathbb{E}\left[\left(V\left(\hat{x}_i^{(1)}\right)\right)\left(\mathbb{E}\left[\mathcal{T}V'\left(x_i^{(1)}\right)\right] - \left[r\left(x_i^{(1)}\right) + \gamma V'\left(x_i^{(2)}\right)\right]\right)\right]$$

$$= \mathbb{E}\left[V\left(\hat{x}_i^{(1)}\right)\right]\underbrace{\left(\mathbb{E}\left[\mathbb{E}\left[\mathcal{T}V'\left(x_i^{(1)}\right)\right]\right] - \mathbb{E}\left[\left[r\left(x_i^{(1)}\right) + \gamma V'\left(x_i^{(2)}\right)\right]\right]\right)}_{=0} = 0.$$

The second remaining summand is independent of $\hat{p}$ and therefore irrelevant to the minimization problem.

Therefore, the MuZero model learning loss minimizes

$$\mathbb{E}\left[\hat{\mathcal{L}}^1_{\text{MuZero}}(\hat{p}, V; \mathcal{D}, V')\right] = \mathbb{E}\left[\left(V\left(\hat{x}_i^{(1)}\right) - \mathbb{E}\left[\mathcal{T}V'\left(x_i^{(1)}\right)\right]\right)^2\right].$$

In conclusion, the MuZero loss optimizes a closely related function to the IterVAML loss when used solely to update the model. There are three differences: First, the bootstrap value function estimator is used instead of the value function as the target value. Second, the current value function estimate is used for the model sample and the target network (if used) is applied for the bootstrap estimate. If the target network is equal to the value function estimate, this difference disappears. Finally, the loss does not contain the inner expectation around the model value function. This can easily be added to the loss and its omission in MuZero is unsurprising, as the loss was designed for deterministic environments and models.

The similarity between the losses suggests a potential family of decision-aware algorithms with different bias-variance characteristics, of which MuZero and IterVAML can be seen as two instances. It is also interesting to note that even without updating the value function, the MuZero loss performs an implicit minimization of the difference between the current value estimate and the Bellman operator via the model prediction. This is an avenue for further research, as it might explain some of the empirical success of the method disentangled from the value function update.

### A.3 Propositions from Bertsekas & Shreve (1978)

For convenience, we quote some results from Bertsekas & Shreve (1978). These are used in the proof of Lemma 1.

**Proposition 3 (Proposition 7.30 of Bertsekas & Shreve 1978)** *Let $\mathcal{X}$ and $\mathcal{Y}$ be separable metrizable spaces and let $q(\mathrm{d}y|x)$ be a continuous stochastic kernel on $\mathcal{Y}$ given $\mathcal{X}$. If $f \in \mathcal{C}(\mathcal{X} \times \mathcal{Y})$, the function $\lambda : \mathcal{X} \to \mathbb{R}$ defined by*

$$\lambda(x) = \int f(x,y)q(\mathrm{d}y|x)$$

*is continuous.*

**Proposition 4 (Proposition 7.33 of Bertsekas & Shreve 1978)** *Let $\mathcal{X}$ be a metrizable space, $\mathcal{Y}$ a compact metrizable space, $\mathcal{D}$ a closed subset of $\mathcal{X} \times \mathcal{Y}$, $\mathcal{D}_x = \{y|(x,y) \in \mathcal{D}\}$, and let $f : \mathcal{D} \to \mathbb{R}^*$ be lower semicontinuous. Let $f^* : proj_{\mathcal{X}}(\mathcal{D}) \to \mathbb{R}^*$ be given by*

$$f^*(x) = \min_{y \in \mathcal{D}_x} f(x,y).$$

*Then $proj_{\mathcal{X}}(\mathcal{D})$ is closed in $\mathcal{X}$, $f^*$ is lower semicontinuous, and there exists a Borel-measurable function $\phi : proj_{\mathcal{X}}(\mathcal{D}) \to \mathcal{Y}$ such that $range(\phi) \subset \mathcal{D}$ and*

$$f[x, \phi(x)] = f^*(x), \quad \forall x \in proj_{\mathcal{X}}(\mathcal{D}).$$

In our proof, we construct $f^*$ as the minimum of an IterVAML style loss and equate $\phi$ with the function we call $f$ in our proof. The change in notation is chosen to reflect the modern notation in MBRL – in the textbook, older notation is used.

### A.4 Bias due to the stabilizing loss

As highlighted in Subsection 3.1, the addition of a stabilizing loss is necessary to achieve good performance with any of the loss functions. With deterministic models, the combination $\hat{\mathcal{L}}_{\mathrm{IterVAML}} + \hat{\mathcal{L}}_{\mathrm{latent}}^n$ is stable, but the conditions for recovering the optimal model are not met anymore. This is due to the fact that $\arg\min_{\hat{f}} \mathbb{E}\left[\hat{\mathcal{L}}_{\mathrm{latent}}(\hat{f}, \phi; \mathcal{D})\right] = \mathbb{E}\left[\phi(x')\right]$, but in general $\mathbb{E}\left[V(\phi(x'))\right] \neq V\left(\mathbb{E}\left[\phi(x')\right]\right)$. While another stabilization technique could be found that does not have this problem, we leave this for future work.

### A.5 Multi-step IterVAML

In Subsection 2.1 the multi-step extension of IterVAML is introduced. As MVE and SVG require multi-step rollouts to be effective, it became apparent that simply forcing the one-step prediction of the value function to be correct is insufficient to obtain good performance. We therefore extended the loss into a multi-step variant.

Using linear algebra notation for simplicity, the single-step IterVAML loss enforces

$$\min \left| \langle V, P(x,a) - \hat{P}(x,a) \rangle \right|^2$$

The $n$-step variant then seeks to enforce a minimum between $n$ applications of the respective transition operators

$$\min \left| \langle V, P^n(x,a) - \hat{P}^n(x,a) \rangle \right|^2$$

The sample-based variant is a proper regression target, as $V(x^{(j)}$ is an unbiased sample from $P^n(x,a)$. It is easy to show following the same techniques as used in the proofs of propositions 1 and 2 that the sample-based version indeed minimizes the IterVAML loss in expectation.

Finally, we simply sum over intermediate $n$-step versions which results in the network being forced to learn a compromise between single-step and multi-step fidelity to the value function.

|  | Model loss | Value est. | Policy est. | Actor policy | DAML | Latent |
|---|---|---|---|---|---|---|
| $\lambda$-IterVAML | BYOL | MVE | SVG | direct | ✓ | ✓ |
| $\lambda$-MuZero | BYOL | MuZero | SVG | direct | ✓ | ✓ |
| MuZero (Schrittwieser et al., 2020) | - | MuZero | - | MCTS | ✓ | ✓ |
| Eff.-MuZero (Ye et al., 2021) | - | MuZero | - | MCTS | ✓ | ✓ |
| ALM (Ghugare et al., 2023) | ALM-ELBO | m.-free | ALM-SVG | direct | ✓ | ✓ |
| TD-MPC (Hansen et al., 2022) | BYOL | m.-free | DDPG | MPC | ✓ | ✓ |
| MBPO (Janner et al., 2019) | MLE | SAC | SAC | direct | - | - |
| Dreamer (Hafner et al., 2020) | ELBO | MVE | SVG | direct | - | ✓ |
| IterVAML (Farahmand, 2018) | - | Dyna | - | direct | ✓ | - |
| VaGraM (Voelcker et al., 2022) | weigthed MLE | SAC | SAC | direct | ✓(?) | - |

Table 1: An overview of different model-based RL algorithms and how they fit into the $\lambda$ family. The first two are the empirical algorithms tested in this work. The next section contains work that falls well into the $\lambda$ family as described in this paper. The similarities between these highlight that further algorithms are easily constructed, i.e. ALM (Ghugare et al., 2023) combined with MPC. The final section contains both popular algorithms and closely related work which inform the classification, but are not part of it since they are either not latent or not decision-aware.

## B  $\lambda$-REINFORCEMENT LEARNING ALGORITHMS

We introduce the idea of the $\lambda$ designation in Section 3. The characterization of the family is fairly broad, and it contains more algorithms than those directly discussed in this paper. An overview of related algorithms is presented in Table 1. However, we found it useful to establish that many recently proposed algorithms are closely related and contain components that can be combined freely with one another. While the community treats, i.e. MuZero, as a fixed, well-defined algorithm, it might be more useful to treat it as a certain set of implementation and design decisions under an umbrella of related algorithms. Given the findings of this paper for example, it is feasible to evaluate a MuZero alternative which simply replaces the value function learning algorithm with MVE, which should be more robust in stochastic environments.

A full benchmarking and comparison effort is out of scope for this work. However, we believe that a more integrative and holistic view over the many related algorithms in this family is useful for the community, which is why we present it here.

## C  IMPLEMENTATION DETAILS

### C.1  ENVIRONMENTS

Since all used environments are deterministic, we designed a stochastic extension to investigate the behavior of the algorithms. These are constructed by adding noise to the actions before they are passed to the simulator, but after they are recorded in the replay buffer. We used uniform and mixture of Gaussian noise of different magnitudes to mimic different levels of stochasticity in the environments. Uniform noise is sampled in the interval $[-0.2, 2]$ (uniform-small) and $[-0.4, .4]$ (uniform-large) and added to the action $a$. The Gaussian mixture distribution is constructed by first sampling a mean at $\mu = a - x$ or $a + x$ for noise levels $x$. Then the perturbation it sampled from Gaussians $\mathcal{N}(\mu, \sigma = x)$. For small and large noise we picked $x = 0.1$ and $x = 0.4$ respectively

The experiments in the main body are conducted with the uniform noise formulation, the evaluation over the Gaussian mixture noise is presented in Appendix D.

### C.2  ALGORITHM AND PSEUDOCODE

The full code is provided in the supplementary material and will be publicly released on Github after the review period is over. A pseudo-code for the IterVAML implementation of $\lambda$ is provided for reference (algorithm 1). Instead of taking the simple $n$-step rollout presented in the algorithm, we use an average over all bootstrap targets up to $n$ for the critic estimate, but used a single $n$-step bootstrap target for the actor update. We found that this slightly increases the stability of the TD update. A TD-$\lambda$ procedure as used by Hafner et al. (2020) did not lead to significant performance changes.

---

**Algorithm 1:** $\lambda$-Actor Critic (IterVAML)

---

Initialize , latent encoder $\phi_\theta$, model $\hat{f}_\theta$, policy $\pi_\omega$, value function $Q_\psi$, dataset $\mathcal{D}$;
**for** $i$ *environment steps* **do**
    Take action in env according to $\pi_\phi$; add to $\mathcal{D}$;
    Sample batch $(x_0, a_0, r_0, \ldots, x_n, r_n, a_n)$ from $\mathcal{D}_{\text{env}}$;
    # Model update
    $(z_0, \ldots, z_n) \leftarrow \phi(x_0, \ldots, x_n)$;
    $\hat{z}_0 \leftarrow z_0$;
    $\mathcal{L}_{\text{IterVAML}} \leftarrow 0$;
    $\mathcal{L}_{\text{Reward}} \leftarrow 0$;
    $\mathcal{L}_{\text{Latent}} \leftarrow 0$;
    **for** $i \leftarrow 1;\ i \leq n;\ i \leftarrow i+1$ **do**
        $\hat{z}_i, \hat{r}_{i-1} \leftarrow \hat{f}_\theta(\hat{z}_{i-1}, a_{i-1})$;
        $\mathcal{L}_{\text{IterVAML}} \leftarrow \mathcal{L}_{\text{IterVAML}} + \rho^i \left[ Q(\hat{z}_i, \pi(\hat{z}_i)) - [Q(z_i, \pi(z_i))]_{\text{sg}} \right]^2$;
        $\mathcal{L}_{\text{Reward}} \leftarrow \mathcal{L}_{\text{Reward}} + \rho^i \left[ \hat{r}_{i-1} - r_{i-1} \right]^2$;
        $\mathcal{L}_{\text{Latent}} \leftarrow \mathcal{L}_{\text{Latent}} + \rho^i \left[ \hat{z}_i - [z_i]_{\text{sg}} \right]^2$
    **end**
    $\theta \leftarrow \theta + \alpha_\theta \nabla_\theta \left( \mathcal{L}_{\text{IterVAML}} + \mathcal{L}_{\text{Reward}} + \mathcal{L}_{\text{Latent}} \right)$;
    # RL update
    **for** $i \leftarrow 1;\ i \leq n;\ i \leftarrow i+1$ **do**
        $\hat{z}_i^k \leftarrow \hat{z}_i$;
        $\hat{r} \leftarrow 0$;
        **for** $j \leftarrow 0;\ j < k;\ j \leftarrow j+1$ **do**
            $\hat{z}^k, r_j \leftarrow \hat{f}\left( \hat{z}^k, \pi\left( \hat{z}^k \right) \right)$;
            $\hat{r} \leftarrow \hat{r} + \gamma^j \hat{r}_j$
        **end**
        $J \leftarrow \hat{r} + \gamma^k Q\left( \hat{z}^k, \pi\left( \hat{z}^k \right) \right)$;
        $\mathcal{L}_q = \left[ Q\left( \hat{z}_i, \pi(\hat{z}_i) \right) - [J]_{\text{sg}} \right]^2$;
        $\psi \leftarrow \psi + \alpha_\psi \nabla \mathcal{L}_Q$;
        $\omega \leftarrow \omega + \alpha_\omega \nabla_\omega J$
    **end**
**end**

---

Table 2: Hyperparameters. $[i, j]$ *in* $k$ refers to a linear schedule from $i$ to $j$ over $k$ env steps. The feature dimension was increased for Humanoid.

| RL HP | Value |
|---|---|
| Discount factor $\gamma$ | 0.99 |
| Polyak average factor $\tau$ | 0.005 |
| Batch size | 1024 |
| Initial steps | 5000 |
| Model rollout depth (k) | [0,4] in 25.000 |
| Actor Learning rate | 0.001 |
| Critic Learning rate | 0.001 |
| Grad clip (RL) threshold | 10 |
| hidden_dim | 512 |
| feature_dim | 50 (100 for hum.) |

| Model HP | Value |
|---|---|
| Reward loss coef. | 1.0 |
| Value loss coef. | 1.0 |
| Model depth discounting ($\rho$) | 0.99 |
| Value learning coef. (MuZero) | 0.1 |
| Model learning horizon (n) | 5 |
| Batch size | 1024 |
| Encoder Learning Rate | 0.001 |
| Model Learning Rate | 0.001 |
| Grad clip (model) threshold | 10 |
| hidden_dim | 512 |
| feature_dim | 50 (100 for hum.) |

Table 3: Model architectures. The encoder is shared between all models. The state prediction and reward prediction use the output of the model core layers.

| Encoder layers | Size |
| --- | --- |
| Linear | hidden_dim |
| ELU | – |
| Linear | feature_dim |

| Model core layers | Size |
| --- | --- |
| Linear | hidden_dim |
| ELU | – |
| Linear | hidden_dim |
| ELU | – |
| Linear | hidden_dim |

| State prediction layers | Size |
| --- | --- |
| Linear | hidden_dim |
| ELU | – |
| Linear | hidden_dim |
| ELU | – |
| Linear | feature_dim |

| Reward prediction layers | Size |
| --- | --- |
| Linear | hidden_dim |
| ELU | – |
| Linear | hidden_dim |
| ELU | – |
| Linear | 1 |

| Q function layer | Size |
| --- | --- |
| Linear | hidden_dim |
| LayerNorm, | – |
| Tanh | – |
| Linear | feature_dim |
| ELU | – |
| Linear | 1 |

| Actor layer | Size |
| --- | --- |
| Linear | hidden_dim |
| LayerNorm, | – |
| Tanh | – |
| Linear | feature_dim |
| ELU | – |
| Linear | action_dim |

## C.3    MODEL ARCHITECTURE AND HYPERPARAMETER CHOICES

Similar to previous work in MBRL (Janner et al., 2019) we slowly increased the length of model rollouts for actor and critic training following a linear schedule over the first 25.000 steps. All hyperparameters are detailed in Table 2. Where relevant, variables refer to those used in algorithm 1 for clarity.

All neural networks are implemented as two or three layer MLPs, adapted from Hansen et al. (2022). We found that minor architecture variations, such as adapting the recurrent architecture from Amos et al. (2021) did not have a large impact on the performance of the algorithm in the evaluated environments. Similarly, adding regularization such as Layer Norm or Batch Norm to the model as in Paster et al. (2021) or Ghugare et al. (2023) did not change the outcome of the experiments noticeably. We did keep the LayerNorm in the critic and actor input, as it had some stabilizing effect for the value-aware losses. The full impact of regularization in end-to-end MBRL architectures is an open question that deserves future study.

The full architecture of the model implementation is presented in Table 4 for reference. This architecture was not varied between experiments, except for the feature dimension for Humanoid experiments and the small model experiments where layer count in the model was reduced by 1 and the hidden dimension was reduced to 128.

## D    ADDITIONAL EXPERIMENTS AND ABLATIONS

### D.1    RUNTIME ESTIMATE

Computationally efficiency was estimated on a dedicated machine with a RTX 2090 and rounded to the closest 5 minute mark. Exact runtimes can vary depending on hardware setup. Overall we find that the $\lambda$ algorithms obtains similar runtime efficiency as related approaches which also differentiate the model for policy gradients (Ghugare et al., 2023) and is slightly slower than just using the model

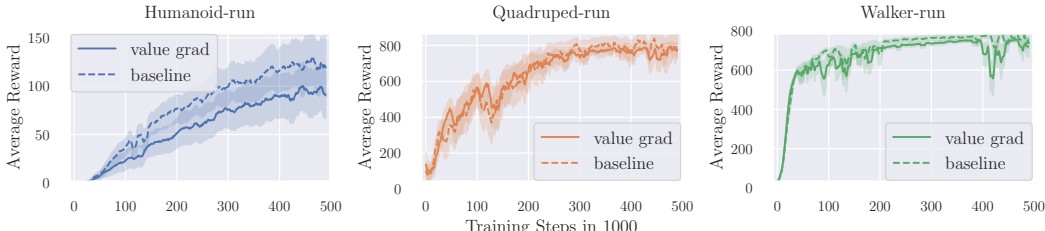

Figure 8: Comparison of using the TD gradient for model learning as well in $\lambda$-IterVAML.

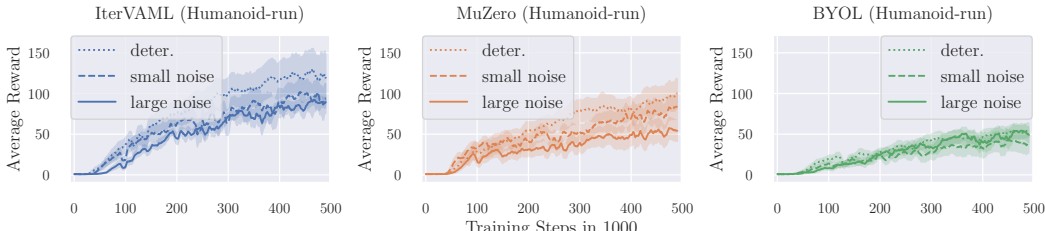

Figure 9: Impact of mixture noise on the learning performance.

for MPC (Hansen et al., 2022). Using the IterVAML loss requires another network pass through the value function compared to MuZero, therefore it is slightly slower in our current setup. However, these differences can most likely be improved with a more efficient implementation with more clever reuse of intermediate computation.

All model-based algorithms have a slowdown of roughly the model depth over SAC ($\times 4 - 6$). This is expected, since the model is rolled out sequentially and the complexity of the computational graph grows linearly with the model rollout depth. Since SAC requires large networks and high UTD to obtain similar performance to the model-based approaches, MBRL is still runtime competitive to model-free RL (compare Ghugare et al. (2023)).

### D.2 GRADIENT PROPAGATION IN ITERVAML

We tested propagating the gradient from the IterVAML value function learning into the model and encoder as well. Results are presented in Figure 8. It is clear that propagating the TD error gradient into the model does not improve the quality of the model. Especially in Humanoid we observe a drop in average performance, although it is not statistically significant under our 99.7 certainty interval.

### D.3 FURTHER EXPERIMENTS IN NOISY ENVIRONMENTS

Here we present the impact of another noise distribution on Humanoid. We tested with the mixture of Gaussian noise as outlined above. The results are presented in Figure 9. It is clear that the same pattern holds independent of the exact form of the noise distribution.

Table 4: Runtime estimates for all used algorithms. Runtime estimates were obtained on a dedicated machine with a RTX 2090 and rounded. Exact runtimes can vary depending on hardware setup.

|  | $\lambda$-IterVAML | $\lambda$-MuZero | BYOL ablation | SAC |
|---|---|---|---|---|
| Runtime (500k) | 7:45h | 5:40h | 5:20h | 1:30h |

### D.3.1 TABULAR BENCHMARK

As discussed, no widely used large-scale benchmark for stochastic MDPs exists. Therefore, we created a small scale benchmark based on the Garnet MDP framework (Archibald et al., 1995) as follows:

We constructed a deterministic Markov Reward Process with a finite state space consisting of states ($|\mathcal{S}| = n$). Its transition matrix is a permutation of the identity matrix. The reward was randomly sampled from a standard normal distribution and an additional reward of 10 was assigned to the first state to further increase the variance of the reward over all states, as this is the critical scenario for MuZero's bias.

To introduce stochasticity, we allowed for $m - 1$ additional successor states for each state $\mathcal{S}_x \subset \mathcal{S}$. We also introduced an additional parameter $\rho$ to interpolate between the deterministic Markov Reward Process and the stochastic one. The first successor state taken from the identity matrix was assigned the transition probability $\rho$ and all additional successor states were assigned probability $(1 - \rho)/(m - 1)$. This allowed us to control the stochasticity of the problem simply by varying $\rho$.

Since finite-state, deterministic models are non differentiable, we instead modeled the impact of the model class bias by constraining the capacity of the model. The model class is constructed from low rank factorized matrices, represented by a random representation matrix $\Phi$ and a learnable weight matrix $\Psi$ of dimensions $(n, k)$, so that $\hat{P} = \sigma(\Phi^T \Psi)$. The probability constraint are ensured via the row-wise softmax operation $\sigma$. By reducing k, the capacity of the model is constrained so that a correct model cannot be represented.

We did not constrain the representation of the value function and simply learned tabular value functions using fitted value iteration with a model-based target estimate and with the MuZero value function learning scheme. We used the original formulation of each loss as accurately as possible, with some minor adaptation for MuZero. This was necessary as the end-to-end gradient through the value function is impossible to compute with discrete samples. Instead, we used the difference between the expected value over the next state and the bootstrapped target value, similarly to the IterVAML formulation presented in **??**.

We conducted our experiment over 16 randomly generated garnets, rewards and representation matrices with $n = 50$ and $m = 10$. The data was sampled from a uniform distribution and next states according to the transition matrix. The reward function was not learned to simplify the setup. We sampled sufficient data points ($100{,}000$) to ensure that an unconstrained MLE model can be learned to high accuracy. The model and value function were learned with full batch gradient descent using Adam. All experiments were simple value estimation for a fixed reward process with constrained model classes; we did not address policy improvement. We varied the constraint k between 10 and 1 to assess the impact of reduced model capacity on the algorithms, with varying $\rho \in [0.5, 0.75, 1.0]$. We also added experiments for $k = 50$ to assess an unconstrained model, although in most cases, performance was already close to optimal with $k = 10$.

The results of these experiments are presented in Table 5.

We found that for all values of stochasticity, the IterVAML model performs best for value function learning. Curiously, in all environments it is matched closely or outperformed slightly by the MLE solution with extremely low values of $k$, which suggests that the model does not have sufficient flexibility to model any value estimate correctly. In these cases, the MLE can provide a more stable learning target, even though none of the evaluated algorithms will be useful under extreme constraints.

Curiously, even in the deterministic case, MuZero does not perform well unless the model capacity is large enough. We investigated this further and found that due to our constraint setup, the model is unable to fully capture deterministic dynamics in the constrained case and will instead remain stochastic. Therefore the bias of the MuZero loss still impacts the system, as the variance over next state values induced by the model is high. We also noticed that the gradient descent based optimization of value function and model in MuZero has a tendency to converge to suboptimal local minima, which might explain some of the extreme outliers.

When replacing the value learning scheme with model-based TD learning, we recover almost identical performance between IterVAML and MuZero, which further highlights the results presented in this paper and by Grimm et al. (2021) that both losses optimize a similar target.

| $\rho = 0.5$ | IterVAML | MuZero | MLE |
|---|---|---|---|
| 1 | 0.9406 +/- 0.0955 | 3.6476 +/- 0.11205 | **0.8417 +/- 0.0532** |
| 2 | 0.9149 +/- 0.02915 | 3.6486 +/- 0.115875 | **0.8671 +/- 0.035975** |
| 3 | 0.8949 +/- 0.07915 | 3.5668 +/- 0.1151 | **0.7687 +/- 0.031925** |
| 4 | **0.5728 +/- 0.0723** | 3.5281 +/- 0.123425 | 0.6941 +/- 0.05015 |
| 5 | **0.29 +/- 0.023825** | 3.3755 +/- 0.136025 | 0.5256 +/- 0.0268 |
| 6 | **0.2325 +/- 0.01965** | 3.1761 +/- 0.116925 | 0.5618 +/- 0.046075 |
| 7 | **0.174 +/- 0.02195** | 3.1889 +/- 0.160375 | 0.4792 +/- 0.047225 |
| 8 | **0.1391 +/- 0.0179** | 2.9965 +/- 0.12555 | 0.3936 +/- 0.038925 |
| 9 | **0.1032 +/- 0.0139** | 2.8649 +/- 0.11805 | 0.3623 +/- 0.029225 |
| 10 | **0.0577 +/- 0.005925** | 2.8381 +/- 0.151475 | 0.273 +/- 0.022625 |
| 50 | **0.0017 +/- 0.000225** | 2.5494 +/- 0.115075 | **0.0017 +/- 0.0002** |

| $\rho = 0.75$ | IterVAML | MuZero | MLE |
|---|---|---|---|
| 1 | 2.8279 +/- 0.283675 | 5.1395 +/- 0.257475 | **2.6313 +/- 0.210275** |
| 2 | **2.7452 +/- 0.1289** | 5.1149 +/- 0.27055 | **2.76 +/- 0.1613** |
| 3 | **1.8714 +/- 0.195725** | 4.8157 +/- 0.27655 | 2.4053 +/- 0.153125 |
| 4 | **0.7264 +/- 0.043275** | 4.5975 +/- 0.2393 | 1.6601 +/- 0.1318 |
| 5 | **0.3794 +/- 0.026425** | 4.3195 +/- 0.2547 | 1.1971 +/- 0.042725 |
| 6 | **0.2066 +/- 0.016675** | 3.4099 +/- 0.24005 | 0.9698 +/- 0.0657 |
| 7 | **0.1213 +/- 0.013075** | 3.1378 +/- 0.231275 | 0.5805 +/- 0.05625 |
| 8 | **0.1086 +/- 0.006675** | 3.4061 +/- 0.167275 | 0.6449 +/- 0.080075 |
| 9 | **0.0799 +/- 0.009** | 3.0232 +/- 0.23425 | 0.4361 +/- 0.042275 |
| 10 | **0.0456 +/- 0.0036** | 3.2613 +/- 0.29505 | 0.2999 +/- 0.028375 |
| 50 | **0.0027 +/- 0.00025** | 2.5132 +/- 0.194225 | **0.0026 +/- 0.000275** |

| $\rho = 1.$ | IterVAML | MuZero | MLE |
|---|---|---|---|
| 1 | 14.0457 +/- 1.50455 | 16.5522 +/- 1.79805 | 14.2642 +/- 1.527775 |
| 2 | 13.0414 +/- 1.100675 | 16.9774 +/- 1.636875 | 13.7411 +/- 1.30145 |
| 3 | **6.1846 +/- 0.688375** | 11.9329 +/- 1.19215 | 9.5593 +/- 1.193025 |
| 4 | **1.1092 +/- 0.219475** | 3.335 +/- 0.7467 | 2.248 +/- 0.355225 |
| 5 | **0.0977 +/- 0.026175** | 0.3616 +/- 0.107525 | 0.2503 +/- 0.08265 |
| 6 | **0.01 +/- 0.0031** | 36.276 +/- 23.983825 | **0.0042 +/- 0.001** |
| 7 | 0.0065 +/- 0.0028 | 0.0134 +/- 0.001125 | 0.0052 +/- 0.001325 |
| 8 | 0.0083 +/- 0.0021 | 0.0128 +/- 0.001275 | 0.0045 +/- 0.001475 |
| 9 | 0.0032 +/- 0.0004 | 0.0117 +/- 0.001075 | 0.0021 +/- 0.000425 |
| 10 | 0.0054 +/- 0.001675 | 0.0092 +/- 0.00105 | 0.0035 +/- 0.000725 |
| 50 | 0.0023 +/- 0.00035 | 0.0048 +/- 0.0013 | 0.0044 +/- 0.000775 |

Table 5: Garnet experiment results. Each cell depicts mean and standard deviation of value estimation over 16 randomly generated garnet MDPs for each algorithm. Minimum value highlighted (when ambiguous due to confidence interval, multiple values are highlighted). For the case of $\rho = 1$, all losses achieve strong performance with $k \geq 7$, therefore no highlighting was necessary.

Overall, these experiments provide further evidence that the MuZero value learning scheme is flawed in stochastic environments, and with stochastic models.

## D.4 REMOVING LATENT FORMULATION AND STABILIZING LOSS

Finally, we present the results of removing both the latent and the auxiliary stabilization in all $\lambda$ algorithms in Figure 10. IterVAML diverges catastrophically, with value function error causing numerical issues, which is why the resulting graph is cut off: no run in 8 seeds was able to progress beyond 100.000 steps. This is similar to results described by Voelcker et al. (2022). MuZero on the other hand profits from not using a stabilizing BYOL loss when operating in observation space. This substantiates claims made by Tang et al. (2022), which conjecture that latent self-prediction losses such as BYOL can capture good representation of MDPs for reinforcement learning but reconstruction accuracy might not be a useful goal.

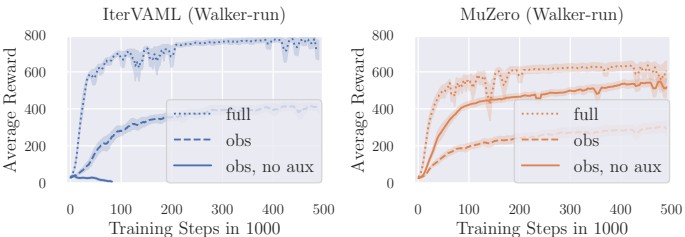

Figure 10: Comparison for removing both the latent formulation and the stabilization loss

