# OpenReview forum: "$\lambda$-AC: Effective decision-aware reinforcement learning with latent models"
_ICLR.cc/2024/Conference — Submitted to ICLR 2024_

### Official Review · Reviewer_gnhB · 2023-10-30

**Soundness:** 4 excellent
**Presentation:** 4 excellent
**Contribution:** 3 good
**Rating:** 8
**Confidence:** 2

**Summary:**

The paper proposes a new framework for analyzing decision-aware model learning, which aims to learn a model for the environment and requires the model to sufficiently accurately represent the value functions associated with the environment. The paper first shows the relationship between existing SOTA methods IterVAML and MuZero. The paper then proposes $\lambda$-AC as a unifying framework for analyzing the two algorithms, and investigates the design choices that lead to the different performances the two algorithms achieve on various continuous control tasks.

**Strengths:**

- The paper is well-written and easy to follow. The discussions offer good insight into the design choices made by the paper and the contribution of the work in relation to prior literature.
- The included experiments are comprehensive and illustrates how design choices and the environments themselves lead to the performance differences between MuZero and IterVAML.
- The provided experiment details are comprehensive and should be sufficient for replicating the results in the paper.

**Weaknesses:**

1. While the paper is well-written overall, there are some technical details and notations that can make key formulas harder to parse (see the minor comments below). For a paper with relatively heavy notation, as it needs to consider ground-truth models, estimated models, transitions collected from interactions, and transitions from learned models, such mistakes, while understandable, can make some key concepts hard to grasp. As someone with little prior knowledge on MuZero and IterVAML, I found the mathematical details behind both losses hard to understand.
2. The legends and labels in the figures are slightly confusing. The paper's text seems to use $\lambda$-MuZero and $\lambda$-IterVAML to refer to the variants of the two losses under the $\lambda$-AC framework, and MuZero and IterVAML to refer to the "vanilla" versions of the algorithms. However, I cannot find $\lambda$-MuZero nor $\lambda$-IterVAML in figures 4 - 7, which seems to suggest that the experiments are done on the "vanilla" algorithms themselves. Is this the correct interpretation?

Minor Comment
1. I am slightly concerned by the correctness of eq (1) and eq (2) as written here. Particularly, eq (1) takes expectation over some $\mu ∈ \Delta(S × A)$ (even though here the paper lets $\mu$ be a distribution over the state space only), whereas in eq (2) the samples follow a distribution where the initial state $x\_{i\_1}$ is drawn from some $\mu ∈ \Delta(S)$, and actions $a$ and subsequent states are drawn from the ground-truth transition kernel $p$ (which generates $\mathcal{D}$). Without any assumption on $\mu$, I am not sure if (2) provides an unbiased estimate of (1). Note that the cited work (Farahmand, 2018) also seems to take expectation over some joint distribution over state and action.
2. In Algorithm 1, $\mathcal{L}\_{\rm Latent}$ is used instead of $\mathcal{L}\_{\rm latent}$ and the notation is not consistent.
3. In eq (4), is $\hat{f}^{(j)}$ a typo? Should it be $\hat{f}^j$ instead? If not, what is the relation ship between the two?
4. Shouldn't the sample-based versions of MuZero and IterVAML depend on some policy $\pi$ as well? Aren't the actions $a\_{i\_j}$ are collected by some particular policy $\pi$?

**Questions:**

1. Would it be possible to generalize the $\lambda$-AC framework to other decision-aware losses, such as the ones discussed in the related work section?
2. I am not familiar with either MuZero or IterVAML, so my assumption may be misguided. However, shouldn't it be expected that MuZero will always have some bias? From equation (3), my understanding is that the loss function uses some deterministic mapping to estimate the transition. As such, the assumption is inherently not compatible with stochastic transition. Is this intuition oversimplifying the problem? (Of course this is not to say the theoretical results are trivial or not interesting. Rigorous proofs cannot be replaced by hand-waving "analysis".)

---

> ### Author Response · Authors · 2023-11-17
> **Author reply**
>
> We thank you for your thorough review and your detailed suggestions! We acknowledge that there are several typos that make key concepts harder to grasp, and we have used your feedback to update the manuscript. As a general rule of thumb, we aim to use superscript (i) to denote an i-th sample, while we aim to use superscript i without brackets to denote power operations such as chained autoregressive model applications. The places you pointed out are indeed inconsistencies in this usage.
>
> Addressing other writing clarifications:
>
> > “The legends and labels in the figures are slightly confusing.”
>
> This is an oversight on our part, we only compare the lambda variants in this paper. We will change the labels to reflect this.
>
> > “I am slightly concerned by the correctness of eq (1) and eq (2) as written here.”
>
> The expectation is indeed meant to be taken over the state-action space, we have fixed this omission.
>
> > “Shouldn't the sample-based versions of MuZero and IterVAML depend on some policy as well?”
>
> Both are written for an off-policy case, so actions can be taken from any number of policies. We clarify this in the updated draft. The off-policyness is not as relevant for the model loss as it would be for a value function loss, therefore previous work has not investigated this in detail and we follow this convention.
>
> > “However, shouldn't it be expected that MuZero will always have some bias?”
>
> This is a very good question: in part, this is the (somewhat surprising) result of our first proposal: if value awareness is the goal, a deterministic model can be sufficient and no stochastic transition model is needed. Intuitively the reason is that instead of aiming to predict a next state distribution, we are only interested in the expectation of the value function, which is a simple scalar. This scalar is all that is needed for value iteration or TD learning. The proof highlights that this scalar can be predicted with a deterministic function.
>
> In addition, proof 2 highlights that even with a stochastic model, the MuZero value function learning algorithm will be biased (potentially quite badly). Note that our proof places no assumptions on the implementation of the model. This is the second highly surprising result, as merely fixing the model architecture is insufficient to fully adapt MuZero and descendant algorithms to stochastic environments. Our tabular experiments in the appendix highlight this fact: we use stochastic models there (just with constrained capacity), and the MuZero algorithm is still incapable of learning the correct value function.
>
> You are completely correct to point out that MuZero (as well as almost any other algorithm in the field) has some bias due to the choice of model class, irrespective of whether stochastic or deterministic models are used. However, the bias we show is separate from expected bias-variance tradeoffs and is irreducible with better models or more samples. The problem we highlight exists _in addition_ to the regular approximation error that any model class incurs when predicting a complex function, which makes it unique from the problems that arise from simply picking the wrong function class for the problem.

---

### Official Review · Reviewer_fdkF · 2023-11-01

**Soundness:** 2 fair
**Presentation:** 3 good
**Contribution:** 2 fair
**Rating:** 6
**Confidence:** 3

**Summary:**

This paper studies decision-aware model-based reinforcement learning, in which the objective of model learning also takes into account the value of the policy. It proposes the $\lambda$-AC framework for characterizing such model-based algorithms, which includes a latent model, a decision-aware model loss, and a model-based actor-critic algorithm. Specifically, it focuses on analyzing decision-aware model losses from two existing approaches, IterVAML and Muzero: it shows that IterVAML can learn a sound expectation (deterministic) model for stochastic environments under some conditions. At the same time, Muzero doesn’t share this nice property. Empirical results of instantiations with the two losses on a difficult task validate this theoretical finding. In addition, the paper also provides some other insights on $\lambda$-AC algorithms.

**Strengths:**

The main strength of this paper is its originality. The paper shows the soundness of learning an expectation (or deterministic) model using IterVAML in stochastic environments for the first time, which sheds light on the promising approach to learning deterministic models. Meanwhile, it also establishes the issue of MuZero’s value loss. These original results may be of interest to relevant model-based RL researchers and inspire further research.

In addition, the paper is well-organized and easy to follow in general. Nevertheless, here are some suggestions for improving the clarity further:
1. Including a table like Table 1 in the main text may be helpful when explaining the $\lambda$-AC framework and the instances. In addition, the caption of Table 1 seems to be outdated.
2. Since the weaker performance of MuZero on the walker-run seems to be an outlier, consider using results on another, more representative task.

**Weaknesses:**

Speaking of weaknesses, the paper is weak in its significance and soundness, in my opinion. For the significance part, the paper mainly focuses on analyzing two existing value-aware losses and obtains a few insights that only apply to the two specific losses. In addition, the paper proposes a framework that contains three components, while only two instances pivoting the value-aware loss component are investigated. It may be worthwhile to step further and understand the effect of other components.

On the soundness of the paper, some statements are not well justified:
1. It is claimed that “$\lambda$-IterVAML leads to an unbiased solution in the infinite sample limit, [conditions]…” However, as discussed at the bottom of Page 4, Proposition 1 only shows the existence of such an unbiased solution. It’s not immediately apparent that $\lambda$-IterVAML *leads to* it. If this is an implied result, it may be helpful to clarify this.
2. In the caption of Figure 6, the performance decrease of IterVAML is explained to be due to the lack of real reward signal in the value function loss, which is not supported by evidence.

**Questions:**

1. This question doesn’t impact the assessment. Are there results similar to Proposition 1 when $\mathcal{X}$ is a discrete space? If not, how likely could there be such a result?

Minor clarification questions and typos that don’t have an impact on the assessment:
1. On page 2, “​​refer to approximate model[s]”
2. Be consistent with the style of superscripts. For example, the n-step deterministic model is $\hat f^j$ in Eq. (3) but $\hat f^{(j)}$ in Eq. (4). For another example, in Section 2.2, there are $\hat x^0$ vs. $x^{(0)}$ and $\hat x^j$ vs. $\hat x^{(j)}$, which appears to be the same variable.
3. On page 5, “stabilizing loss” is used without a definition or introduction. From the context, it can be inferred that it’s $\mathcal{L}^n_{\text{latent}}$. However, it is quite confusing.
4. On page 5, “sepcifically”
5. On page 5, “compare Section 5” seems to be grammatically incorrect.
6. On page 5, there should be a comma after “In Proposition 1”
7. On page 6, there is something wrong in “the bias to impact the solution”
8. On page 6, what is “the model’s value function”?
9. On page 9, redundant period “..”

---

> ### Author Response · Authors · 2023-11-17
> **Author reply**
>
> We thank you for your thorough review and for the many helpful comments and questions. We have updated our draft using your suggestions, and address the detailed questions in the following. We are also working on expanding our experimental results to strengthen our claims, efforts are ongoing.
>
> > “[...] the paper mainly focuses on analyzing two [...] losses and obtains a few insights that only apply to the two specific losses
>
> We focus on these two losses as they are the main losses considered in the literature on value-aware/value equivalent model learning. Other approaches do exist, and a wider comparison is indeed interesting, but we decided that the “core” approaches deserved a closer analysis. We would also like to point out that MuZero is not a niche approach, but a landmark approach in model-based RL with almost 2000 citations in the field. Its loss has been used in several follow up works and can be considered a “backbone” component of the field, comparable to Dreamer. We therefore think that our results are highly relevant also for the many other algorithms building on MuZero.
>
> > “If this is an implied result, it may be helpful to clarify this”
>
> The proof indeed only shows existence and we point this out below proposition 1 in the main text. Showing convergence is a significantly harder problem which we decided not to tackle in this paper. Even for well established algorithms like MLE, we cannot obtain general convergence results with deep models. We still think the inclusion of this proof is relevant to the community because it shows that the goal of learning value-aware models, even with deterministic backbones, is not impossible. This result has been a core argument used in the MuZero line of work, starting from Value Prediction Networks, but no proof of the conjecture existed in the literature so far. The necessity of invoking complex measure theoretic topics highlights that this is a highly non-trivial result.
>
> In our opinion, the necessity of the stabilizing loss points to the fact that the learning problem is hard and convergence is not directly implied by existence here. Further analysis is indeed necessary, yet out of scope for this work.
>
> > “In the caption of Figure 6, the performance decrease of IterVAML is explained to be due to the lack of real reward signal in the value function loss, which is not supported by evidence.”
>
> We agree that this is a conjecture that is not fully proven by the evidence and we should have been more careful in the caption. We will clarify the claim to be in accordance with the main text. To be clear: this is our working hypothesis as to why the difference between the different approaches was so drastic, even though we expected them to behave very similarly in this experiment.
>
>
> > “Are there results similar to Proposition 1 when is a discrete space? If not, how likely could there be such a result?”
>
> This is an excellent question, and the answer is no, not immediately. Intuitively the problem is that the average value over a set of discrete points does not have to correspond to the value of any single one of them, so no deterministic mapping is possible. The main technical problem stems from the fact that no meaningful continuous topology can be defined on a discrete space. However the introduction of a continuous representation space “fixes” this. If the discrete space is represented as a set of points embedded in a continuous space (i.e. by encoding each state as a one-hot vector and allowing the model to predict continuous vectors) in which the value function is also continuous, that space can be used to obtain our proposition. We will add this note to the appendix with more formal language.

---

> > ### Comment · Reviewer_fdkF · 2023-11-22
> >
> > Thank you very much for your reply. It helps me resolve some of the confusion. I still think it is a limitation only to consider two losses, but I agree that the study based on these two losses could contribute to the field. However, I also agree with Reviewer Y2Uk and LuyF that the current experiment results are not significant enough. I will reevaluate the paper once the broader experimental evaluation is provided.

---

### Official Review · Reviewer_LuyF · 2023-11-01

**Soundness:** 2 fair
**Presentation:** 2 fair
**Contribution:** 2 fair
**Rating:** 3
**Confidence:** 2

**Summary:**

This paper investigated the $\lambda$-AC framework for model-based reinforcement learning with decision-aware models. It intensively compares the performance of three different loss functions - IterVAML, MuZero, and BYOL. This paper is interested in showing what components of algorithms lead to performance differences in practice. It shows that with a sufficiently flexible function class, IterVAML can recover an optimal deterministic model for value function prediction. And MuZero is a biased method such that it will not recover the correct value function in stochastic environments even if the correct model is used and the function class is Bellman complete. With theoretical analysis, it shows that MuZero is most susceptible to the noise among all three loss functions. This paper also empirically shows that decision-aware losses IterVAML and MuZero have better performance over the simple BYOL loss in changeling tasks for both value function learning and policy improvement.

**Strengths:**

1. From a theoretical perspective, this paper did a mathematical analysis for decision-aware losses IterVAML and MuZero. It confirms that IterVAML is able to recover an optimal deterministic model for value function prediction. But MuZero is a biased method such that it will not recover the correct value function in stochastic environments even with the correct model and Bellman completeness.

2. Their empirical results show that MuZero is most susceptible to the noise among all three loss functions. This observation supports their theoretical results.

3. They empirically show that decision-aware losses IterVAML and MuZero have better performance over the simple BYOL loss in challenging tasks for both value function learning and policy improvement.

**Weaknesses:**

1. This paper assumes too much background on the reader. It uses jargon without clearly and sufficiently introducing them, for example, latent model, decision-aware learning framework, IterVAML, MuZero, BYOL loss, and so on. Most importantly, it is very hard to figure out what is the contribution of this paper. Both the introduction and the conclusion did not clearly point the main contribution out.

2. The readability of this paper could be greatly improved by deleting unnecessary words and sentences. More tables should be introduced in place of large paragraphs of words.

3. The author should directly articulate their research goal at the beginning of the research paper. Currently, readers cannot understand the research goal until the first full pass of the paper.

4. In terms of contribution, a comparison among three different loss functions in three environments may not be significant enough to offer strong insights. And the novelty of this work is limited because it is a direct completeness extension and evaluation of previous works cited in section 2.1.

**Questions:**

Why do you choose IterVAML, MuZero, and BYOL loss functions as benchmarks to compare? Are they broad enough to give a representative comparison of model-based RL methods?

---

> ### Author Response · Authors · 2023-11-11
> **Request for clarification**
>
> Dear reviewer, thank you for your review. We will post a more detailed answer soon addressing your feedback. To properly engage with the points raised by your review, we would quickly like to ask for some clarifications. Which sentences did you feel did not contribute to your understanding of the paper, and what tables or figures would be helpful to replace these? We look forward to hearing from you.

---

> ### Author Response · Authors · 2023-11-17
> **Author reply**
>
> Dear reviewer, thank you for your comments. We will address them one by one in detail here.
>
> > This paper assumes too much background on the reader. It uses jargon without clearly and sufficiently introducing them
>
> We are happy to improve our writing. We present background descriptions in Section 2 and would love to hear what parts are not sufficiently well explained? We have added additional text explaining the difference between observation-space and latent space models. If additional clarification is needed, we are happy two write an expanded background section in the appendix.
>
> > Both the introduction and the conclusion did not clearly point the main contribution out.
>
> Regarding the contribution, the final 4 paragraphs of the introduction as well as the conclusion list our main insights. We have updated the draft to put more clear emphasis on these:
>
> _In this work, we ask three major question: *(a)* what design decisions explain the performance difference between IterVAML and Muzero, *(b)* what are the significant theoretical or practical differences between the two, and *(c)* in what scenarios do decision-aware losses help in model-based reinforcement learning?_
>
> _[...]_
>
> _In summary, our contributions are *a)* showing that IterVAML is a stable loss when a latent model is used, *b)* proving and verifying that the MuZero value loss is biased in stochastic environments, and *c)* showing how these algorithms provide benefit over decision-unaware baseline losses._
>
> > More tables should be introduced in place of large paragraphs of words.
>
> We are happy to strengthen our writing, however we are not fully sure how tables could replace the insights presented in the text? What particular tables would you like to see? Which of the text do you consider redundant or not informative? We will be happy to streamline the writing.
>
> > The author should directly articulate their research goal at the beginning of the research paper.
>
> We present the main contributions in the introduction, with a summary in the last paragraph. We have updated the writing accordingly (note the quote in the previous paragraph). We are happy to include suggestions of things that could be added to make the goal clearer?
>
> > A comparison among three different loss functions in three environments may not be significant enough to offer strong insights.
>
> We will answer this together with the following question on why we focus on IterVAML and MuZero, and do not compare against Dreamer: We purposefully chose VAML and MuZero because they are the main approaches in the family of value equivalent learning. Our paper is not meant (and not introduced) as a survey on model-based RL, but as a comparison of value-aware/value-equivalent approaches. BYOL has been used in several papers as an additional stabilizing loss, so it is a natural ablation. It could also be replaced by Dreamer, but Dreamer or other models have not been adapted in the framework of decision-aware learning before. Our insights are specifically targeted at the two major approaches to the value equivalent model family.
>
> > And the novelty of this work is limited because it is a direct completeness extension and evaluation of previous works cited in section 2.1
>
> We want to gently push back against the claim that there is limited novelty in a thorough evaluation. Our work is in line with similar papers at top conferences (such as “Hamrick et al. On the role of planning in model-based deep reinforcement learning” published at ICLR 2021), which aim to understand the problems of existing methods instead of proposing additional ones. Our proofs are new contributions to the literature and touch upon previously unknown *and completely unrecognized* issues in a landmark algorithm in the field in addition to highlighting a possible solution. We do not merely extend MuZero, we show that it is an incorrect algorithm in a large number of important scenarios.
>
> We are working on providing additional experiments in the appendix to broaden our basis of comparison.

---

### Official Review · Reviewer_Y2Uk · 2023-11-01

**Soundness:** 2 fair
**Presentation:** 3 good
**Contribution:** 2 fair
**Rating:** 5
**Confidence:** 4

**Summary:**

The paper investigates the known decision-aware models like InterVAML and MuZero. It comes up with a framework called $\lambda$-AC that includes both these models. The authors discuss the benefits of decision-aware models over other models like BYOL and how different design choices affect the performance. The authors evaluate these models on continuous domain tasks for which they modify models like MuZero were designed for discrete action spaces.

**Strengths:**

(1) The paper discusses the different design choices of MuZero and InterVAML and their effects on their performance.

(2) They show that in stochastic dynamics, InterVAML produces unbiased results but MuZero produces biased value functions.

(3) They adapt these models to continuous domains over which they compare these with BYOL.

**Weaknesses:**

(1) The authors do compare design choices, and raise research questions but the story is still incomplete. They do not come up with any answer to these questions. They do not present any new algorithm or an unknown insight.

(2) The framework $\lambda$-AC seems to be vague. Towards the end, when the authors discuss about using model for policy learning or not, both of these will fall under this framework as per the definition: "and an actor-critic algorithm to obtain policy".

(3) If presenting an evaluation paper, why not compare more model-based methods like Dreamer and using discrete settings as well.

**Questions:**

(1) A small preliminary on model based value gradients like SVG should be presented.

(2) When it is established in the first experiment, that the auxiliary loss does not add the MuZero, why is it still used as it is adding additional bias. What will happen if I use the MuZero directly (without auxiliary loss) in Section 4.3?

---

> ### Author Response · Authors · 2023-11-11
> **Request for clarification**
>
> Dear reviewer, thank you for your review. We will be happy to address your concerns in our paper. As we prepare a full detailed answer, we would quickly like to ask for some clarifications. Could you point out which questions raised in the introduction are unanswered in your opinion? We will cover this in our response.

---

> ### Author Response · Authors · 2023-11-17
> **Author reply**
>
> We thank the reviewer for their review and for recognizing that our work analyzes design choices, shows bias properties, and adapts the considered models to continuous domains.
>
> *To address the major questions:*
>
> > “ … the story is still incomplete”
>
> Our main contributions are listed in the introduction. We have updated the draft slightly to put more emphasis on these.
> From the updated draft (Section 1: Introduction):
>
> _In this work, we ask three major question: *(a)* what design decisions explain the performance difference between IterVAML and Muzero, *(b)* what are the significant theoretical or practical differences between the two, and *(c)* in what scenarios do decision-aware losses help in model-based reinforcement learning?_
>
> _[...]_
>
> _In summary, our contributions are *a)* showing that IterVAML is a stable loss when a latent model is used, *b)* proving and verifying that the MuZero value loss is biased in stochastic environments, and *c)* showing how these algorithms provide benefit over decision-unaware baseline losses._
>
> We ask why previous work on the VAML family has failed to provide strong empirical results, which we answer by highlighting the different network design decisions in previous work as well as the importance of additional stabilizing losses. We then ask if there are relevant differences between the MuZero and VAML approaches, which we answer with both theoretical and empirical results. Previous work has treated both of this as almost identical, while we highlight there are relevant differences. Finally, we evaluate how well the two losses compare to the decision-unaware ablation when used in different scenarios, such as models with different sizes, more difficult tasks, and other model-based actor critic strategies.
>
> If there are issues with this framing, we will seek to improve our writing and ask the reviewer to advise what questions they feel are unanswered?
>
> > “They do not come up with any answer to these questions. They do not present any new algorithm or an unknown insight.”
>
> We would like to point out that all of our core mathematical results are novel and the bias in MuZero is not discussed in the literature at all. While we do not propose a novel algorithm, we present novel insights, such as a bias affecting the landmark algorithm MuZero, as well as a big oversights in previous work on IterVAML. We also highlight how to approach these issues (use MVE and VAML instead of the MuZero value function learning scheme), and prove a conjecture on the applicability of deterministic latent models in stochastic environments. This is in line with similar papers at top conferences (such as “Hamrick et al. On the role of planning in model-based deep reinforcement learning” published at ICLR 2021), which aim to understand the problems of existing methods instead of proposing additional ones.
>
> > “The framework -AC seems to be vague.”
>
> We use the term lambda-AC as a way to highlight similarities, not differences, between the presented algorithms. It is correct that both variants of the algorithms presented fall into the same framework, several other extensions do as well. We point this out in detail in Section B in the appendix.
>
> We acknowledge that the term can lead readers to believe that $\lambda$-AC is intended as a novel algorithm. We have updated the writing accordingly to clarify our usage of the term.
>
> > If presenting an evaluation paper, why not compare more model-based methods like Dreamer and using discrete settings as well.
> We purposefully do not compare to Dreamer or many other model-based RL methods, because they do not fall into the family of value equivalent learning and therefore insights are not naturally transferable to these other cases. Our paper is not meant (and not introduced) as an evaluation or survey on model-based RL, but as an investigation of value-aware/value-equivalent approaches, of which we focus on the two most prominent examples.
>
> *Regarding the minor questions:*
>
> The second paragraph of 2.2 addresses SVG, however, the term is not mentioned explicitly there, only the eponymous paper is. We corrected that oversight in the updated draft.
>
> We are working on expanding our experimental evaluation with a test of using the MuZero loss without stabilization in other environments. Preliminary results show that, due to the similarities with the IterVAML loss, we do not obtain good performance this way. Note that experiment number 1 tests the removal of the latent encoder, not the removal of the stabilizing loss.

---

> > ### Comment · Reviewer_Y2Uk · 2023-11-21
> >
> > Thanks a lot for your response. The story is clearer now. I agree that the analysis that you have put forward is valuable for the community. However, the writing needs to be improved. Also, if you are analyzing how and when these losses help over decision-unaware losses, you should consider more environments and more baselines. Out of three environments, decision-aware losses improved only in Humanoid which is understandable but it was just one environment. You should have presented several environments where these losses are providing performance improvements and studied what is common in these environments.
> >
> > I have increased my score as the story has improved but the paper still requires further work.

---

### Author Response · Authors · 2023-11-17
**General response**

We thank the reviewers for their thoughtful feedback and are happy to see our work praised for its originality (reviewer fdkF), its structure and readability (reviewers fdkF and gnhB) and the good insights it offers (gnhB). We are happy to see that reviewers highlight our comprehensive experiments (gnhB) and its potential to inspire future research (fdkF).


We have used the suggestions and comments to strengthen our writing. An updated draft is uploaded with changes highlighted. Due to some formatting issues with the diff tool, we provide two copies, one with the updates highlighted for convenience, and after the appendix, the full updated draft with proper formatting. We mostly
* clarified the research question and contributions (reviewers Y2Uk, LuyF),
* fixed notational inconsistencies (reviewers fdkF, gnhB), and
* clarified used terms more precisely (especially the difference between latent- and observation-space models and the stochastic value gradient algorithm).


In addition, we were convinced by the reviewer feedback to de-emphasize the $\lambda$-AC family. We still think it is important to highlight that many related algorithms have strong similarities and their individual components can be freely combined. However, we acknowledge that our framing led several reviewers to understand our papers more as a general survey paper, instead of a detailed investigation into core algorithms at the heart of decision-aware learning. We still use $\lambda$ to denote our empirical implementation, to highlight the vitally important role that the latent model plays, but we have removed it from the introduction and the list of contributions.

Finally, we are working on providing broader experimental evaluation. The updated draft does not yet contain these, but we will update it as soon as possible. We are furthermore working on obtaining baseline results for additional MBRL algorithms, as requested. While we do not think they are necessary to understand our core contributions, we agree that they are important to situate our results in the field.

We provide individual responses to each reviewer as well with detailed answers to raised questions.

---

### Meta-Review · Area_Chair_YD8f · 2023-12-07

**Metareview:**

a) Contributions: This paper compares the performance of three different decision-aware loss functions in three different environments.  It introduces a framework ($\lambda$-AC) to generalize three axes of design decisions, and investigates the effect of varying one of these dimensions.

b) Strengths: The reviewers agree that this work considers an important question; the comparisons that it makes are likely to be of interest to model-based RL researchers.

c) Weaknesses: After discussion, the reviewers were unanimous that the paper's experimental evaluation was incomplete.  For the kinds of claims that the paper wants to make, three loss functions in three environments were not sufficient.  Most reviewers had strong concerns about the paper's clarity.  The paper requires a careful pass of editing to ensure that all claims are correct and/or justified by the evidence that the paper provides.

**Justification For Why Not Higher Score:**

My main concerns from the discussion were widespread correctness issues in the writing and insufficient empirical evaluations.

**Justification For Why Not Lower Score:**

n/a

---

### Decision · Program_Chairs · 2024-01-16

Reject